# A Continuous Time Framework for Discrete Denoising Models

**Andrew Campbell**[1]    **Joe Benton**[1]    **Valentin De Bortoli**[2]

**Tom Rainforth**[1]    **George Deligiannidis**[1]    **Arnaud Doucet**[1]

[1]Department of Statistics, University of Oxford, UK    [2]CNRS ENS Ulm, Paris, France
{campbell, benton, rainforth, deligian, doucet}@stats.ox.ac.uk
valentin.debortoli@gmail.com

## Abstract

We provide the first complete continuous time framework for denoising diffusion models of discrete data. This is achieved by formulating the forward noising process and corresponding reverse time generative process as Continuous Time Markov Chains (CTMCs). The model can be efficiently trained using a continuous time version of the ELBO. We simulate the high dimensional CTMC using techniques developed in chemical physics and exploit our continuous time framework to derive high performance samplers that we show can outperform discrete time methods for discrete data. The continuous time treatment also enables us to derive a novel theoretical result bounding the error between the generated sample distribution and the true data distribution.

## 1   Introduction

Diffusion/score-based/denoising models [1, 2, 3, 4] are a popular class of generative models that achieve state-of-the-art sample quality with good coverage of the data distribution [5] all whilst using a stable, non-adversarial, simple to implement training objective. The general framework is to define a forward noising process that takes in data and gradually corrupts it until the data distribution is transformed into a simple distribution that is easy to sample. The model then learns to reverse this process by learning the logarithmic gradient of the noised marginal distributions known as the score.

Most previous work on denoising models operates on a continuous state space. However, there are many problems for which the data we would like to model is discrete. This occurs, for example, in text, segmentation maps, categorical features, discrete latent spaces, and the direct 8-bit representation of images. Previous work has tried to realize the benefits of the denoising framework on discrete data problems, with promising initial results [6, 7, 8, 9, 10, 11, 12, 13].

All of these previous approaches train and sample the model in discrete *time*. Unfortunately, working in discrete time has notable drawbacks. It generally forces the user to pick a partition of the process at training time and the model only learns to denoise at these fixed time points. Due to the fixed partition, we are then limited to a simple ancestral sampling strategy. In continuous time, the model instead learns to denoise for any arbitrary time point in the process. This complete specification of the reverse process enables much greater flexibility in defining the reverse sampling scheme. For example, in continuous state spaces, continuous time samplers that greatly reduce the sampling time have been devised [14, 15, 16, 17] as well as ones that improve sample quality [4, 18]. The continuous time interpretation has also enabled the derivation of interesting theoretical properties such as error bounds [19] in continuous state spaces.

36th Conference on Neural Information Processing Systems (NeurIPS 2022).

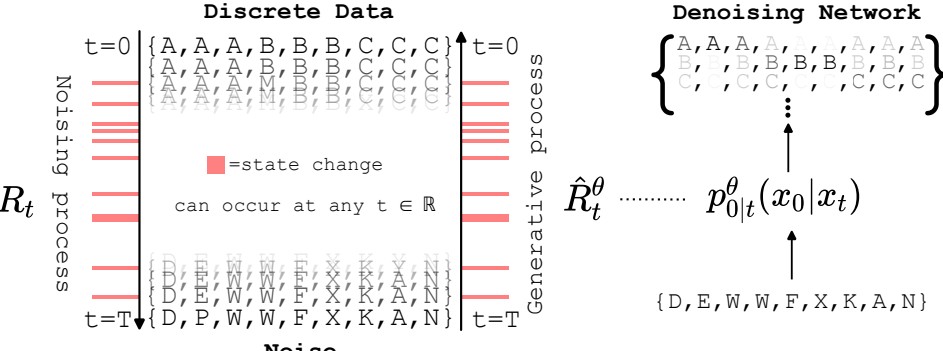

Figure 1: The forward noising process corrupts data according to $R_t$, the rate of corruption events at time $t$. The noising process' time reversal gives the generative process which is defined through $\hat{R}_t^\theta$, the rate of generative events at time $t$. $\hat{R}_t^\theta$ is parameterized through the denoising network, $p_{0|t}^\theta(x_0|x_t)$, which outputs categorical probabilities over clean $x_0$ values conditioned on a noisy $x_t$.

To allow these benefits to be exploited for discrete state spaces as well, we formulate a continuous time framework for discrete denoising models. Specifically, our contributions are as follows. We formulate the forward noising process as a Continuous Time Markov Chain (CTMC) and identify the generative CTMC that is the time-reversal of this process. We then bound the log likelihood of the generated data distribution, giving a continuous time equivalent of the ELBO that can be used for efficient training of a parametric approximation to the true generative reverse process. To efficiently simulate the parametric reverse process, we leverage tau-leaping [20] and propose a novel predictor-corrector type scheme that can be used to improve simulation accuracy. The continuous time framework allows us to derive a bound on the error between the true data distribution and the samples generated from the approximate reverse process simulated with tau-leaping. Finally, we demonstrate our proposed method on the generative modeling of images from the CIFAR-10 dataset and monophonic music sequences. Notably, we find our tau-leaping with predictor-corrector sampler can provide higher quality CIFAR10 samples than previous discrete time discrete state approaches, further closing the performance gap between when images are modeled as discrete data or as continuous data.

Proofs for all propositions and theorems are given in the Appendix.

## 2  Background on Discrete Denoising Models

In the discrete time, discrete state space case, we aim to model discrete data $x_0 \in \mathcal{X}$ with finite cardinality $S = |\mathcal{X}|$. We assume $x_0 \sim p_{\text{data}}(x_0)$ for some discrete data distribution $p_{\text{data}}(x_0)$. We define a forward noising process that transforms $p_{\text{data}}(x_0)$ to some distribution $q_K(x_K)$ that closely approximates an easy to sample distribution $p_{\text{ref}}(x_K)$. This is done by defining forward kernels $q_{k+1|k}(x_{k+1}|x_k)$ that all admit $p_{\text{ref}}$ as a stationary distribution and mix reasonably quickly. For example, one can use a simple uniform kernel [6, 8], $q_{k+1|k}(x_{k+1}|x_k) = \delta_{x_{k+1},x_k}(1-\beta) + (1-\delta_{x_{k+1},x_k})\beta/(S-1)$ where $\delta$ is a Kronecker delta. The corresponding $p_{\text{ref}}$ is the uniform distribution over all states. Other choices include: an absorbing state kernel—where for each state there is a small probability that it transitions to some absorbing state—or a discretized Gaussian kernel—where only transitions to nearby states have significant probability (valid for spaces with ordinal structure) [8].

After defining $q_{k+1|k}$, we have a forward joint decomposition as follows

$$q_{0:K}(x_{0:K}) = p_{\text{data}}(x_0)\prod_{k=0}^{K-1} q_{k+1|k}(x_{k+1}|x_k).$$

The joint distribution $q_{0:K}(x_{0:K})$ also admits a reverse decomposition:

$$q_{0:K}(x_{0:K}) = q_K(x_K)\prod_{k=0}^{K-1} q_{k|k+1}(x_k|x_{k+1}) \text{ where } q_{k|k+1}(x_k|x_{k+1}) = \frac{q_{k+1|k}(x_{k+1}|x_k)q_k(x_k)}{q_{k+1}(x_{k+1})}.$$

Here $q_k(x_k)$ denotes the marginal of $q_{0:K}(x_{0:K})$ at time $k$. If one had access to $q_{k|k+1}$ and could sample $q_K$ exactly, then samples from $p_{\text{data}}(x_0)$ could be produced by first sampling $x_K \sim q_K(\cdot)$ and then ancestrally sampling the reverse kernels, i.e. $x_k \sim q_{k|k+1}(\cdot|x_{k+1})$.

However, in practice, $q_{k|k+1}$ is intractable and needs to be approximated with a parametric reverse kernel, $p^\theta_{k|k+1}$. This kernel is commonly defined through the analytic $q_{k|k+1,0}$ distribution and a parametric 'denoising' model $p^\theta_{0|k+1}$ [6, 8],

$$p^\theta_{k|k+1}(x_k|x_{k+1}) \triangleq \sum_{x_0} q_{k|k+1,0}(x_k|x_{k+1},x_0)p^\theta_{0|k+1}(x_0|x_{k+1})$$

$$= q_{k+1|k}(x_{k+1}|x_k) \sum_{x_0} \frac{q_{k|0}(x_k|x_0)}{q_{k+1|0}(x_{k+1}|x_0)} p^\theta_{0|k+1}(x_0|x_{k+1}). \qquad (1)$$

Though $q_K(x_K)$ is also intractable, for large $K$ we can reliably approximate it with $p_{\text{ref}}(x_K)$. Note that the faster the transitions mix, the more accurate this approximation becomes. Approximate samples from $p_{\text{data}}(x_0)$ can then be obtained by sampling the generative joint distribution

$$p^\theta_{0:K}(x_{0:K}) = p_{\text{ref}}(x_K) \prod_{k=0}^{K-1} p^\theta_{k|k+1}(x_k|x_{k+1}),$$

where $\theta$ is trained through minimizing the negative discrete time (DT) ELBO which is an upper bound on the negative model log-likelihood

$$\mathbb{E}_{p_{\text{data}}(x_0)}\left[-\log p^\theta_0(x_0)\right] \leq \mathbb{E}_{q_{0:K}(x_{0:K})}\left[-\log \frac{p^\theta_{0:K}(x_{0:K})}{q_{1:K|0}(x_{1:K}|x_0)}\right] = \mathcal{L}_{\text{DT}}(\theta).$$

It was shown in [1] that $\mathcal{L}_{\text{DT}}$ can be re-written as

$$\mathcal{L}_{\text{DT}}(\theta) = \mathbb{E}_{p_{\text{data}}(x_0)}\Big[\text{KL}(q_{K|0}(x_K|x_0)||p_{\text{ref}}(x_K)) - \mathbb{E}_{q_{1|0}(x_1|x_0)}\Big[\log p^\theta_{0|1}(x_0|x_1)\Big]$$

$$+ \sum_{k=1}^{K-1} \mathbb{E}_{q_{k+1|0}(x_{k+1}|x_0)}\Big[\text{KL}(q_{k|k+1,0}(x_k|x_{k+1},x_0)||p^\theta_{k|k+1}(x_k|x_{k+1}))\Big]\Big]$$

where KL is the Kullback–Leibler divergence. The forward kernels $q_{k+1|k}$ are chosen such that $q_{k|0}(x_k|x_0)$ can be computed efficiently in a time independent of $k$. With this, $\theta$ can be efficiently trained by taking a random selection of terms from $\mathcal{L}_{\text{DT}}$ in each minibatch and performing a stochastic gradient step.

## 3 Continuous Time Framework

### 3.1 Forward process and its time reversal

Our method is built upon a continuous time process from $t = 0$ to $t = T$. State transitions can occur at any time during this process as opposed to the discrete time case where transitions only occur when one of the finite number of transition kernels is applied (see Figure 1). This process is known as a Continuous Time Markov Chain (CTMC), we provide a short overview of CTMCs in Appendix A for completeness. Giving an intuitive introduction here, we can define a CTMC through an initial distribution $q_0$ and a transition rate matrix $R_t \in \mathbb{R}^{S \times S}$. If the current state is $\tilde{x}$, then the transition rate matrix entry $R_t(\tilde{x}, x)$ is the instantaneous rate (occurrences per unit time) at which state $\tilde{x}$ transitions to state $x$. Loosely speaking, the next state in the process will likely be one for which $R_t(\tilde{x}, x)$ is high, and furthermore, the higher the rate is, the less time it will take for this transition to occur.

It turns out that the transition rate, $R_t$, also defines the infinitesimal transition probability for the process between the two time points $t - \Delta t$ and $t$

$$q_{t|t-\Delta t}(x|\tilde{x}) = \delta_{x,\tilde{x}} + R_t(\tilde{x}, x)\Delta t + o(\Delta t),$$

where $o(\Delta t)$ represents terms that tend to zero at a faster rate than $\Delta t$. Comparing to the discrete time case, we see that $R_t$ assumes an analogous role to the discrete time forward kernel $q_{k+1|k}$ in how we define the forward process. Therefore, just as in discrete time, we design $R_t$ such that: i) the forward process mixes quickly towards an easy to sample (stationary) distribution, $p_{\text{ref}}$, (e.g. uniform), ii) we can analytically obtain $q_{t|0}(x_t|x_0)$ distributions to enable efficient training (see Section 4.1 for how this is done). We initialize the forward CTMC at $q_0(x_0) = p_{\text{data}}(x_0)$ at time $t = 0$. We denote the marginal at time $t = T$ as $q_T(x_T)$, which should be close to $p_{\text{ref}}(x_T)$.

We now consider the time reversal of the forward process, which will take us from the marginal $q_T(x_T)$ back to the data distribution $p_{\text{data}}(x_0)$ through a reverse transition rate matrix, $\hat{R}_t \in \mathbb{R}^{S \times S}$:

$$q_{t|t+\Delta t}(\tilde{x}|x) = \delta_{\tilde{x},x} + \hat{R}_t(x, \tilde{x})\Delta t + o(\Delta t).$$

In discrete time, one uses Bayes rule to go from $q_{k+1|k}$ to $q_{k|k+1}$. We can use similar ideas to calculate $\hat{R}_t$ from $R_t$ as per the following result.

**Proposition 1.** *For a forward in time CTMC, $\{x_t\}_{t\in[0,T]}$, with rate matrix $R_t$, initial distribution $p_{\text{data}}(x_0)$ and terminal distribution $q_T(x_T)$, there exists a CTMC with initial distribution $q_T(x_T)$ at $t = T$, terminal distribution $p_{\text{data}}(x_0)$ at $t = 0$ and transition rate matrix $\hat{R}_t$ that runs backwards in time and is almost everywhere equivalent to the time reversal of the forward CTMC, $\{x_t\}_{t\in[T,0]}$. Furthermore, $\hat{R}_t$ is related to $R_t$ by the following expression*

$$\hat{R}_t(x,\tilde{x}) = R_t(\tilde{x},x) \sum_{x_0} \frac{q_{t|0}(\tilde{x}|x_0)}{q_{t|0}(x|x_0)} q_{0|t}(x_0|x) \quad for \quad x \neq \tilde{x},$$

*where $q_{t|0}(x|x_0)$ are the conditional marginals of the forward process and $q_{0|t}(x_0|x) = q_{t|0}(x|x_0)p_{\text{data}}(x_0)/q_t(x)$ with $q_t(x)$ being the marginal of the forward process at time $t$. When $x = \tilde{x}$, $\hat{R}_t(x,x) = -\sum_{x'\neq x} \hat{R}_t(x,x')$ because the rows must sum to zero (see Appendix A).*

Unfortunately, $\hat{R}_t$ is intractable due to the intractability of $q_t(x)$ and thus of $q_{0|t}(x_0|x)$. Therefore, we consider an approximation $\hat{R}_t^\theta$ of $\hat{R}_t$ by approximating $q_{0|t}(x_0|x)$ with a parametric denoising model, $p_{0|t}^\theta(x_0|x)$:

$$\hat{R}_t^\theta(x,\tilde{x}) = R_t(\tilde{x},x) \sum_{x_0} \frac{q_{t|0}(\tilde{x}|x_0)}{q_{t|0}(x|x_0)} p_{0|t}^\theta(x_0|x) \quad \text{for} \quad x \neq \tilde{x}$$

and $\hat{R}_t^\theta(x,x) = -\sum_{x'\neq x} \hat{R}_t^\theta(x,x')$ as before. As a further analogy to the discrete time case, notice that when $x \neq \tilde{x}$, $\hat{R}_t^\theta$ has the same form as the discrete time parametric reverse kernel, $p_{k|k+1}^\theta$ defined in eq (1) but with the forward kernel, $q_{k+1|k}$, replaced by the forward rate, $R_t$.

## 3.2 Continuous Time ELBO

In discrete time, $\theta$ is trained by minimizing the discrete time negative ELBO, $\mathcal{L}_{\text{DT}}$, formed from the forward and reverse processes. We mirror this approach in continuous time by minimizing the corresponding continuous time (CT) negative ELBO, $\mathcal{L}_{\text{CT}}$, as derived below.

**Proposition 2.** *For the reverse in time CTMC with initial distribution $p_{\text{ref}}(x_T)$, terminal distribution $p_0^\theta(x_0)$, and reverse rate $\hat{R}_t^\theta$, an upper bound on the negative model log-likelihood, $\mathbb{E}_{p_{\text{data}}(x_0)}[-\log p_0^\theta(x_0)]$, is given by*

$$\mathcal{L}_{\text{CT}}(\theta) = T\,\mathbb{E}_{t\sim\mathcal{U}(0,T)q_t(x)r_t(\tilde{x}|x)} \left[ \left\{ \sum_{x'\neq x} \hat{R}_t^\theta(x,x') \right\} - \mathcal{Z}^t(x) \log\left( \hat{R}_t^\theta(\tilde{x},x) \right) \right] + C,$$

*where $C$ is a constant independent of $\theta$ and*

$$\mathcal{Z}^t(x) = \sum_{x'\neq x} R_t(x,x') \qquad r_t(\tilde{x}|x) = (1-\delta_{\tilde{x},x})R_t(x,\tilde{x})/\mathcal{Z}^t(x).$$

Here $r_t(\tilde{x}|x)$ gives the probability of transitioning from $x$ to $\tilde{x}$, given that we know a transition occurs at time $t$. We can optimize this objective efficiently with stochastic gradient descent. For a gradient update, we sample a batch of datapoints from $p_{\text{data}}(x_0)$, noise each datapoint using a random time, $t \sim \mathcal{U}(0,T)$, $x \sim q_{t|0}(x|x_0)$ and finally sample an auxiliary $\tilde{x}$ from $r_t(\tilde{x}|x)$ for each $x$. Intuitively, $(x,\tilde{x})$ are a pair of states following the forward in time noising process. Minimizing the second term in $\mathcal{L}_{\text{CT}}$ maximizes the reverse rate for this pair, but going in the backwards direction, $\tilde{x}$ to $x$. This is how $\hat{R}_t^\theta$ learns to reverse the noising process. Intuition on the first term and a direct comparison to $\mathcal{L}_{\text{DT}}$ is given in Appendix C.1.

The first argument of $\hat{R}_t^\theta$ is input into $p_{0|t}^\theta$ so we naively require two network forward passes on $x$ and $\tilde{x}$ to evaluate the objective. We can avoid this by approximating the $q_t(x)$ sample in the first term with $\tilde{x}$ meaning we need only evaluate the network once on $\tilde{x}$. The approximation is valid because, as we show in Appendix C.4, $\tilde{x}$ is approximately distributed according to $q_{t+\delta t}$ for $\delta t$ very small.

# 4 Efficient Forward and Backward Sampling

## 4.1 Choice of Forward Process

The transition rate matrix $R_t$ needs to be chosen such that the forward process: i) mixes quickly towards $p_{\text{ref}}$, and ii) the $q_{t|0}(x|x_0)$ distributions can be analytically obtained. The Kolmogorov

differential equation for the CTMC needs to be integrated to obtain $q_{t|0}(x|x_0)$. This can be done analytically when $R_t$ and $R_{t'}$ commute for all $t, t'$, see Appendix E. An easy way to meet this condition is to let $R_t = \beta(t)R_b$ where $R_b \in \mathbb{R}^{S \times S}$ is a user-specified time independent base rate matrix and $\beta(t) \in \mathbb{R}$ is a time dependent scalar. We then obtain the analytic expression

$$q_{t|0}(x = j|x_0 = i) = \left( Q\exp\left[\Lambda \int_0^t \beta(s)ds\right] Q^{-1} \right)_{ij}$$

where $R_b = Q\Lambda Q^{-1}$ is the eigendecomposition of matrix $R_b$ and $\exp[\cdot]$ the element-wise exponential.

Our choice of $\beta$ schedule is guided by [3, 4], $\beta(t) = ab^t \log(b)$. The hyperparameters $a$ and $b$ are selected such that $q_T(x) \approx p_{\text{ref}}(x)$ at the terminal time $t = T$ while having a steady speed of 'information corruption' which ensures that $\hat{R}_t$ does not vary quickly in a short span of time.

We experiment with a variety of $R_b$ matrices, for example, a uniform rate, $R_b = \mathbb{1}\mathbb{1}^T - S\text{Id}$, where $\mathbb{1}\mathbb{1}^T$ is a matrix of ones and Id is the identity. For problems with a heavy spatial bias, e.g. images, we can instead use a forward rate that only encourages transitions to nearby states; details and the links to the corresponding discrete time processes can be found in Appendix E.

## 4.2 Factorizing Over Dimensions

Our aim is to model data that is $D$ dimensional, with each dimension taking one value from $S$ possibilities. We now slightly redefine notation and say $\boldsymbol{x}^{1:D} \in \mathcal{X}^D$, $|\mathcal{X}| = S$. In this setting, calculating transition probabilities naively would require calculating $S^D$ rate values corresponding to each of the possible next states. This is intractable for any reasonably sized $S$ and $D$. We avoid this problem simply by factorizing the forward process such that each dimension propagates independently. Since this is a continuous time process and each dimension's forward process is independent of the others, the probability two or more dimensions transition at exactly the same time is zero. Therefore, overall in the full dimensional forward CTMC, each transition only ever involves a change in exactly one dimension. For the time reversal CTMC, it will also be true that exactly one dimension changes in each transition. This makes computation tractable because of the $S^D$ rate values, only $D \times (S - 1) + 1$ are non-zero - those corresponding to transitions where exactly one dimension changes plus the no change transition. Finally, we note that even though dimensions propagate independently in the forward direction, they are not independent in the reverse direction because the starting points for each dimension's forward process are not independent for non factorized $p_{\text{data}}$. The following proposition shows the exact forms for the forward and reverse rates in this case.

**Proposition 3.** *If the forward process factorizes as* $q_{t|s}(\boldsymbol{x}_t^{1:D}|\boldsymbol{x}_s^{1:D}) = \prod_{d=1}^D q_{t|s}(x_t^d|x_s^d)$, $t > s$, *then the forward and reverse rates are of the form*

$$R_t^{1:D}(\tilde{\boldsymbol{x}}^{1:D}, \boldsymbol{x}^{1:D}) = \sum_{d=1}^D R_t^d(\tilde{x}^d, x^d)\delta_{\boldsymbol{x}^{1:D\setminus d}, \tilde{\boldsymbol{x}}^{1:D\setminus d}},$$

$$\hat{R}_t^{1:D}(\boldsymbol{x}^{1:D}, \tilde{\boldsymbol{x}}^{1:D}) = \sum_{d=1}^D R_t^d(\tilde{x}^d, x^d)\delta_{\boldsymbol{x}^{1:D\setminus d}, \tilde{\boldsymbol{x}}^{1:D\setminus d}} \sum_{x_0^d} q_{0|t}(x_0^d|\boldsymbol{x}^{1:D})\frac{q_{t|0}(\tilde{x}^d|x_0^d)}{q_{t|0}(x^d|x_0^d)},$$

*where $R_t^d \in \mathbb{R}^{S \times S}$ and $\delta_{\boldsymbol{x}^{1:D\setminus d}, \tilde{\boldsymbol{x}}^{1:D\setminus d}}$ is 1 when all dimensions except for $d$ are equal.*

To find $\hat{R}_t^{\theta\,1:D}$ we simply replace $q_{0|t}(x_0^d|\boldsymbol{x}^{1:D})$ with $p_{0|t}^\theta(x_0^d|\boldsymbol{x}^{1:D})$ which is easily modeled with a neural network that outputs conditionally independent state probabilities in each dimension. In Appendix C.3 we derive the form of $\mathcal{L}_{\text{CT}}$ when we use this factorized form for $R_t^{1:D}$ and $\hat{R}_t^{\theta\,1:D}$.

## 4.3 Simulating the Generative Reverse Process with Tau-Leaping

The parametric generative reverse process is a CTMC with rate matrix $\hat{R}_t^{\theta\,1:D}$. Simulating this process from distribution $p_{\text{ref}}(\boldsymbol{x}_T^{1:D})$ at time $t = T$ back to $t = 0$ will produce approximate samples from $p_{\text{data}}(\boldsymbol{x}_0^{1:D})$. The process could be simulated exactly using Gillespie's Algorithm [21, 22, 23] which alternates between i) sampling a holding time to remain in the current state and ii) sampling a new state according to the current rate matrix, $\hat{R}_t^{\theta\,1:D}$ (see Appendix F). This is inefficient for large $D$ because we would need to step through each transition individually and so only one dimension would change for each simulation step.

Instead, we use tau-leaping [20, 23], a very popular approximate simulation method developed in chemical physics. Rather than step back through time one transition to the next, tau-leaping leaps

from $t$ to $t-\tau$ and applies all transitions that occurred in $[t-\tau, t]$ simultaneously. To make a leap, we assume $\hat{R}_t^{\theta\,1:D}$ and $\boldsymbol{x}_t^{1:D}$ remain constant in $[t-\tau, t]$. As we propagate from $t$ to $t-\tau$, we count all of the transitions that occur, but hold off on actually applying them until we reach $t-\tau$, such that $\boldsymbol{x}_t^{1:D}$ remains constant in $[t-\tau, t]$. Assuming $\hat{R}_t^{\theta\,1:D}$ and $\boldsymbol{x}_t^{1:D}$ remain constant, the number of times a transition from $\boldsymbol{x}_t^{1:D}$ to $\tilde{\boldsymbol{x}}^{1:D}$ occurs in $[t-\tau, t]$ is Poisson distributed with mean $\tau \hat{R}_t^{\theta\,1:D}(\boldsymbol{x}_t^{1:D}, \tilde{\boldsymbol{x}}^{1:D})$. Once we reach $t-\tau$, we apply all transitions that occurred simultaneously i.e. $\boldsymbol{x}_{t-\tau}^{1:D} = \boldsymbol{x}_t^{1:D} + \sum_i P_i(\tilde{\boldsymbol{x}}_i^{1:D} - \boldsymbol{x}_t^{1:D})$ where $P_i$ is a Poisson random variable with mean $\tau \hat{R}_t^{\theta\,1:D}(\boldsymbol{x}_t^{1:D}, \tilde{\boldsymbol{x}}_i^{1:D})$. Note the sum assumes a mapping from $\mathcal{X}$ to $\mathbb{Z}$.

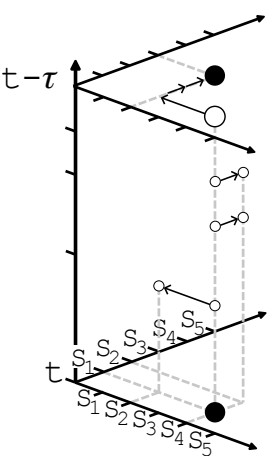

Figure 2: 3D visualization of one tau-leaping step from $x_t^{1:2} = \{S_4, S_1\}$ to $x_{t-\tau}^{1:2} = \{S_2, S_3\}$. Here, $D = 2$, $|\mathcal{X}| = 5$, $P_{12} = 1$, $P_{22} = 2$, all other $P_{ds} = 0$.

Using our knowledge of $\hat{R}_t^{\theta\,1:D}$, we can further unpack this update. Namely, $\hat{R}_t^{\theta\,1:D}(\boldsymbol{x}_t^{1:D}, \tilde{\boldsymbol{x}}^{1:D})$ can only be non-zero when $\tilde{\boldsymbol{x}}^{1:D}$ has a different value to $\boldsymbol{x}_t^{1:D}$ in exactly one dimension (rates for multi-dimensional changes are zero). Explicitly summing over these options we get $\boldsymbol{x}_{t-\tau}^{1:D} = \boldsymbol{x}_t^{1:D} + \sum_{d=1}^D \sum_{s=1\setminus x_t^d}^S P_{ds}(s - x_t^d)\boldsymbol{e}^d$ where $\boldsymbol{e}^d$ is a one-hot vector with a 1 at dimension $d$ and $P_{ds}$ is a Poisson random variable with mean $\tau \hat{R}_t^{\theta\,1:D}(\boldsymbol{x}_t^{1:D}, \boldsymbol{x}_t^{1:D} + (s-x_t^d)\boldsymbol{e}^d)$. Since multiple $P_{ds}$ can be non-zero, we see that tau-leaping allows $\boldsymbol{x}_t^{1:D}$ to change in multiple dimensions in a single step. Figure 2 visualizes this idea. During the $[t-\tau, t]$ interval, one jump occurs in dimension 1 and two jumps occur in dimension 2. These are all applied simultaneously once we reach $t-\tau$. When our discrete data has ordinal structure (e.g. Section 6.2) our mapping to $\mathbb{Z}$ is not arbitrary and making multiple jumps within the same dimension ($\sum_{s=1\setminus x_t^d}^S P_{ds} > 1$) is meaningful. In the non-ordinal/categorical case (e.g. Section 6.3) the mapping to $\mathbb{Z}$ is arbitrary and so, although taking simultaneous jumps in different dimensions is meaningful, taking multiple jumps within the same dimension is not. For this type of data, we reject changes to $x_t^d$ for any $d$ for which $\sum_{s=1\setminus x_t^d}^S P_{ds} > 1$. In practice, the rejection rate is very small when $R_t^{1:D}$ is suitable for categorical data (e.g. uniform), see Appendix H.3. In Section 4.5, our error bound accounts for this low probability of rejection and also the low probability of an out of bounds jump that we observe in practice in the ordinal case.

The tau-leaping approximation improves with smaller $\tau$, recovering exact simulation in the limit as $\tau \to 0$. Exact simulation is similar to an autoregressive model in that only one dimension changes per step. Increasing $\tau$ and thus the average number of dimensions changing per step gives us a natural way to modulate the 'autoregressiveness' of the model and trade sample quality with compute (Figure 4 right). We refer to our method of using tau-leaping to simulate the reverse CTMC as $\tau$LDR (tau-leaping denoising reversal) which we formalize in Algorithm 1 in Appendix F.

We note that theoretically, one could approximate $\hat{R}_t^{\theta\,1:D}$ as constant in the interval $[t-\tau, t]$, and construct a transition probability matrix by solving the forward Kolmogorov equation with the matrix exponential $P_{t-\tau|t} \approx \exp(\tau \hat{R}_t^{\theta\,1:D})$. However, for the learned $\hat{R}_t^{\theta\,1:D} \in \mathbb{R}^{S^D \times S^D}$ matrix, it is intractable to compute this matrix exponential so we use tau-leaping for sampling instead.

## 4.4 Predictor-Corrector

During approximate reverse sampling, we aim for the marginal distribution of samples at time $t$ to be close to $q_t(x_t)$ (the marginal at time $t$ of the true CTMC). The continuous time framework allows us to exploit additional information to more accurately follow the reverse progression of marginals, $\{q_t(x_t)\}_{t \in [T,0]}$ and improve sample quality. Namely, after a tau-leaping 'predictor' step using rate $\hat{R}_t^\theta$, we can apply 'corrector' steps with rate $R_t^c$ which has $q_t(x_t)$ as its stationary distribution. The corrector steps bring the distribution of samples at time $t$ closer to the desired $q_t(x_t)$ marginal. $R_t^c$ is easy to calculate as stated below

**Proposition 4.** *For a forward CTMC with marginals $\{q_t(x_t)\}_{t \in [0,T]}$, forward rate, $R_t$, and corresponding reverse CTMC with rate $\hat{R}_t$, the rate $R_t^c = R_t + \hat{R}_t$ has $q_t(x_t)$ as its stationary distribution.*

In practice, we approximate $R_t^c$ by replacing $\hat{R}_t$ with $\hat{R}_t^\theta$. This is directly analogous to Predictor-Corrector samplers in continuous state spaces [4] that predict by integrating the reverse SDE and correct with score-based Markov chain Monte Carlo steps, see Appendix F.2 for further discussion.

### 4.5 Error Bound

Our continuous time framework also allows us to provide a novel theoretical bound on the error between the true data distribution and the sample distribution generated via tau-leaping (without predictor-corrector steps), in terms of the error in our approximation of the reverse rate and the mixing of the forward noising process.

We assume we have a time-homogeneous rate matrix $R_t$ on $\mathcal{X}$, from which we construct the factorized rate matrix $R_t^{1:D}$ on $\mathcal{X}^D$ by setting $R_t^d = R_t$ for each $d$. Note that by rescaling time by a factor of $\beta(t)$ we can transform our choice of rate from Section 4.1 to be time-homogeneous. We will denote $|R| = \sup_{t \in [0,T], x \in \mathcal{X}} |R_t(x, x)|$, and let $t_{\text{mix}}$ be the (1/4)-mixing time of the CTMC with rate $R_t$ (see [24, Chapter 4.5]).

**Theorem 1.** *For any $D \geq 1$ and distribution $p_{\text{data}}$ on $\mathcal{X}^D$, let $\{x_t\}_{t \in [0,T]}$ be a CTMC starting in $p_{\text{data}}$ with rate matrix $R_t^{1:D}$ as above. Suppose that $\hat{R}_t^{\theta\, 1:D}$ is an approximation to the reverse rate matrix and let $(y_k)_{k=0,1,\ldots,N}$ be a tau-leaping approximation to the reverse dynamics with maximum step size $\tau$. Suppose further that there is some constant $M > 0$ independent of $D$ such that*

$$\sum_{y \neq x} \left| \hat{R}_t^{1:D}(x, y) - \hat{R}_t^{\theta\, 1:D}(x, y) \right| \leq M \tag{2}$$

*for all $t \in [0, T]$. Then under the assumptions in Appendix B.5, there are constants $C_1, C_2 > 0$ depending on $\mathcal{X}$ and $R_t$ but not $D$ such that, if $\mathcal{L}(y_0)$ denotes the law of $y_0$, we have the total variation bound*

$$||\mathcal{L}(y_0) - p_{\text{data}}||_{\text{TV}} \leq 3MT + \left\{ \left( |R|SDC_1 \right)^2 + \tfrac{1}{2}C_2(M + C_1SD|R|) \right\} \tau T + 2 \exp \left\{ -\frac{T \log^2 2}{t_{\text{mix}} \log 4D} \right\}$$

The first term of the above bound captures the error introduced by our approximation of the reverse rate $\hat{R}_t^{1:D}$ with $\hat{R}_t^{\theta\, 1:D}$. The second term reflects the error introduced by the tau-leaping approximation, and is linear in both $T$ and $\tau$, showing that as we take our tau-leaping steps to be arbitrarily small, the error introduced by tau-leaping goes to zero. The final term describes the mixing of the forward chain, and captures the error introduced since $p_{\text{ref}}$ and $q_T$ are not exactly equal.

We choose to make the dependence of the bound on the dimension $D$ explicit, since we are specifically interested in applying tau-leaping to high dimensional problems where we make transitions in different dimensions simultaneously in a single time step. The bound grows at worst quadratically in the dimension, versus e.g. exponentially. The bound is therefore useful in showing us that we do not need to make $\tau$ impractically small in high dimensions. Other than gaining these intuitions, we do not expect the bound to be particularly tight in practice and further it would not be practical to compute because of the difficulty in finding $M$, $C_1$ and $C_2$.

The assumptions listed in Appendix B.5 hold approximately for tau-leaping in practice when we use spatially biased rates for ordinal data such that jump sizes are small or uniform rates for non-ordinal data such that the dimensional rejection rate is small. These assumptions could be weakened, however, Theorem 1 would become much more involved, obscuring the intuition and structure of the problem.

## 5 Related Work

The application of denoising models to discrete data was first described in [1] using a binomial diffusion process for a binary dataset. Each reverse kernel $p_{k|k+1}^\theta$ was directly parameterized without using a denoising model $p_{0|k}^\theta$. In [25] an approach for discrete categorical data was suggested using a uniform forward noising kernel, $q_{k+1|k}$, and a reverse kernel parameterized through a denoising model, though no experiments were performed with the approach. Experiments on text and segmentation maps were then performed with a similar model in [6]. Other forward kernels were introduced in [8] that are more appropriate for certain data types such as the spatially biased Gaussian kernel. [9, 13] apply the approach to discrete latent space modeling using uniform and absorbing state forward

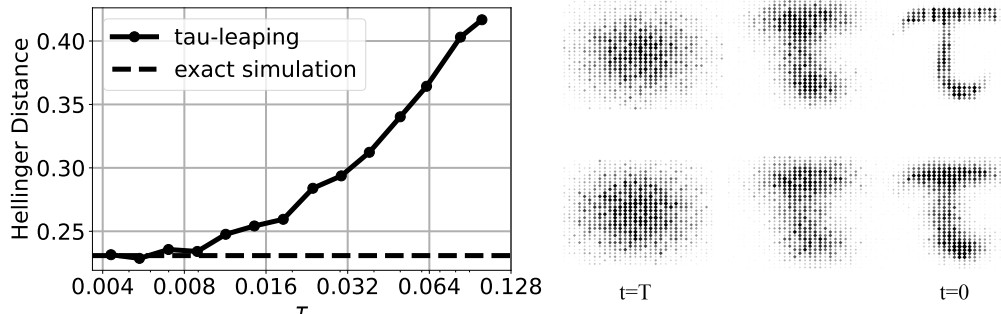

Figure 3: **Left:** Hellinger distance between the true training distribution and generated sample distributions with exact simulation or tau-leaping. With $\tau$ small, we simulate the reverse CTMC with the same fidelity as the exact simulation. **Top Right:** Histograms of the marginals during the reverse generative process simulated using tau-leaping with $\tau = 0.004$. Darker and larger diamonds represent increased density. **Bottom Right:** The same for $\tau = 0.1$, note the reduced sample quality.

kernels. Whilst a link to continuous time for the forward process is mentioned in [8], all of these approaches train and sample in discrete time. We show in Appendix G that this involves making an implicit approximation for multi-dimensional data. We extend this line of work by training and sampling in continuous time.

Other works also operate in discrete space but less rigidly follow the diffusion framework. A corruption process tailored to text is proposed in [12], whereby token deletion and insertion is also incorporated. [26] also focus on text, creating a generative reverse chain that repeatedly applies the same denoising kernel. The corruption distribution is also defined through the same denoising kernel to reduce distribution shift between training and sampling. In [7], a more standard masking based forward process is used but the reversal is interpreted from an order agnostic autoregressive perspective. They also describe how their model can be interpreted as the reversal of a continuous time absorbing state diffusion but do not utilize this perspective in training or sampling. [27] propose a denoising type framework that can be used on binary data where the forward and reverse process share the same transition kernel. Finally, in [11], the discrete latent space of a VQVAE is modeled by quantizing an underlying continuous state space diffusion with probabilistic quantization functions.

## 6 Experiments

### 6.1 Demonstrative Example

We first verify the method can accurately produce samples from the entire support of the data distribution and that tau-leaping can accurately simulate the reverse CTMC. To do this, we create a dataset formed of 2d samples of a state space of 32 arranged such that the histogram of the training dataset forms a '$\tau$' shape. We train a denoising model using the $\mathcal{L}_{\mathrm{CT}}$ objective with $p^{\theta}_{0|t}$ parameterized through a residual MLP (full details in Appendix H.1). We then sample the parameterized reverse process using an exact method (up to needing to numerically integrate the reverse rate) and tau-leaping. Figure 3 top-right shows the marginals during reverse simulation with $\tau = 0.004$ and we indeed produce samples from the entire support of $p_{\mathrm{data}}$. Furthermore, we find that with sufficiently small $\tau$, we can match the fidelity of exact simulation of the reverse CTMC (Figure 3 left). The value of $\tau$ dictates the number of network evaluations in the reverse process according to NFE $= T/\tau$. In all experiments we use $T = 1$. Exact simulation results in a non zero Hellinger distance between the generated and training distributions because of imperfections in the learned $\hat{R}^{\theta}_t$ model.

### 6.2 Image Modeling

We now demonstrate that our continuous time framework gives us improved generative modeling performance versus operating in discrete time. We show this on the CIFAR-10 image dataset. Images are typically stored as discrete data, each pixel channel taking one value from 256 possibilities. Continuous state space methods have to somehow get around this fact by, for example, adding a discretization function at the end of the generative process [3] or adding uniform noise to the data.

Table 1: Sample quality metrics and model likelihoods for diffusion methods modeling CIFAR10 in discrete state space. Diffusion methods modeling CIFAR10 in continuous space are included for reference. The Inception Score (IS) and Fréchet Inception Distance (FID) are calculated using 50000 generated samples with respect to the training dataset as is standard practice. The ELBO values are reported on the test set in bits per dimension.

|  | Method | IS ($\uparrow$) | FID ($\downarrow$) | ELBO ($\uparrow$) |
|---|---|---|---|---|
| Discrete state | D3PM Absorbing [8] | 6.78 | 30.97 | $-4.40$ |
|  | D3PM Gauss [8] | 8.56 | 7.34 | $-\mathbf{3.44}$ |
|  | $\tau$LDR-0 (ours) | 8.74 | 8.10 | $-3.59$ |
|  | $\tau$LDR-10 (ours) | **9.49** | **3.74** | $-3.59$ |
| Continuous state | DDPM [3] | 9.46 | 3.17 | $-3.75$ |
|  | NCSN [4] | 9.89 | 2.20 | - |

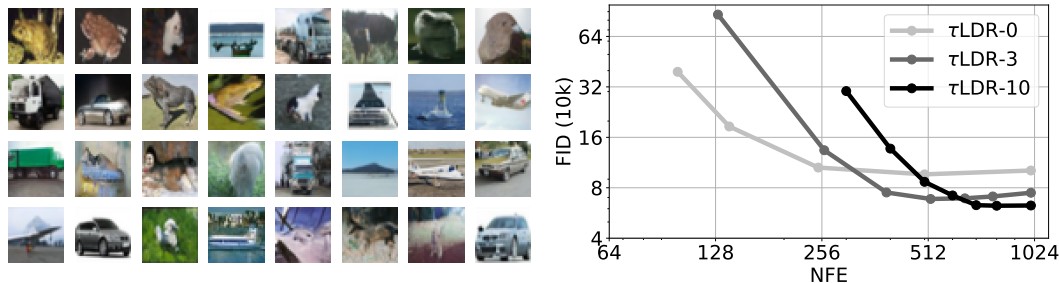

Figure 4: **Left:** Unconditional CIFAR10 samples from our $\tau$LDR-10 model **Right:** FID scores for the generated CIFAR10 samples versus number of $p_{0|t}^{\theta}$ evaluations during sampling (variation induced by varying $\tau$). Calculated with 10k samples, hence the discrepancy with Table 1 [28].

Here, we model the images directly in discrete space. We parameterize $p_{0|t}^{\theta}$ using the standard U-net architecture [3] with the modifications for discrete state space suggested by [8]. We use a spatially biased rate matrix and train with an augmented $\mathcal{L}_{CT}$ loss including direct $p_{0|t}^{\theta}$ supervision, full experimental details are in Appendix H.2.

Figure 4 left shows randomly generated unconditional CIFAR10 samples from the model and we report sample quality metrics in Table 1. We see that our method ($\tau$LDR-0) with 0 corrector steps has better Inception Score but worse FID than the D3PM discrete time method. However, our $\tau$LDR-10 method with 10 corrector steps per predictor step at the end of the reverse sampling process ($t < 0.1T$) greatly improves sample quality, beating the discrete time method in both metrics and further closes the performance gap with methods modeling images as continuous data. The derivation of the corrector rate which gave us this improved performance required our continuous time framework. D3PM achieves the highest ELBO but we note that this does not correlate well with sample quality. In Table 1, $\tau$ was adjusted such that both $\tau$LDR-0 and $\tau$LDR-10 used 1000 $p_{0|t}^{\theta}$ evaluations in the reverse sampling procedure. We show how FID score varies with number of $p_{0|t}^{\theta}$ evaluations for $\tau$LDR-$\{0, 3, 10\}$ in Figure 4 right. The optimum number of corrector steps depends on the sampling budget, with lower numbers of corrector steps being optimal for tighter budgets. This is due to the increased $\tau$ required to maintain a fixed budget when we use a larger number of corrector steps.

## 6.3 Monophonic Music

In this experiment, we demonstrate our continuous time model improves generation quality on non-ordinal/categorical discrete data. We model songs from the Lakh pianoroll dataset [29, 30]. We select all monophonic sequences from the dataset such that at each of the 256 time steps either one from 128 notes is played or it is a rest. Therefore, our data has state space size $S = 129$ and dimension $D = 256$. We scramble the ordering of the state space when mapping to $\mathbb{Z}$ to destroy any ordinal structure. We parameterize $p_{0|t}^{\theta}$ with a transformer architecture [31] and train using a conditional form of $\mathcal{L}_{CT}$ targeting the conditional distribution of the final 14 bars (224 time steps) given the first 2 bars of the song. We use a uniform forward rate matrix, $R_t$, full experimental details

Table 2: Metrics comparing generated conditional samples and ground truth completions. We compute these over the test set showing mean±std with respect to 5 samples for each test song.

| Model | Hellinger Distance | Proportion of Outliers |
|---|---|---|
| $\tau$LDR-0 Birth/Death | $0.3928 \pm 0.0010$ | $0.1316 \pm 0.0012$ |
| $\tau$LDR-0 Uniform | $0.3765 \pm 0.0013$ | $0.1106 \pm 0.0010$ |
| $\tau$LDR-2 Uniform | $\mathbf{0.3762 \pm 0.0015}$ | $\mathbf{0.1091 \pm 0.0014}$ |
| D3PM Uniform [8] | $0.3839 \pm 0.0002$ | $0.1137 \pm 0.0010$ |

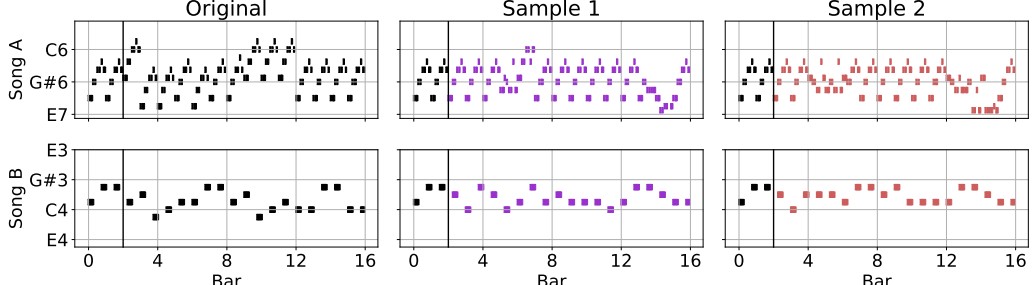

Figure 5: Conditional completions of an unseen music sequence. The conditioning 2 bars are shown to the left of the black line. More examples and audio recordings are linked in Appendix H.3.

are given in Appendix H.3. Conditional completions of unseen test songs are shown in Figure 5. The model is able to faithfully complete the piece in the same style as the conditioning bars.

We quantify sample quality in Table 2. We use two metrics: the Hellinger distance between the histograms of generated and ground truth notes and the proportion of outlier notes in the generations but not in the ground truth. Using our method, we compare between a birth/death and uniform forward rate matrix $R_t$. The birth/death rate is only non-zero for adjacent states whereas the uniform rate allows transitions between arbitrary states which is more appropriate for the categorical case thus giving improved sample quality. Adding 2 corrector steps per predictor step further improves sample quality. We also compare to the discrete time method D3PM [8] with its most suitable corruption process for categorical data. We find it performs worse than our continuous time method.

## 7 Discussion

We have presented a continuous time framework for discrete denoising models. We showed how to efficiently sample the generative process with tau-leaping and provided a bound on the error of the generated samples. On discrete data problems, we found our predictor-corrector sampler improved sample quality versus discrete time methods. Regarding limitations, our model requires many model evaluations to produce a sample. Our work has opened the door to applying the work improving sampling speed on continuous data [14, 15, 16, 17, 32] to discrete data problems too. Modeling performance on images is also slightly behind continuous state space models, we hope this gap is further closed with bespoke discrete state architectures and corruption process tuning. Finally, we note that the ELBO values for the discrete time model on CIFAR10 are better than for our method. In this work, we focused on sample quality rather than using our model to give data likelihoods e.g. for compression downstream tasks.

## Acknowledgements

Andrew Campbell and Joe Benton acknowledge support from the EPSRC CDT in Modern Statistics and Statistical Machine Learning (EP/S023151/1). Arnaud Doucet is partly supported by the EPSRC grant EP/R034710/1. He also acknowledges support of the UK Defence Science and Technology Laboratory (DSTL) and EPSRC under grant EP/R013616/1. This is part of the collaboration between US DOD, UK MOD and UK EPSRC under the Multidisciplinary University Research Initiative. This project made use of time on Tier 2 HPC facility JADE2, funded by EPSRC (EP/T022205/1).

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
