# Appendix

## Contents

The appendix is organized as follows. In Section A, we provide a short introduction to Continuous Time Markov Chains, including the relevant results we use in this work. Proofs for all the Propositions and Theorems from the main text are in Section B. We then describe in Section C some additional intuitions and forms of our proposed objective, $\mathcal{L}_{\mathrm{CT}}$. In Section D, we describe how an additional direct denoising model supervision term can be added to the objective to improve empirical performance. Details for how we define the forward process in our model can be found in Section E. Section F describes in more detail how CTMCs can be simulated and includes the algorithmic description of tau-leaping. We argue in Section G that operating in discrete time forces an implicit assumption when using a factorized forward process on multi-dimensional data. Full experimental details for all investigations can be found in Section H as well as additional plots and results from our models. Finally, in Section I, we consider the social impacts of our research.

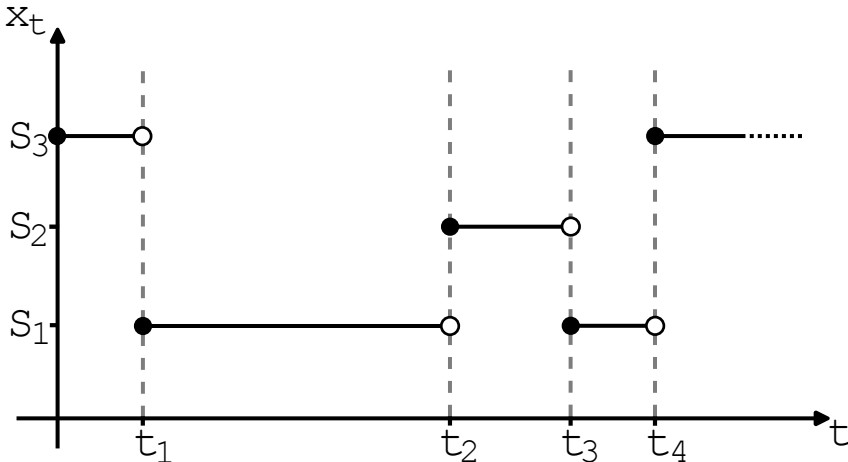

Figure 6: Schematic representation of a 1-dimensional CTMC with 3 states.

## A   Primer on Continuous Time Markov Chains

A Continuous Time Markov Chain (CTMC) is a right continuous stochastic process $\{x_t\}_{t \in [0,T]}$ satisfying the Markov property, with $x_t$ taking values in a discrete state space $\mathcal{X}$. Since the CTMC is Markov, future behaviour of the process depends only on the current state and not the history. A schematic representation of a CTMC path is shown in Figure 6. The process repeatedly transitions from one state to another after having waited in the previous state for a randomly determined amount of time.

A CTMC can be completely characterised by its jumps and holding times. Specifically, the time between each jump or *holding time* is exponentially distributed with mean $\nu(x)$ where $x$ is the state in which the process is holding. The next state that is jumped to is drawn from a jump probability distribution $r(\tilde{x}|x)$. The holding and jumping procedure is then repeated.

There is an equivalent definition involving the transition rate matrix, $R \in \mathbb{R}^{S \times S}$, that we use in the main paper. The transition rate matrix is defined as

$$R(\tilde{x}, x) = \lim_{\Delta t \to 0} \frac{q_{t|t-\Delta t}(x|\tilde{x}) - \delta_{x,\tilde{x}}}{\Delta t} \tag{3}$$

where $R(\tilde{x}, x)$ is the $(\tilde{x}, x)$ element of the transition rate matrix and $q_{t|t-\Delta t}(x|\tilde{x})$ is the infinitesimal transition probability of being in state $x$ at time $t$ given that the process was in state $\tilde{x}$ at time $t - \Delta t$. Conversely, the CTMC can itself be defined through this infinitesimal transition probability

$$q_{t|t-\Delta t}(x|\tilde{x}) = \delta_{x,\tilde{x}} + R(\tilde{x}, x)\Delta t + o(\Delta t) \tag{4}$$

where $o(\Delta t)$ represents terms that tend to zero at a faster rate than $\Delta t$. From this definition of the transition rate matrix, we can infer the following properties:

$$R(\tilde{x}, x) \geq 0 \quad \text{for} \quad \tilde{x} \neq x, \qquad R(x, x) \leq 0, \qquad R(x, x) = -\sum_{x' \neq x} R(x, x') \tag{5}$$

$R(\tilde{x}, x)$ is the rate at which probability mass moves from state $\tilde{x}$ to $x$. $R(x, x)$ is the total rate at which probability mass moves out of state $x$ and is thus negative.

In the time-homogeneous case, $R$ has simple relations to the jump and holding time definitions.

$$\nu(x) = -\frac{1}{R(x, x)} \qquad\qquad r(\tilde{x}|x) = (1 - \delta_{\tilde{x},x})\frac{R(x, \tilde{x})}{-R(x, x)}$$

In the time-inhomogeneous case, our transition rate matrix will now depend on time, $R_t$, and these simple relations to the jump and holding time definition do not hold. However, $R_t$ will still follow equations (3), (4) and (5).

The CTMC transition probabilities satisfy the Kolmogorov forward and backward equations. For $t > s$,

$$\text{Kolmogorov forward equation} \quad \partial_t q_{t|s}(x|\tilde{x}) = \sum_y q_{t|s}(y|\tilde{x}) R_t(y, x)$$

$$\text{Kolmogorov backward equation} \quad \partial_s q_{t|s}(x|\tilde{x}) = -\sum_y R_s(\tilde{x}, y) q_{t|s}(x|y)$$

The Kolmogorov forward equation also gives us a differential equation for the marginals of the CTMC.

$$\partial_t q_t(x) = \sum_y q_t(y) R_t(y, x).$$

**Exponential and Poisson Random Variables** In the time homogeneous case, holding times are exponentially distributed with mean $\nu(x) = -1/R(x, x)$. The tau-leaping algorithm relies on the fact that the number of events in interval $[0, t]$ is Poisson distributed with mean $\frac{1}{\nu} t$ when the inter-event times are exponentially distributed with mean $\nu$.

## B Proofs

### B.1 Proof of Proposition 1

*Proof.* We recall that a process $\{x_t\}_{t \in [0,T]}$ taking values in $\mathcal{X}$ is called a CTMC if it is right-continuous and satisfies the Markov property. Denote $\{y_t\}_{t \in [0,T]} = \{x_{T-t}\}_{t \in [0,T]}$ except at the jump times of the forward process $\tau_n$ with $n \in \mathbb{N}$, where $y_{T - \tau_n} = x_{\tau_n}^- = \lim_{t \le \tau_n, t \to \tau_n} x_t$. Hence, $\{y_t\}_{t \in [0,T]}$ is almost surely equal to $\{x_{T-t}\}_{t \in [0,T]}$ and is right-continuous. Since the Markov property is symmetric, we get that $\{y_t\}_{t \in [0,T]}$ is a CTMC. We now compute its transition matrix. Let $x, \tilde{x} \in \mathcal{X}$ with $x \ne \tilde{x}$, using the Kolmogorov forward equation, we have

$$\partial_t p_{t|s}(\tilde{x}|x) = \sum_{y \in \mathcal{X}} p_{t|s}(y|x) \hat{R}_{T-t}(y, \tilde{x}) ,$$

where $\{p_{t|s}, \ s, t \in [0, T], \ t > s\}$ is the transition probability system associated with $\{y_t\}_{t \in [0,T]}$ and $\{\hat{R}_{T-t}\}_{t \in [0,T]}$ is the transition rate matrix associated with $\{y_t\}_{t \in [0,T]}$. Note that

$$
\begin{aligned}
p_{t|s}(x = j|\tilde{x} = i) &= \mathbb{P}(y_t = j \mid y_s = i) \\
&= \mathbb{P}(x_{T-t} = j \mid x_{T-s} = i) \\
&= \mathbb{P}(x_{T-s} = i | x_{T-t} = j) \frac{\mathbb{P}(x_{T-t} = j)}{\mathbb{P}(x_{T-s} = i)} \\
&= q_{T-s|T-t}(\tilde{x} = i | x = j) \frac{q_{T-t}(x = j)}{q_{T-s}(\tilde{x} = i)}
\end{aligned}
$$

where $\{q_{t|s}, s, t \in [0, T], t > s\}$ is the transition probability system associated with $\{x_t\}_{t \in [0,T]}$ and $\{q_t, t \in [0, T]\}$ are the marginals of $\{x_t\}_{t \in [0,T]}$. Now, writing the backward Kolmogorov equation for $\{x_t\}_{t \in [0,T]}$

$$\partial_s q_{t|s}(\tilde{x}|x) = -\sum_{y \in \mathcal{X}} R_s(x, y) q_{t|s}(\tilde{x}|y)$$

Re-labeling the time indices we obtain,

$$
\begin{aligned}
\partial_{T-t} q_{T-s|T-t}(\tilde{x}|x) &= -\sum_{y \in \mathcal{X}} R_{T-t}(x, y) q_{T-s|T-t}(\tilde{x}|y) \\
\partial_t q_{T-s|T-t}(\tilde{x}|x) &= \sum_{y \in \mathcal{X}} R_{T-t}(x, y) q_{T-s|T-t}(\tilde{x}|y)
\end{aligned}
$$

Letting $s \to t$ and using that $\lim_{s \to t} q_{T-s|T-t}(x|\tilde{x}) = 0$, we get that

$$
\begin{aligned}
\hat{R}_{T-t}(x, \tilde{x}) &= \lim_{s \to t} \partial_t p_{t|s}(\tilde{x}|x) \\
&= \lim_{s \to t} \partial_t \left( q_{T-s|T-t}(x|\tilde{x}) \frac{q_{T-t}(\tilde{x})}{q_{T-s}(x)} \right) \\
&= \lim_{s \to t} \left[ \partial_t \left( q_{T-s|T-t}(x|\tilde{x}) \right) \frac{q_{T-t}(\tilde{x})}{q_{T-s}(x)} + q_{T-s|T-t}(x|\tilde{x}) \frac{\partial_t q_{T-t}(\tilde{x})}{q_{T-s}(x)} \right] \\
&= \lim_{s \to t} \partial_t \left( q_{T-s|T-t}(x|\tilde{x}) \right) \frac{q_{T-t}(\tilde{x})}{q_{T-s}(x)} \\
&= R_{T-t}(\tilde{x}, x) \frac{q_{T-t}(\tilde{x})}{q_{T-t}(x)}
\end{aligned}
$$

Re-labeling the time-indices on the rate matrices, we obtain

$$
\hat{R}_t(x, \tilde{x}) = R_t(\tilde{x}, x) \frac{q_t(\tilde{x})}{q_t(x)}
$$

Now we write the marginal ratio $\frac{q_t(\tilde{x})}{q_t(x)}$ in a different form

$$
\begin{aligned}
\frac{q_t(\tilde{x})}{q_t(x)} &= \sum_{x_0} \frac{p_{\text{data}}(x_0)}{q_t(x)} q_{t|0}(\tilde{x}|x_0) \\
&= \sum_{x_0} \frac{q_{0|t}(x_0|x)}{q_{t|0}(x|x_0)} q_{t|0}(\tilde{x}|x_0).
\end{aligned}
$$

Substituting in this form for the marginal ratio concludes the proof.

$\square$

## B.2 Proof of Proposition 2

In this section, we detail two proofs for Proposition 2. The first is a formal proof using results from stochastic processes. We then provide a second informal proof for the same result to gain intuition into the $\mathcal{L}_{\text{CT}}$ objective that only relies on elementary results from CTMCs.

### Proof 1 - Stochastic Processes

*Proof.* Let us write $\mathbb{Q}$ for the path measure of the forward CTMC with rate matrix $R_t$, $\hat{\mathbb{Q}}$ for the path measure of its exact time reversal and $\mathbb{P}^\theta$ for the path measure of the approximate reverse process with rate matrix $\hat{R}_t^\theta$. Also, we use superscripts to notate conditioning on the starting point, for example $\mathbb{Q}^{x_0}$ denotes the path measure of the forward process conditioned to start in $x_0$.

With this notation, we have

$$
\begin{aligned}
-\log p_0^\theta(x_0) &= -\log \int p_{\text{ref}}(\mathrm{d}x_T) \int_{\{\hat{W}_T = x_0\}} \mathbb{P}^{\theta, x_T}(\mathrm{d}w) \\
&= -\log \int q_{T|0}(\mathrm{d}x_T) \int_{\{\hat{W}_T = x_0\}} \hat{\mathbb{Q}}^{x_T}(\mathrm{d}w) \frac{\mathrm{d}p_{\text{ref}}}{\mathrm{d}q_{T|0}}(x_T) \frac{\mathrm{d}\mathbb{P}^{\theta, x_T}}{\mathrm{d}\hat{\mathbb{Q}}^{x_T}}(w) \\
&= -\log \int q_{T|0}(\mathrm{d}x_T) \int \hat{\mathbb{Q}}(\mathrm{d}w|\hat{W}_0 = x_T, \hat{W}_T = x_0) \frac{\mathrm{d}p_{\text{ref}}}{\mathrm{d}q_{T|0}}(x_T) \frac{\mathrm{d}\mathbb{P}^{\theta, x_T}}{\mathrm{d}\hat{\mathbb{Q}}^{x_T}}(w) \mathbb{Q}^{x_T}\{\hat{W}_0 = x_0\} \\
&\leq \int q_{T|0}(\mathrm{d}x_T) \int \hat{\mathbb{Q}}(\mathrm{d}w|\hat{W}_0 = x_T, \hat{W}_T = x_0) \left\{ -\log \frac{\mathrm{d}\mathbb{P}^{\theta, x_T}}{\mathrm{d}\hat{\mathbb{Q}}^{x_T}}(w) \right\} + C,
\end{aligned}
$$

where $\mathbb{P}^\theta, \hat{\mathbb{Q}}$ run in the reverse time direction. Writing $\hat{W}_s$ for a reverse path and integrating wrt $p_{\text{data}}(\mathrm{d}x_0)$ we have

$$\int p_{\text{data}}(x_0)[-\log p_0^\theta(x_0)] \le \int p_{\text{data}}(x_0) \int q_{T|0}(\mathrm{d}x_T) \int \hat{\mathbb{Q}}(\mathrm{d}\hat{W}|\hat{W}_0 = x_T, \hat{W}_T = x_0)$$

$$\times \left\{ \int_{s=0}^{T} \hat{R}_{T-s}^\theta(\hat{W}_s)\mathrm{d}s - \sum_{s:\hat{W}_{s-} \ne \hat{W}_s} \log \mathbb{P}_{T-s}^\theta(\hat{W}_s|\hat{W}_{s-})R_{T-s}^\theta(\hat{W}_{s-}) \right\} + C,$$

where $\hat{R}_t^\theta(x)$ is shorthand for $-\hat{R}_t^\theta(x,x)$.

When $x_0 \sim p_{\text{data}}, x_T \sim q_{T|0}(\cdot|x_0), \hat{W} \sim \hat{\mathbb{Q}}(\mathrm{d}W|\hat{W}_0 = x_T, \hat{W}_T = x_0)$, the reverse path is distributed according to $p_{\text{data}}(\mathrm{d}x_0)\mathbb{Q}_{x_0}(\mathrm{d}W)$ and therefore $(\hat{W}_{s-}, \hat{W}_s)$ is distributed like $(W_{T-s}, W_{(T-s)-})$ and thus we have

$$\int p_{\text{data}}(x_0)[-\log p_0^\theta(x_0)]$$

$$\le \int p_{\text{data}}(x_0)\mathbb{Q}_{x_0}(\mathrm{d}W) \left\{ \int_{s=0}^{T} \hat{R}_{T-s}^\theta(W_{(T-s)-})\mathrm{d}s - \sum_{s:W_{(T-s)-} \ne W_{T-s}} \log \mathbb{P}_{T-s}^\theta(W_{(T-s)-}|W_{T-s})\hat{R}_{T-s}^\theta(W_{T-s}) \right\} + C$$

Using Dynkin's lemma and the fact that $\mathbb{P}_t^\theta(x|y)\hat{R}_t^\theta(y) = \hat{R}_t^\theta(y,x)$ we can re-express this final line as

$$= \iint_{s=0}^{T} q_{T-s}(\mathrm{d}x) \left\{ \sum_{z \ne x} \hat{R}_{T-s}^\theta(x,z) - \sum_{z \ne x} R_{T-s}(x,z) \frac{\sum_{y \ne x} R_{T-s}(x,y)}{\sum_{z \ne x} R_{T-s}(x,z)} \log \hat{R}_{T-s}^\theta(y,x) \right\}$$

$$= \iint_{s=0}^{T} q_{T-s}(\mathrm{d}x)r_{T-s}(\mathrm{d}y|x) \left\{ \sum_{z \ne x} \hat{R}_{T-s}^\theta(x,z) - \sum_{z \ne x} R_{T-s}(x,z) \log \hat{R}_{T-s}^\theta(y,x) \right\}$$

$$= \iint_{s=0}^{T} q_s(\mathrm{d}x)r_s(\mathrm{d}y|x) \left\{ \sum_{z \ne x} \hat{R}_s^\theta(x,z) - \sum_{z \ne x} R_s(x,z) \log \hat{R}_s^\theta(y,x) \right\}$$

which rearranges to give the continuous time ELBO in the form of Proposition 2.

$\square$

**Proof 2 - Limit of Discrete Time ELBO**

*Proof.* Consider a partitioning of $[0, T]$, $0 = t_0 < t_1 < \cdots < t_{k-1} < t_k < t_{k+1} < \cdots < t_{K-1} < t_K = T$. Let $t_k - t_{k-1} = \Delta t$ for all $k$. In subscripts we use $k$ as a shorthand for $t_k$ when this does not cause confusion. Considering a CTMC with this time partitioning converts the problem into a discrete time Markov Chain with forward transition kernel, $q_{k+1|k}(x_{k+1}|x_k)$ and parameterized reverse kernel, $p_{k|k+1}^\theta(x_k|x_{k+1})$. Therefore, we can write the negative ELBO in its discrete time form, $\mathcal{L}_{\text{DT}}$

$$\mathcal{L}_{\text{DT}}(\theta) = \mathbb{E}_{p_{\text{data}}(x_0)} \left[ \text{KL}(q_{K|0}(x_K|x_0)||p_{\text{ref}}(x_K)) - \mathbb{E}_{q_{1|0}(x_1|x_0)} \left[ \log p_{0|1}^\theta(x_0|x_1) \right] \right.$$

$$\left. + \sum_{k=1}^{K-1} \mathbb{E}_{q_{k+1|0}(x_{k+1}|x_0)} \left[ \text{KL}(q_{k|k+1,0}(x_k|x_{k+1}, x_0)||p_{k|k+1}^\theta(x_k|x_{k+1})) \right] \right]$$

In the following, we will write the transition kernels in terms of the CTMC rate matrices and take the limit as $\Delta t \to 0$ to obtain a continuous time negative ELBO.

First, consider one item from the inner sum of $\mathcal{L}_{\text{DT}}$

$$L_k = \mathbb{E}_{p_{\text{data}}(x_0)q_{k+1|0}(x_{k+1}|x_0)} \left[ \text{KL}(q_{k|k+1,0}(x_k|x_{k+1}, x_0)||p_{k|k+1}^\theta(x_k|x_{k+1})) \right]$$

$$= -\mathbb{E}_{p_{\text{data}}(x_0)q_{k+1|0}(x_{k+1}|x_0)q_{k|k+1,0}(x_k|x_{k+1}, x_0)} \left[ \log p_{k|k+1}^\theta(x_k|x_{k+1}) \right] + C$$

$$= -\mathbb{E}_{q_k(x_k)q_{k+1|k}(x_{k+1}|x_k)} \left[ \log p_{k|k+1}^\theta(x_k|x_{k+1}) \right] + C$$

where we have absorbed terms that do not depend on $\theta$ into $C$. We now write $p^\theta_{k|k+1}(x_k|x_{k+1})$ in terms of $\hat{R}^\theta_k$.

$$p^\theta_{k|k+1}(x_k|x_{k+1}) = \delta_{x_k,x_{k+1}} + \hat{R}^\theta_k(x_{k+1},x_k)\Delta t + o(\Delta t)$$

$$
\begin{aligned}
\log p^\theta_{k|k+1}(x_k|x_{k+1}) &= \log\left(\delta_{x_k,x_{k+1}} + \hat{R}^\theta_k(x_{k+1},x_k)\Delta t + o(\Delta t)\right)\\
&= \delta_{x_k,x_{k+1}}\log\left(1 + \hat{R}^\theta_k(x_k,x_k)\Delta t + o(\Delta t)\right)\\
&\quad + (1-\delta_{x_k,x_{k+1}})\log\left(\hat{R}^\theta_k(x_{k+1},x_k)\Delta t + o(\Delta t)\right)\\
&= \delta_{x_k,x_{k+1}}\left(\hat{R}^\theta_k(x_k,x_k)\Delta t + o(\Delta t)\right)\\
&\quad + (1-\delta_{x_k,x_{k+1}})\log\left(\hat{R}^\theta_k(x_{k+1},x_k)\Delta t + o(\Delta t)\right) \quad (6)
\end{aligned}
$$

where on the last line we have used the series expansion for $\log(1+z) = z - \frac{z^2}{2} + o(z^2)$ valid for $|z| \le 1, z \ne -1$. For any finite $R^\theta_k(x_k,x_k)$, $\Delta t$ can be taken small enough such that the series expansion holds. We now substitute this form for $\log p^\theta_{k|k+1}$ into $L_k$ and further write the expectation over $q_{k+1|k}(x_{k+1}|x_k) = \delta_{x_k,x_{k+1}} + R_k(x_k,x_{k+1})\Delta t + o(\Delta t)$ as an explicit sum.

$$
\begin{aligned}
L_k = -\mathbb{E}_{q_k(x_k)}\Bigg[\sum_{x_{k+1}}\Bigg\{&\left[\delta_{x_k,x_{k+1}} + R_k(x_k,x_{k+1})\Delta t + o(\Delta t)\right]\times\\
&\left[\delta_{x_k,x_{k+1}}\left(\hat{R}^\theta_k(x_k,x_k)\Delta t + o(\Delta t)\right)\right.\\
&\left.+ \left(1-\delta_{x_k,x_{k+1}}\right)\log\left(\hat{R}^\theta_k(x_{k+1},x_k)\Delta t + o(\Delta t)\right)\right]\Bigg\}\Bigg] + C
\end{aligned}
$$

$$
\begin{aligned}
L_k = -\mathbb{E}_{q_k(x_k)}\Bigg[\sum_{x_{k+1}}\Bigg\{&\delta_{x_k,x_{k+1}}\hat{R}^\theta_k(x_k,x_k)\Delta t\\
&+ (1-\delta_{x_k,x_{k+1}})R_k(x_k,x_{k+1})\Delta t\times\\
&\log\left(\hat{R}^\theta_k(x_{k+1},x_k)\Delta t + o(\Delta t)\right) + o(\Delta t)\Bigg\}\Bigg] + C
\end{aligned}
$$

We can isolate $\hat{R}^\theta_k$ within the $\log$ through the following re-arrangement

$$
\begin{aligned}
\Delta t\log&\left(\hat{R}^\theta_k(x_{k+1},x_k)\Delta t + o(\Delta t)\right)\\
&= \Delta t\log\Delta t + \Delta t\log\left(\hat{R}^\theta_k(x_{k+1},x_k) + o(1)\right)\\
&= \Delta t\log\Delta t + \Delta t\log\left(1 + o(1)\right) + \Delta t\log\left(\hat{R}^\theta_k(x_{k+1},x_k)\right)
\end{aligned}
$$

where the first two terms are independent of $\theta$ and tend to 0 as $\Delta t \to 0$. Note that we assume $\hat{R}^\theta_k(x_{k+1},x_k) > 0$ for $x_{k+1} \ne x_k$ pairs which have $R_k(x_k,x_{k+1}) > 0$. This assumption is valid because, for $x_{k+1} \ne x_k$, we have

$$\hat{R}^\theta_k(x_{k+1},x_k) = R_k(x_k,x_{k+1})\sum_{x_0}\frac{q_{k|0}(x_k|x_0)}{q_{k|0}(x_{k+1}|x_0)}p^\theta_{0|k}(x_0|x_{k+1})$$

and we assume $p^\theta_{0|k}(x_0|x_{k+1}) > 0$ which is valid when we parameterize $p^\theta_{0|k}$ with a softmax output. We assume an irreducible Markov chain, hence $q_{k|0} > 0$ for $t_k > 0$.

With this re-arrangement, and absorbing constant terms into $C$, we obtain

$$L_k = -\mathbb{E}_{q_k(x_k)}\left[\sum_{x_{k+1}}\left\{\delta_{x_k,x_{k+1}}\hat{R}_k^\theta(x_k,x_k)\Delta t\right.\right.$$

$$+ (1-\delta_{x_k,x_{k+1}})R_k(x_k,x_{k+1})\Delta t\log\left(\hat{R}_k^\theta(x_{k+1},x_k)\right)$$

$$\left.\left.+ o(\Delta t)\right\}\right] + C$$

$$L_k = -\mathbb{E}_{q_k(x_k)}\left[\hat{R}_k^\theta(x_k,x_k)\Delta t + \sum_{x_{k+1}\neq x_k}R_k(x_k,x_{k+1})\Delta t\log\hat{R}_k^\theta(x_{k+1},x_k) + o(\Delta t)\right]$$

The second term can be re-written so that it is more efficient to approximate with Monte Carlo. Currently the denoising model $p_{0|k}^\theta$ has to be evaluated for each term in the sum $\sum_{x_{k+1}\neq x_k}$ which would require multiple forward passes of the neural network. We can instead create a new probability distribution to sample from as follows. Define

$$r_k(x_{k+1}|x_k) = (1-\delta_{x_k,x_{k+1}})\frac{R_k(x_k,x_{k+1})}{\mathcal{Z}^k(x_k)}$$

where

$$\mathcal{Z}^k(x_k) = \sum_{x'_{k+1}\neq x_k}R_k(x_k,x'_{k+1})$$

So we now have

$$L_k = -\mathbb{E}_{q_k(x_k)r_k(x_{k+1}|x_k)}\left[\hat{R}_k^\theta(x_k,x_k)\Delta t + \mathcal{Z}^k(x_k)\Delta t\log\hat{R}_k^\theta(x_{k+1},x_k) + o(\Delta t)\right]$$

Examining the other terms in $\mathcal{L}_{\text{DT}}$ we have $\mathbb{E}_{p_{\text{data}}(x_0)}\left[\text{KL}(q_{K|0}(x_K|x_0)||p_{\text{ref}}(x_K))\right]$ which does not depend on $\theta$ and $\mathbb{E}_{q_{1|0}(x_1|x_0)}\left[\log p_{0|1}^\theta(x_0|x_1)\right]$ which we expand here

$$\mathbb{E}_{q_{1|0}(x_1|x_0)}\left[\log p_{0|1}^\theta(x_0|x_1)\right]$$

$$= \sum_{x_1}\left\{\delta_{x_1,x_0} + \Delta t R_1(x_0,x_1) + o(\Delta t)\right\}\log p_{0|1}^\theta(x_0|x_1)$$

$$= \log p_{0|1}^\theta(x_0|x_0) + \Delta t\sum_{x_1}R_1(x_0,x_1)\log p_{0|1}^\theta(x_0|x_1) + o(\Delta t)$$

$$= \Delta t\hat{R}_1^\theta(x_0,x_0) + \Delta t\sum_{x_1}R_1(x_0,x_1)\log p_{0|1}^\theta(x_0|x_1) + o(\Delta t)$$

where on the final line we have used eq 6. In summary,

$$\mathcal{L}_{\text{DT}} = \Delta t\mathbb{E}_{p_{\text{data}}(x_0)q_{1|0}(x_1|x_0)}\left[-\hat{R}_1^\theta(x_0,x_0) + \sum_{x_1}R_1(x_0,x_1)\log p_{0|1}^\theta(x_0|x_1)\right]$$

$$- \Delta t\sum_{k=1}^{K-1}\mathbb{E}_{q_k(x_k)r_k(x_{k+1}|x_k)}\left[\hat{R}_k^\theta(x_k,x_k) + \mathcal{Z}^k(x_k)\log\hat{R}_k^\theta(x_{k+1},x_k)\right]$$

$$+ o(\Delta t) + C$$

We now take the limit of $\mathcal{L}_{\text{DT}}$ as $\Delta t \to 0$ and $K \to \infty$.

$$\lim_{\Delta t\to 0}\mathcal{L}_{\text{DT}} = \mathcal{L}_{\text{CT}} = -\int_0^T\mathbb{E}_{q_t(x)r_t(\tilde{x}|x)}\left[\hat{R}_t^\theta(x,x) + \mathcal{Z}^t(x)\log\left(\hat{R}_t^\theta(\tilde{x},x)\right)\right]dt + C$$

We can estimate the integral with Monte Carlo if we consider it to be an expectation with respect to a uniform distribution over times $(0,T)$. We also write $\hat{R}_t^\theta(x,x)$ explicitly as the negative off diagonal row sum to obtain

$$\mathcal{L}_{\text{CT}}(\theta) = T\,\mathbb{E}_{t\sim\mathcal{U}(0,T)q_t(x)r_t(\tilde{x}|x)}\left[\left\{\sum_{x'\neq x}\hat{R}_t^\theta(x,x')\right\} - \mathcal{Z}^t(x)\log\left(\hat{R}_t^\theta(\tilde{x},x)\right)\right] + C.$$

$\square$

## B.3 Proof of Proposition 3

*Proof.* We assume $q_{t|s}(\boldsymbol{x}_t^{1:D}|\boldsymbol{x}_s^{1:D})$ factorizes as $\prod_{d=1}^{D} q_{t|s}(x_t^d|x_s^d)$ where $q_{t|s}(x_t^d|x_s^d)$, $d = 1, \ldots, D$ are the transition probabilities for independent singular dimensional CTMCs each with forward rate $R_t^d(\tilde{x}^d, x^d)$. In the following, we will drop time subscripts on $x$ arguments. To find the correspondence between $R_t^{1:D}$ and $R_t^d$, we use the Kolmogorov forward equation

$$\partial_t q_{t|s}(\boldsymbol{x}^{1:D}|\tilde{\boldsymbol{x}}^{1:D}) = \sum_{\boldsymbol{y}^{1:D}} q_{t|s}(\boldsymbol{y}^{1:D}|\tilde{\boldsymbol{x}}^{1:D}) R_t^{1:D}(\boldsymbol{y}^{1:D}, \boldsymbol{x}^{1:D})$$

Substitute in our factorized form for $q_{t|s}$ into the LHS

$$\partial_t q_{t|s}(\boldsymbol{x}^{1:D}|\tilde{\boldsymbol{x}}^{1:D}) = \partial_t \left\{ \prod_{d=1}^{D} q_{t|s}(x^d|\tilde{x}^d) \right\}$$

$$= \sum_{d=1}^{D} q_{t|s}(\boldsymbol{x}^{1:D\backslash d}|\tilde{\boldsymbol{x}}^{1:D\backslash d}) \partial_t q_{t|s}(x^d|\tilde{x}^d)$$

$$= \sum_{d=1}^{D} q_{t|s}(\boldsymbol{x}^{1:D\backslash d}|\tilde{\boldsymbol{x}}^{1:D\backslash d}) \sum_{y^d} q_{t|s}(y^d|\tilde{x}^d) R_t^d(y^d, x^d)$$

$$= \sum_{d=1}^{D} \sum_{\boldsymbol{y}^{1:D}} q_{t|s}(\boldsymbol{x}^{1:D\backslash d}|\tilde{\boldsymbol{x}}^{1:D\backslash d}) q_{t|s}(y^d|\tilde{x}^d) R_t^d(y^d, x^d) \delta_{\boldsymbol{x}^{1:D\backslash d}, \boldsymbol{y}^{1:D\backslash d}}$$

$$= \sum_{d=1}^{D} \sum_{\boldsymbol{y}^{1:D}} q_{t|s}(\boldsymbol{y}^{1:D\backslash d}|\tilde{\boldsymbol{x}}^{1:D\backslash d}) q_{t|s}(y^d|\tilde{x}^d) R_t^d(y^d, x^d) \delta_{\boldsymbol{x}^{1:D\backslash d}, \boldsymbol{y}^{1:D\backslash d}}$$

$$= \sum_{\boldsymbol{y}^{1:D}} q_{t|s}(\boldsymbol{y}^{1:D}|\tilde{\boldsymbol{x}}^{1:D}) \sum_{d=1}^{D} R_t^d(y^d, x^d) \delta_{\boldsymbol{x}^{1:D\backslash d}, \boldsymbol{y}^{1:D\backslash d}}$$

We therefore obtain

$$\sum_{\boldsymbol{y}^{1:D}} q_{t|s}(\boldsymbol{y}^{1:D}|\tilde{\boldsymbol{x}}^{1:D}) R_t^{1:D}(\boldsymbol{y}^{1:D}, \boldsymbol{x}^{1:D}) = \sum_{\boldsymbol{y}^{1:D}} q_{t|s}(\boldsymbol{y}^{1:D}|\tilde{\boldsymbol{x}}^{1:D}) \sum_{d=1}^{D} R_t^d(y^d, x^d) \delta_{\boldsymbol{x}^{1:D\backslash d}, \boldsymbol{y}^{1:D\backslash d}}$$

This must be true for all possible factorizable forward process transitions, $q_{t|s}$, including $q_{t|s}(\boldsymbol{y}^{1:D}|\tilde{\boldsymbol{x}}^{1:D}) = \delta_{\boldsymbol{y}^{1:D}, \tilde{\boldsymbol{x}}^{1:D}}$. This choice gives us our forward rate relation

$$R_t^{1:D}(\tilde{\boldsymbol{x}}^{1:D}, \boldsymbol{x}^{1:D}) = \sum_{d=1}^{D} R_t^d(\tilde{x}^d, x^d) \delta_{\boldsymbol{x}^{1:D\backslash d}, \tilde{\boldsymbol{x}}^{1:D\backslash d}}$$

Substituting this into our expression for the reverse rate from Proposition 1 we obtain

$$\hat{R}_t^{1:D}(\boldsymbol{x}^{1:D}, \tilde{\boldsymbol{x}}^{1:D}) = \sum_{\boldsymbol{x}_0^{1:D}} \sum_{d=1}^{D} R_t^d(\tilde{x}^d, x^d) \frac{q_t(\tilde{\boldsymbol{x}}^{1:D}|\boldsymbol{x}_0^{1:D})}{q_t(\boldsymbol{x}^{1:D}|\boldsymbol{x}_0^{1:D})} q_{0|t}(\boldsymbol{x}_0^{1:D}|\boldsymbol{x}^{1:D}) \delta_{\boldsymbol{x}^{1:D\backslash d}, \tilde{\boldsymbol{x}}^{1:D\backslash d}}$$

$$= \sum_{\boldsymbol{x}_0^{1:D}} \sum_{d=1}^{D} R_t^d(\tilde{x}^d, x^d) \frac{q_{t|0}(\tilde{x}^d|x_0^d)}{q_{t|0}(x^d|x_0^d)} q_{0|t}(\boldsymbol{x}_0^{1:D}|\boldsymbol{x}^{1:D}) \delta_{\boldsymbol{x}^{1:D\backslash d}, \tilde{\boldsymbol{x}}^{1:D\backslash d}}$$

$$= \sum_{d=1}^{D} R_t^d(\tilde{x}^d, x^d) \delta_{\boldsymbol{x}^{1:D\backslash d}, \tilde{\boldsymbol{x}}^{1:D\backslash d}} \sum_{x_0^d} q_{0|t}(x_0^d|\boldsymbol{x}^{1:D}) \frac{q_{t|0}(\tilde{x}^d|x_0^d)}{q_{t|0}(x^d|x_0^d)} \sum_{\boldsymbol{x}_0^{1:D\backslash d}} q_{0|t}(\boldsymbol{x}_0^{1:D\backslash d}|x_0^d, \boldsymbol{x}^{1:D})$$

$$= \sum_{d=1}^{D} R_t^d(\tilde{x}^d, x^d) \delta_{\boldsymbol{x}^{1:D\backslash d}, \tilde{\boldsymbol{x}}^{1:D\backslash d}} \sum_{x_0^d} q_{0|t}(x_0^d|\boldsymbol{x}^{1:D}) \frac{q_{t|0}(\tilde{x}^d|x_0^d)}{q_{t|0}(x^d|x_0^d)}$$

$\square$

## B.4  Proof of Proposition 4

*Proof.* By the Kolmogorov forward equation applied to the forwards process, we have

$$\partial_t q_t(x_t) = \sum_y R_t(y, x_t) q_t(y)$$

In addition, applying the Kolmogorov forward equation to the reverse process, which has the same marginals as the forward but time-reversed, we get

$$-\partial_t q_t(x_t) = \sum_y \hat{R}_t(y, x_t) q_t(y)$$

Summing these two equations gives

$$\sum_y \left\{ R_t(y, x_t) + \hat{R}_t(y, x_t) \right\} q_t(y) = 0$$

Therefore, by comparison with the Kolmogorov equation, $R_t + \hat{R}_t$ is the rate matrix of a CTMC with invariant distribution $q_t$. $\qquad\square$

## B.5  Proof of Theorem 1

In this section, we derive a bound on the error of our tau-leaping diffusion model. Because the tau-leaping approximation is only interesting in the case where multiple jumps are made along different dimensions in a single step, we choose to make the dependence of our bound on the dimension of our model explicit, rather than simply considering the case of fixed $D$ and $\tau \to 0$.

Recall from the main text that we have a time-homogeneous rate matrix $R_t$ on $\mathcal{X}$, from which we construct the factorised rate matrix $R_t^{1:D}$ on $\mathcal{X}^D$ by setting $R_t^d = R_t$ for each $d$, and will denote $|R| = \sup_{t \in [0,T], x \in \mathcal{X}} |R_t(x, x)|$, and let $t_{\text{mix}}$ be the $(1/4)$-mixing time of the CTMC with rate $R_t$. We also define addition on the state space $\mathcal{X}^D$ using a mapping from $\mathcal{X}$ to $\mathbb{Z}$ as in Section 4.3 and component-wise addition.

**Theorem 1.** *For any $D \geq 1$ and distribution $p_{\text{data}}$ on $\mathcal{X}^D$, let $\{x_t\}_{t \in [0,T]}$ be a CTMC starting in $p_{\text{data}}$ with rate matrix $R_t^{1:D}$ as above. Suppose that $\hat{R}_t^{\theta\, 1:D}$ is an approximation to the reverse rate matrix and let $(y_k)_{k=0,1,\ldots,N}$ be a tau-leaping approximation to the reverse dynamics with maximum step size $\tau$. Suppose further that there is some constant $M > 0$ independent of $D$ such that*

$$\sum_{y \neq x} \left| \hat{R}_t^{1:D}(x, y) - \hat{R}_t^{\theta\, 1:D}(x, y) \right| \leq M$$

*for all $t \in [0, T]$. Then under the assumptions listed below, there are constants $C_1, C_2 > 0$ depending on $\mathcal{X}$ and $R_t$ but not $D$ such that, if $\mathcal{L}(y_0)$ denotes the law of $y_0$, we have the total variation bound*

$$||\mathcal{L}(y_0) - p_{\text{data}}||_{\text{TV}} \leq 3MT + \left\{ \left( |R| SDC_1 \right)^2 + \tfrac{1}{2} C_2(M + C_1 SD|R|) \right\} \tau T + 2 \exp \left\{ -\frac{T \log^2 2}{t_{\text{mix}} \log 4D} \right\}$$

The above theorem holds under the following assumptions, where we write $x \sim y$ for $x, y \in S^D$ if they differ in at most one coordinate.

**Assumption 1.** *The data distribution $p_{\text{data}}$ is strictly positive.*

**Assumption 2.** *There exists a constant $C_1 > 0$, depending on $S$ and $R_t$ but not $D$, such that for all $t \in [0, T]$ and $x, y \in S^D$ such that $x \sim y$, we have*

$$\frac{q_t(x)}{q_t(y)} \leq C_1.$$

**Assumption 3.** *There exists a constant $C_2 > 0$, depending on $S$ and $R_t$ but not $D$, such that for all $t \in [0, T]$ and all $x, y \in S^D$ such that $x \sim y$, we have*

$$\sum_z \left| \hat{R}_t(x, x + z) - \hat{R}_t(y, y + z) \right| \leq C_2.$$

If instead we were to allow $C_1$ and $C_2$ to depend on the dimension $D$, then Assumptions 2 and 3 follow trivially from Assumption 1 and the finiteness of the state space. However, we choose the stronger formulation above in order to make explicit the dependence of the error bound on the dimension, as previously explained.

As remarked in the main text, in most cases of practical interest (including the two examples explored in Section 6), Assumption 3 holds only approximately. However, we still expect the bound in Assumption 3 to hold whenever $x, y$ are in addition chosen such that the tau-leaping approximation of the reverse process makes a jump between them with reasonably high probability. For example, in the case where our data is ordinal, we expect that for any $x \sim y$ jumps from $x$ to $y$ are only common when $x$ is close to $y$, and thus $\hat{R}_t(x, x+z)$ and $\hat{R}_t(y, y+z)$ should be reasonably close whenever a jump from $x$ to $y$ occurs. Under a weaker assumption of this form, the proof of Theorem 1 can be adapted to work along similar lines, at the cost of a significant increase in technicality. We therefore choose to focus on the simpler case where Assumption 3 holds as it illustrates the key ideas.

In order to prove Theorem 1 we will require the following lemmas.

**Proposition 5.** *Let $(x_t)_{t \in [0,T]}$ and $(y_t)_{t \in [0,T]}$ be continuous time Markov chains on a finite state space $S$ with generators $G_t$ and $H_t$ respectively which are both bounded and continuous in $t$. Let the Markov kernels associated to $X$ and $Y$ be $K$ and $L$ respectively. Then for any probability distribution $\nu$ on $S$ we have*

$$||\nu K - \nu L||_{\mathrm{TV}} \leq \int_0^T \sup_{x \in S} \left\{ \sum_{y \neq x} |G_t(x,y) - H_t(x,y)| \right\} \mathrm{d}t$$

*Proof.* We define a coupling of $(x_t)_{t \in [0,T]}$ and $(y_t)_{t \in [0,T]}$ as follows, based on the construction in Chapter 20.1 of [24]. First take $Z \sim \nu$ and set $x_0 = y_0 = Z$. Also define the variables $\tilde{x}_0 = \tilde{y}_0 = Z$.

Next, fix $\lambda$ such that $|G_t(x,x)|, |H_t(x,x)| \leq \lambda$ for all $x \in S, t \in [0,T]$, let $(N_s)_{1 \leq s \leq T}$ be a Poisson process on $[0,T]$ of rate $\lambda$, and set $N_0 = 0$. We write $N = N_T$, and $S_1, S_2, \ldots, S_N$ for the arrival times and set $S_{n+1} = T$. We construct $x_t$ and $y_t$ for $t > 0$ inductively as follows. For $t \in [0, S_1)$ let $x_t = y_t = x_0$. Let $1 \leq j \leq N$. Given $(x_r : r < S_j)$, $(y_r : r < S_j)$, and $\tilde{x}_j, \tilde{y}_j$, define the following probability measures

$$\rho_j(\tilde{x}_j, w) := \begin{cases} G_{S_j}(\tilde{x}_j, w)/\lambda, & w \neq \tilde{x}_j \\ 1 - G_{S_j}(\tilde{x}_j, w)/\lambda, & w = \tilde{x}_j, \end{cases}$$

$$\rho'_j(\tilde{y}_j, w) := \begin{cases} H_{S_j}(\tilde{y}_j, w)/\lambda, & w \neq \tilde{y}_j \\ 1 - H_{S_j}(\tilde{y}_j, w)/\lambda, & w = \tilde{y}_j. \end{cases}$$

Sample $(\tilde{x}_{j+1}, \tilde{y}_{j+1})$ from a maximal coupling of $(\rho_j, \rho'_j)$ and for $t \in [S_j, S_{j+1})$ set $x_t = \tilde{x}_{j+1}$, $y_t = \tilde{y}_{j+1}$. Finally set $x_T = x_{S_N}$ and $y_T = y_{S_N}$.

Now, observe that $(x_t, y_t)_{t \in [0,T]}$ defined in this way is a coupling of the given Markov chains. Moreover,

$$||\nu K - \nu L||_{\mathrm{TV}} \leq \mathbb{P}(x_T \neq y_T)$$

$$= \mathbb{E}\left[ \sum_{j=1}^{N} \mathbb{I}\{x_s = y_s, s < S_j\} \mathbb{I}\{x_{S_j} \neq y_{S_j}\} \right]$$

$$= \sum_{n=0}^{\infty} \frac{\lambda^n e^{-\lambda}}{n!} \sum_{j=0}^{n} \mathbb{E}\left[ \mathbb{I}\{x_s = y_s, s < S_j\} \mathbb{I}\{x_{S_j} \neq y_{S_j}\} \right]$$

and using the fact that jumps are coupled maximally

$$
= \sum_{n=0}^{\infty} \frac{\lambda^n e^{-\lambda}}{n!} \sum_{j=0}^{n} \mathbb{E}\Big[ \mathbb{I}\{x_s = y_s,\, s < S_j\} \times \|\rho_j(X_{S_{j-1}}, \cdot) - \tilde{\rho}_j(X_{S_{j-1}}, \cdot)\|_{\mathrm{TV}} \Big]
$$

$$
= \sum_{n=0}^{\infty} \frac{\lambda^n e^{-\lambda}}{n!} \sum_{j=0}^{n} \mathbb{E}\Big[ \mathbb{I}\{x_s = y_s,\, s < S_j\} \frac{1}{\lambda} \sum_{z} |G_{S_j}(x_{S_{j-1}}, z) - H_{S_j}(x_{S_{j-1}}, z)| \Big]
$$

$$
= \frac{1}{\lambda} \mathbb{E}\Big[ \sum_{s:x_s \neq x_{s-}} \sum_{z} |G_s(x_{s-}, z) - H_s(x_{s-}, z)| \Big]
$$

$$
= \frac{1}{\lambda} \int_{s=0}^{T} \mathbb{E}\Big[ \lambda \sum_{z} |G_s(x_{s-}, z) - H_s(x_{s-}, z)| \Big]
$$

$$
= \int_{s=0}^{T} \mathbb{E}\Big[ \sum_{z} |G_s(x_{s-}, z) - H_s(x_{s-}, z)| \Big] \mathrm{d}s
$$

as required. $\qquad\square$

**Proposition 6.** *For all $t \in [0, T]$ and $x, y \in \mathcal{X}^D$ such that $x \sim y$, we have*

$$
|\partial_t \hat{R}_t(x, y)| \leq 2|R|^2 SDC_1^2
$$

*Moreover, it follows that $\hat{R}_t$ is bounded and continuous in $t$.*

*Proof.* Omitting the superscripts for brevity where the notation is clear, we have

$$
\left| \partial_t \hat{R}_t^{1:D}(x^{1:D}, y^{1:D}) \right| = \left| R_t(y, x) \partial_t \left\{ \frac{q_t(y)}{q_t(x)} \right\} \right|
$$

$$
= \left| R_t(y, x) \left\{ \frac{q_t(y)}{q_t(x)} \frac{\sum_z R_t(z, y) q_t(z)}{q_t(y)} - \frac{q_t(y)}{q_t(x)} \frac{\sum_z R_t(z, x) q_t(z)}{q_t(x)} \right\} \right|
$$

$$
\leq 2|R|^2 SDC_1^2
$$

where the second line follows from Kolmogorov's forward equation and the final inequality follows from Assumption 2 plus the fact that $R_t(z, x)$ (resp. $R_t(z, y)$) is only non-zero when $x \sim z$ (resp. $y \sim z$), and there are at most $|S||D|$ values of $x$ (resp. $y$) for which this holds. $\qquad\square$

We now give the proof of Theorem 1.

*Proof of Theorem 1.* Let us label the time steps used in tau-leaping by $0 = t_0 < t_1 < \cdots < t_N = T$, denote $\tau_k = t_k - t_{k-1}$, and denote the target stationary distribution by $\pi^D(x^{1:D}) = \prod_{d=1}^{D} \pi(x^d)$, where $\pi$ is the invariant distribution of the single-dimensional transition matrix $R_t^1$.

Also, let $\mathcal{R}_k^{\theta,(\tau)}$ be the Markov kernel corresponding to applying the tau-leaping approximation with rate matrix $\hat{R}_{t_k}^{\theta}$ to move from $t_k$ to $t_{k-1}$, and denote $\mathcal{R}^{\theta,(\tau)} = \mathcal{R}_N^{\theta,(\tau)} \mathcal{R}_{N-1}^{\theta,(\tau)} \ldots \mathcal{R}_1^{\theta,(\tau)}$ so that $\mathcal{R}^{\theta,(\tau)}$ expresses the full dynamics of the tau-leaping process and we have $\mathcal{L}(\hat{y}_0) = \pi^D \mathcal{R}^{\theta,(\tau)}$.

Then, as in [19] we can decompose

$$
||\pi^D \mathcal{R}^{\theta,(\tau)} - p_d||_{\mathrm{TV}} \leq ||\pi^D \mathcal{R}^{\theta,(\tau)} - \pi^D(\mathbb{P}^R)_{T|0}||_{\mathrm{TV}} + ||\pi^D - q_T||_{\mathrm{TV}}
$$

where $\mathbb{P}^R$ is the path measure of the exact reverse process.

We deal with the second term first. Let $t_{\mathrm{mix}}$ be the $(1/4)$-mixing time of the single-dimension CTMC with rate matrix $R_t^1$, i.e.

$$
t_{\mathrm{mix}} = \inf \left\{ t \geq 0 : \sup_{x_0^1 \in S} ||q_{t|0}(\,\cdot\,|x_0^1) - \pi||_{\mathrm{TV}} \leq \frac{1}{4} \right\}
$$

It then follows from

$$||q_{t|0}(\,\cdot\,|x_0^{1:D}) - \pi^D||_{\text{TV}} \le \sum_{d=1}^{D} ||q_{t|0}(\,\cdot\,|x_0^d) - \pi||_{\text{TV}}$$

that $t_{\text{mix}}^D$, the (1/4)-mixing time of the full CTMC with rate matrix $R_t^{1:D}$, satisfies the inequality $t_{mix}^D \le \{1 + \lceil \log_2 D \rceil\} t_{\text{mix}}$. If we view $(x_{mt_{mix}^D})_{m \in \mathbb{N}}$ as a discrete-time Markov chain, then standard results on Markov chain mixing (see, for example, Chapter 4.5 of [24]) show that

$$||q_{mt_{mix}^D|0}(\,\cdot\,|x_0^{1:D}) - \pi^D||_{\text{TV}} \le 2^{-m}$$

It then follows that for any $T \ge 0$ we have

$$||\pi^D - q_T||_{\text{TV}} \le 2\exp\left\{-\frac{T\log 2}{t_{mix}^D}\right\} \le 2\exp\left\{-\frac{T\log^2 2}{t_{\text{mix}}\log 4D}\right\}$$

completing the bound on the second term.

To bound the first term, we define $\mathcal{P}_k = (\mathbb{P}^R)_{T-t_{k-1}|T-t_k}$ and decompose it as

$$||\pi\mathcal{R}^{\theta,(\tau)} - \pi(\mathbb{P}^R)_{T|0}||_{\text{TV}} \le \sup_\nu ||\nu\mathcal{R}_N^{\theta,(\tau)}\dots\mathcal{R}_1^{\theta,(\tau)} - \nu\mathcal{P}_N\dots\mathcal{P}_1||_{\text{TV}}$$

$$\le \sup_\nu ||\nu\mathcal{R}_N^{\theta,(\tau)}\mathcal{R}_{N-1}^{\theta,(\tau)}\dots\mathcal{R}_1^{\theta,(\tau)} - \nu\mathcal{R}_N^{\theta,(\tau)}\mathcal{P}_{N-1}\dots\mathcal{P}_1||_{\text{TV}}$$

$$+ \sup_\nu ||\nu\mathcal{R}_N^{\theta,(\tau)}\mathcal{P}_{N-1}\dots\mathcal{P}_1 - \nu\mathcal{P}_N\mathcal{P}_{N-1}\dots\mathcal{P}_1||_{\text{TV}}$$

$$\le \sup_\nu ||\nu\mathcal{R}_{N-1}^{\theta,(\tau)}\dots\mathcal{R}_1^{\theta,(\tau)} - \nu\mathcal{P}_{N-1}\dots\mathcal{P}_1||_{\text{TV}} + \sup_\nu ||\nu\mathcal{R}_N^{\theta,(\tau)} - \nu\mathcal{P}_N||_{\text{TV}}$$

$$\le \sum_{k=1}^{N} \sup_\nu ||\nu\mathcal{R}_k^{\theta,(\tau)} - \nu\mathcal{P}_k||_{\text{TV}}$$

by proceeding inductively. So it suffices to find bounds on the total variation distance accumulated on each interval $[t_{k-1}, t_k]$.

Let $\mathcal{R}_k^\theta$ be the Markov kernel corresponding to running the chain from $t_k$ to $t_{k-1}$ with constant rate matrix $\hat{R}_{t_k}^\theta$. Since by Proposition 6 the reverse rate matrix $\hat{R}_t$ is bounded and continuous in $t$, using Proposition 5 we made deduce that for any distribution $\nu$ on $S$ we have

$$||\nu\mathcal{P}_k - \nu\mathcal{R}_k^\theta||_{\text{TV}} \le \int_{t_{k-1}}^{t_k} \sup_{x\in S}\left\{\sum_{y\neq x}|\hat{R}_t(x,y) - \hat{R}_{t_k}^\theta(x,y)|\right\} dt$$

$$\le \int_{t_{k-1}}^{t_k} \sup_{x\in S}\left\{\sum_{y\neq x}|\hat{R}_t(x,y) - \hat{R}_{t_k}(x,y)|\right\} dt$$

$$+ \int_{t_{k-1}}^{t_k} \sup_{x\in S}\left\{\sum_{y\neq x}|\hat{R}_{t_k}(x,y) - \hat{R}_{t_k}^\theta(x,y)|\right\} dt$$

The first half of this expression can be bounded using the Mean Value Theorem, according to

$$\int_{t_{k-1}}^{t_k} \sup_{x\in S}\left\{\sum_{y\neq x}|\hat{R}_t(x,y) - \hat{R}_{t_k}(x,y)|\right\} dt \le \int_{t_{k-1}}^{t_k} |t - t_k| \cdot 2|R|^2 S^2 D^2 C_1^2 \, dt$$

$$\le \left(|R|SDC_1\tau_k\right)^2$$

where in the first line we have used that the summand is only non-zero when $y \sim x$, and there are at most $|S||D|$ values of $y$ for which this holds. The second term can be bounded using condition (2), to get

$$\int_{t_{k-1}}^{t_k} \sup_{x\in S}\left\{\sum_{y\neq x}|\hat{R}_{t_k}(x,y) - \hat{R}_{t_k}^\theta(x,y)|\right\} dt \le M\tau_k$$

Combining these two expressions, we get a bound on $||\nu \mathcal{P}_k - \nu \mathcal{R}_k^\theta||_{\text{TV}}$.

$$||\nu \mathcal{P}_k - \nu \mathcal{R}_k^\theta||_{\text{TV}} \leq \left(|R|SDC_1\tau_k\right)^2 + M\tau_k$$

It remains to bound $||\nu \mathcal{R}_k^\theta - \nu \mathcal{R}_k^{\theta,(\tau)}||_{\text{TV}}$. Note that performing tau-leaping with rate matrix $\hat{R}_{t_k}^\theta$ starting in $x_{t_k}$ is equivalent to running a continuous time Markov chain from time $t_k$ to $t_{k-1}$ with constant rate matrix $\hat{R}_{t_k}^{\theta,(\tau)}$ given by

$$\hat{R}_{t_k}^{\theta,(\tau)}(x,y) = \hat{R}_{t_k}^\theta(x_{t_k}, y - x + x_{t_k})$$

(followed potentially by a clamping operation to keep us within $\mathcal{X}^D$). By an analogous argument to the proof of Proposition 5,

$$||\delta_{x_{t_k}} \mathcal{R}_k^\theta - \delta_{x_{t_k}} \mathcal{R}_k^{\theta,(\tau)}||_{\text{TV}} \leq \int_{t_{k-1}}^{t_k} \mathbb{E}\Big[ \sum_{y \neq x_t} |\hat{R}_{t_k}^\theta(x_t, y) - \hat{R}_{t_k}^\theta(x_{t_k}, y - x_t + x_{t_k})| \Big] \, dt$$

where the expectation is taken over $(x_t)_{t \in [t_{k-1}, t_k]}$ distributed according to the exact CTMC with rate matrix $\hat{R}_{t_k}^\theta$. (Note we have disregarded the clamping operation, since this can only decrease the resulting total variation distance.)

We may rewrite this bound in terms of the exact reverse process using condition (2) to get

$$||\delta_{x_{t_k}} \mathcal{R}_k^\theta - \delta_{x_{t_k}} \mathcal{R}_k^{\theta,(\tau)}||_{\text{TV}} \leq \int_{t_{k-1}}^{t_k} \mathbb{E}\Big[ 2M + \sum_{y \neq x_t} |\hat{R}_{t_k}(x_t, y) - \hat{R}_{t_k}(x_{t_k}, y - x_t + x_{t_k})| \Big] \, dt$$

Let $J_t$ be the number of jumps that $(x_t)$ makes between $t_k$ and $t$, and label the times of these jumps as $s_1, \ldots, s_j$ where $t \leq s_1 \leq \cdots \leq s_j \leq t_k$ and $j = J_t$ for convenience. Then by Assumption 3, we have

$$\sum_{y \neq x_t} |\hat{R}_{t_k}(x_t, y) - \hat{R}_{t_k}(x_{t_k}, y - x_t + x_{t_k})| \leq \sum_z |\hat{R}_{t_k}(x_t, x_t + z) - \hat{R}_{t_k}(x_{s_1}, x_{s_1} + z)| + \ldots$$
$$+ \sum_z |\hat{R}_{t_k}(x_{s_j}, x_{s_j} + z) - \hat{R}_{t_k}(x_{t_k}, x_{t_k} + z)|$$
$$\leq C_2 J_t$$

where we have made the substitution $z = y - x_{t_k}$. We conclude that

$$||\delta_{x_{t_k}} \mathcal{R}_k^\theta - \delta_{x_{t_k}} \mathcal{R}_k^{\theta,(\tau)}||_{\text{TV}} \leq \int_{t_{k-1}}^{t_k} \mathbb{E}\left[2M + C_2 J_t\right] \, dt$$
$$\leq 2M|t_k - t_{k-1}| + C_2 \int_{t_{k-1}}^{t_k} |t_k - t| \cdot \sup_x |\hat{R}_{t_k}^\theta(x,x)| \, dt$$
$$\leq 2M\gamma_k + \frac{1}{2} C_2 |\hat{R}_{t_k}^\theta| \tau_k^2$$
$$\leq 2M\tau_k + \frac{1}{2} C_2 (M + C_1 SD|R|) \tau_k^2$$

where to bound $\mathbb{E}[J_t]$ we have observed that jumps of $(x_t)$ occur at a rate bounded above by $\sup_x |\hat{R}_{t_k}^\theta(x,x)|$, and in the last line we have used the condition (2) and Assumption 2. Since the above holds for any choice of $x_{t_k}$, it follows that

$$\sup_\nu ||\nu \mathcal{R}_k^\theta - \nu \mathcal{R}_k^{\theta,(\tau)}||_{\text{TV}} \leq 2M\tau_k + \frac{1}{2} C_2 (M + C_1 SD|R|) \tau_k^2$$

Summing over $k$ and putting all our bounds together, we get

$$||\mathcal{L}(y_0) - p_{\text{data}}||_{\text{TV}} \leq 3MT + \left\{ \left(|R|SDC_1\right)^2 + \frac{1}{2} C_2 (M + C_1 SD|R|) \right\} \tau T + 2 \exp\left\{ -\frac{T \log^2 2}{t_{\text{mix}} \log 4D} \right\}$$

as required.                                                                                                    $\square$

## C Continuous Time ELBO Details

### C.1 Comparison with the Discrete Time ELBO

It is easiest to gain intuition on the $\mathcal{L}_{\text{CT}}$ objective by comparing it to its discrete time counterpart, $\mathcal{L}_{\text{DT}}$, and examining the way in which $\mathcal{L}_{\text{DT}}$ in the limit becomes $\mathcal{L}_{\text{CT}}$ when we take the time step size to be very small. We repeat the definition of $\mathcal{L}_{\text{CT}}$ here for convenience

$$\mathcal{L}_{\text{CT}}(\theta) = T\, \mathbb{E}_{t \sim \mathcal{U}(0,T)q_t(x)r_t(\tilde{x}|x)} \left[ \left\{ \sum_{x' \neq x} \hat{R}_t^\theta(x, x') \right\} - \mathcal{Z}^t(x) \log \left( \hat{R}_t^\theta(\tilde{x}, x) \right) \right] + C.$$

Recall that a single term from the KL sum in $\mathcal{L}_{\text{DT}}$ up to an additive constant independent of $\theta$ is

$$-\mathbb{E}_{q_k(x_k)q_{k+1|k}(x_{k+1}|x_k)} \left[ \log p_{k|k+1}^\theta(x_k|x_{k+1}) \right].$$

Minimizing this term is to sample $(x_k, x_{k+1})$ from the forward dynamics and then maximize the assigned model probability for the pairing in the reverse direction. A similar idea can be used to understand $\mathcal{L}_{\text{CT}}$. First, we write $\log p_{k|k+1}^\theta(x_k|x_{k+1})$ in terms of $\hat{R}_k^\theta$ as

$$\log p_{k|k+1}^\theta(x_k|x_{k+1}) = \delta_{x_k, x_{k+1}} \left( \hat{R}_k^\theta(x_k, x_k)\Delta t + o(\Delta t) \right)$$
$$+ (1 - \delta_{x_k, x_{k+1}}) \log \left( \hat{R}_k^\theta(x_{k+1}, x_k)\Delta t + o(\Delta t) \right)$$

where we have separated the cases when $x_k = x_{k+1}$ and when $x_k \neq x_{k+1}$ (see the proof of $\mathcal{L}_{\text{CT}}$ for the full details). The first term will become the $\sum_{x' \neq x} \hat{R}_t^\theta(x, x')$ term in $\mathcal{L}_{\text{CT}}$ whilst the second term will become the $\mathcal{Z}^t(x) \log \left( \hat{R}_t^\theta(\tilde{x}, x) \right)$ term. Now, when we minimize $\mathcal{L}_{\text{CT}}$, we are sampling $(x, \tilde{x})$ from the forward process and then maximizing the assigned model probability for the pairing in the reverse direction, just as in $\mathcal{L}_{\text{DT}}$. The slight extra complexity comes from the fact we are considering the case when $x_k = x_{k+1}$ and the case when $x_k \neq x_{k+1}$ separately. When $x_k = x_{k+1}$, this corresponds to the first term in $\mathcal{L}_{\text{CT}}$ which we can see is minimizing the reverse rate out of $x$ which is exactly maximizing the model probability for no transition to occur. When $x_k \neq x_{k+1}$, this corresponds to the second term in $\mathcal{L}_{\text{CT}}$, which is maximizing the reverse rate from $\tilde{x}$ to $x$ which in turn maximizes the model probability for the $\tilde{x}$ to $x$ transition to occur.

### C.2 Conditional Form

For the conditional form of $\mathcal{L}_{\text{CT}}$, denoted as $\bar{\mathcal{L}}_{\text{CT}}$, we instead upper bound the negative conditional model log-likelihood, $\mathbb{E}_{p_{\text{data}}(x_0, y)}[- \log p_0^\theta(x_0|y)]$ where $y$ is our conditioner. $\bar{\mathcal{L}}_{\text{CT}}$ has the following form

$$\bar{\mathcal{L}}_{\text{CT}}(\theta) = T\, \mathbb{E}_{t \sim \mathcal{U}(0,T)p_{\text{data}}(x_0, y)q_{t|0}(x|x_0)r_t(\tilde{x}|x)} \left[ \left\{ \sum_{x' \neq x} \hat{R}_t^\theta(x, x'|y) \right\} - \mathcal{Z}^t(x) \log \left( \hat{R}_t^\theta(\tilde{x}, x|y) \right) \right] + C,$$

where

$$\hat{R}_t^\theta(x, \tilde{x}|y) = R_t(\tilde{x}, x) \sum_{x_0} \frac{q_{t|0}(\tilde{x}|x_0)}{q_{t|0}(x|x_0)} p_{0|t}^\theta(x_0|x, y) \quad \text{for} \quad x \neq \tilde{x}.$$
$$= - \sum_{x' \neq x} \hat{R}_t^\theta(x, x'|y) \quad \text{for} \quad x = \tilde{x}$$

This follows easily from considering the conditional form of the discret time ELBO, $\bar{\mathcal{L}}_{\text{DT}}$ and using the same arguments as before to go from discrete time to continuous time.

$$\mathbb{E}_{p_{\text{data}}(x_0, y)}[- \log p_0^\theta(x_0|y)] \leq \mathbb{E}_{p_{\text{data}}(x_0, y)q_{1:K|0}(x_{1:K}|x_0)} \left[ - \log \frac{p_{0:K}^\theta(x_{0:K}|y)}{q_{1:K|0}(x_{1:K}|x_0)} \right] = \bar{\mathcal{L}}_{\text{DT}}$$

### C.3 Continuous Time ELBO with Factorization Assumptions

In the following Proposition, we show the form of $\mathcal{L}_{\text{CT}}$ when we use a factorized forward process. We note that in the proof we rearrange the sampling distribution from $p_{\text{data}}(\boldsymbol{x}_0^{1:D})q_{t|0}(\boldsymbol{x}^{1:D}|\boldsymbol{x}_0^{1:D})r_t(\tilde{\boldsymbol{x}}^{1:D}|\boldsymbol{x}^{1:D})$ to $p_{\text{data}}(\boldsymbol{x}_0^{1:D})\psi_t(\tilde{\boldsymbol{x}}^{1:D}|\boldsymbol{x}_0^{1:D})\phi_t(\boldsymbol{x}^{1:D}|\tilde{\boldsymbol{x}}^{1:D}, \boldsymbol{x}_0^{1:D})$. This is not strictly necessary but it allows us to analytically sum over the intermediate $\boldsymbol{x}^{1:D}$ variable which greatly reduces the variance of the resulting objective.

**Proposition 7.** *The $\mathcal{L}_{\mathrm{CT}}$ objective when we substitute in the factorized forms for the forward and reverse process given in Proposition 3 is*

$$\mathcal{L}_{\mathrm{CT}} = T\,\mathbb{E}_{t\sim\mathcal{U}(0,T)p_{\mathrm{data}}(\boldsymbol{x}_0^{1:D})q_{t|0}(\boldsymbol{x}^{1:D}|\boldsymbol{x}_0^{1:D})}\left[\sum_{d=1}^{D}\sum_{x'^d\neq x^d}\hat{R}_t^{\theta\,d}(\boldsymbol{x}^{1:D},x'^d)\right]$$

$$-\,T\,\mathbb{E}_{t\sim\mathcal{U}(0,T)p_{\mathrm{data}}(\boldsymbol{x}_0^{1:D})\psi_t(\tilde{\boldsymbol{x}}^{1:D}|\boldsymbol{x}_0^{1:D})}\left[\sum_{d=1}^{D}\sum_{x^d\neq\tilde{x}^d}\phi_t(x^d|\tilde{\boldsymbol{x}}^{1:D},\boldsymbol{x}_0^{1:D})\mathcal{Z}^t(\tilde{\boldsymbol{x}}^{1:D/d}\circ x^d)\log\left(\hat{R}_t^{\theta\,d}(\tilde{\boldsymbol{x}}^{1:D},x^d)\right)\right]$$

$$+\,C$$

*with*

$$\hat{R}_t^{\theta\,d}(\boldsymbol{x}^{1:D},\tilde{x}^d) = R_t^d(\tilde{x}^d,x^d)\sum_{x_0^d}p_{0|t}^\theta(x_0^d|\boldsymbol{x}^{1:D})\frac{q_{t|0}(\tilde{x}^d|x_0^d)}{q_{t|0}(x^d|x_0^d)}$$

$$\mathcal{Z}^t(\boldsymbol{x}^{1:D}) = \sum_{d=1}^{D}\sum_{\tilde{x}^d\neq x^d}R_t^d(x^d,\tilde{x}^d)$$

$$\phi_t(x^d|\tilde{\boldsymbol{x}}^{1:D},\boldsymbol{x}_0^{1:D}) = \frac{R_t^d(x^d,\tilde{x}^d)q_{t|0}(\tilde{\boldsymbol{x}}^{1:D\backslash d}\circ x^d|\boldsymbol{x}_0^{1:D})}{\mathcal{Z}^t(\tilde{\boldsymbol{x}}^{1:D\backslash d}\circ x^d)\sum_{d'=1}^{D}\sum_{x'^{d'}\neq\tilde{x}^{d'}}\frac{R_t^{d'}(x'^{d'},\tilde{x}^{d'})}{\mathcal{Z}^t(\tilde{\boldsymbol{x}}^{1:D\backslash d'}\circ x'^{d'})}q_{t|0}(\tilde{\boldsymbol{x}}^{1:D\backslash d'}\circ x'^{d'}|\boldsymbol{x}_0^{1:D})}$$

*where $\circ$ represents the concatenation of a $D-1$ dimensional vector, $\boldsymbol{x}^{1:D\backslash d}$ with a scalar $x^d$, such that the resultant $D$ dimensional vector has $x^d$ at its $d^{\mathrm{th}}$ dimension. $\psi_t(\tilde{\boldsymbol{x}}^{1:D}|\boldsymbol{x}_0^{1:D})$ is defined as the marginal of the forward noising process joint, $\int q_{t|0}(\boldsymbol{x}^{1:D}|\boldsymbol{x}_0^{1:D})r_t(\tilde{\boldsymbol{x}}^{1:D}|\boldsymbol{x}^{1:D})d\boldsymbol{x}^{1:D}$.*

*Proof.* We first re-write the general form of $\mathcal{L}_{\mathrm{CT}}$ here

$$\mathcal{L}_{\mathrm{CT}}(\theta) = T\,\mathbb{E}_{t\sim\mathcal{U}(0,T)q_t(x)r_t(\tilde{x}|x)}\left[\left\{\sum_{x'\neq x}\hat{R}_t^\theta(x,x')\right\} - \mathcal{Z}^t(x)\log\left(\hat{R}_t^\theta(\tilde{x},x)\right)\right] + C$$

where

$$\mathcal{Z}^t(x) = \sum_{x'\neq x}R_t(x,x') \qquad\qquad r_t(\tilde{x}|x) = (1-\delta_{\tilde{x},x})R_t(x,\tilde{x})/\mathcal{Z}^t(x).$$

With a factorized forward process, $\hat{R}_t^\theta$ becomes

$$\hat{R}_t^{\theta\,1:D}(\boldsymbol{x}^{1:D},\tilde{\boldsymbol{x}}^{1:D}) = \sum_{d=1}^{D}\hat{R}_t^{\theta\,d}(\boldsymbol{x}^{1:D},\tilde{x}^d)\delta_{\boldsymbol{x}^{1:D\backslash d},\tilde{\boldsymbol{x}}^{1:D\backslash d}}$$

where

$$\hat{R}_t^{\theta\,d}(\boldsymbol{x}^{1:D},\tilde{x}^d) = R_t^d(\tilde{x}^d,x^d)\sum_{x_0^d}p_{0|t}^\theta(x_0^d|\boldsymbol{x}^{1:D})\frac{q_{t|0}(\tilde{x}^d|x_0^d)}{q_{t|0}(x^d|x_0^d)}$$

Substituting this form for $\hat{R}_t^{\theta\,1:D}$ into the first term in $\mathcal{L}_{\mathrm{CT}}$ we get

$$\sum_{\boldsymbol{x}'^{1:D}\neq\boldsymbol{x}^{1:D}}\sum_{d=1}^{D}\hat{R}_t^{\theta\,d}(\boldsymbol{x}^{1:D},x'^d)\delta_{\boldsymbol{x}^{1:D\backslash d},\boldsymbol{x}'^{1:D\backslash d}}$$

$$=\sum_{d=1}^{D}\sum_{x'^d}\hat{R}_t^{\theta\,d}(\boldsymbol{x}^{1:D},x'^d)\sum_{\boldsymbol{x}'^{1:D\backslash d}}\delta_{\boldsymbol{x}^{1:D\backslash d},\boldsymbol{x}'^{1:D\backslash d}}\left(1-\delta_{\boldsymbol{x}'^{1:D},\boldsymbol{x}^{1:D}}\right)$$

$$=\sum_{d=1}^{D}\sum_{x'^d\neq x^d}\hat{R}_t^{\theta\,d}(\boldsymbol{x}^{1:D},x'^d)$$

Now we tackle the second term in $\mathcal{L}_{\mathrm{CT}}$. We first re-arrange the distribution over which we take the expectation:

$$p_{\mathrm{data}}(\boldsymbol{x}_0^{1:D})q_{t|0}(\boldsymbol{x}^{1:D}|\boldsymbol{x}_0^{1:D})r_t(\tilde{\boldsymbol{x}}^{1:D}|\boldsymbol{x}^{1:D}) = p_{\mathrm{data}}(\boldsymbol{x}_0^{1:D})\psi_t(\tilde{\boldsymbol{x}}^{1:D}|\boldsymbol{x}_0^{1:D})\phi_t(\boldsymbol{x}^{1:D}|\tilde{\boldsymbol{x}}^{1:D},\boldsymbol{x}_0^{1:D})$$

We have,

$$\phi_t(\boldsymbol{x}^{1:D}|\tilde{\boldsymbol{x}}^{1:D}, \boldsymbol{x}_0^{1:D}) \propto q_{t|0}(\boldsymbol{x}^{1:D}|\boldsymbol{x}_0^{1:D})r_t(\tilde{\boldsymbol{x}}^{1:D}|\boldsymbol{x}^{1:D})$$

$$= q_{t|0}(\boldsymbol{x}^{1:D}|\boldsymbol{x}_0^{1:D})(1 - \delta_{\tilde{\boldsymbol{x}}^{1:D},\boldsymbol{x}^{1:D}})\frac{\sum_{d=1}^{D} R_t^d(x^d, \tilde{x}^d)\delta_{\boldsymbol{x}^{1:D\backslash d},\tilde{\boldsymbol{x}}^{1:D\backslash d}}}{\mathcal{Z}^t(\boldsymbol{x}^{1:D})}$$

$$= \sum_{d=1}^{D} \frac{R_t^d(x^d, \tilde{x}^d)}{\mathcal{Z}^t(\tilde{\boldsymbol{x}}^{1:D\backslash d} \circ x^d)}q_{t|0}(\tilde{\boldsymbol{x}}^{1:D\backslash d} \circ x^d|\boldsymbol{x}_0^{1:D})\delta_{\boldsymbol{x}^{1:D\backslash d},\tilde{\boldsymbol{x}}^{1:D\backslash d}}(1 - \delta_{\tilde{\boldsymbol{x}}^{1:D},\boldsymbol{x}^{1:D}})$$

To find the normalization constant, we can sum the proportional term over $\boldsymbol{x}^{1:D}$

$$\sum_{\boldsymbol{x}^{1:D}} \sum_{d=1}^{D} \frac{R_t^d(x^d, \tilde{x}^d)}{\mathcal{Z}^t(\tilde{\boldsymbol{x}}^{1:D\backslash d} \circ x^d)}q_{t|0}(\tilde{\boldsymbol{x}}^{1:D\backslash d} \circ x^d|\boldsymbol{x}_0^{1:D})\delta_{\boldsymbol{x}^{1:D\backslash d},\tilde{\boldsymbol{x}}^{1:D\backslash d}}(1 - \delta_{\tilde{\boldsymbol{x}}^{1:D},\boldsymbol{x}^{1:D}})$$

$$= \sum_{d=1}^{D} \sum_{x^d \neq \tilde{x}^d} \frac{R_t^d(x^d, \tilde{x}^d)}{\mathcal{Z}^t(\tilde{\boldsymbol{x}}^{1:D\backslash d} \circ x^d)}q_{t|0}(\tilde{\boldsymbol{x}}^{1:D\backslash d} \circ x^d|\boldsymbol{x}_0^{1:D})$$

Therefore,

$$\phi_t(\boldsymbol{x}^{1:D}|\tilde{\boldsymbol{x}}^{1:D}, \boldsymbol{x}_0^{1:D}) = (1 - \delta_{\tilde{\boldsymbol{x}}^{1:D},\boldsymbol{x}^{1:D}}) \sum_{d=1}^{D} \phi_t(x^d|\tilde{\boldsymbol{x}}^{1:D}, \boldsymbol{x}_0^{1:D})\delta_{\boldsymbol{x}^{1:D\backslash d},\tilde{\boldsymbol{x}}^{1:D\backslash d}}$$

where

$$\phi_t(x^d|\tilde{\boldsymbol{x}}^{1:D}, \boldsymbol{x}_0^{1:D}) = \frac{R_t^d(x^d, \tilde{x}^d)q_{t|0}(\tilde{\boldsymbol{x}}^{1:D\backslash d} \circ x^d|\boldsymbol{x}_0^{1:D})}{\mathcal{Z}^t(\tilde{\boldsymbol{x}}^{1:D\backslash d} \circ x^d) \sum_{d'=1}^{D} \sum_{x'^{d'} \neq \tilde{x}^{d'}} \frac{R_t^{d'}(x'^{d'}, \tilde{x}^{d'})}{\mathcal{Z}^t(\tilde{\boldsymbol{x}}^{1:D\backslash d'} \circ x'^{d'})}q_{t|0}(\tilde{\boldsymbol{x}}^{1:D\backslash d'} \circ x'^{d'}|\boldsymbol{x}_0^{1:D})}$$

Now we write the second term as

$$T\, \mathbb{E}_{t\sim\mathcal{U}(0,T)p_{\text{data}}(\boldsymbol{x}_0^{1:D})\psi_t(\tilde{\boldsymbol{x}}^{1:D}|\boldsymbol{x}_0^{1:D})} \left[ -\sum_{\boldsymbol{x}^{1:D}} \phi_t(\boldsymbol{x}^{1:D}|\tilde{\boldsymbol{x}}^{1:D}, \boldsymbol{x}_0^{1:D})\mathcal{Z}^t(\boldsymbol{x}^{1:D}) \log \hat{R}_t^{\theta\,1:D}(\tilde{\boldsymbol{x}}^{1:D}, \boldsymbol{x}^{1:D}) \right]$$

$$= -T\, \mathbb{E}_{t\sim\mathcal{U}(0,T)p_{\text{data}}(\boldsymbol{x}_0^{1:D})\psi_t(\tilde{\boldsymbol{x}}^{1:D}|\boldsymbol{x}_0^{1:D})} \left[ \sum_{d=1}^{D} \sum_{x^d \neq \tilde{x}^d} \phi_t(x^d|\tilde{\boldsymbol{x}}^{1:D}, \boldsymbol{x}_0^{1:D})\mathcal{Z}^t(\tilde{\boldsymbol{x}}^{1:D/d} \circ x^d) \log \left( \hat{R}_t^{\theta\,d}(\tilde{\boldsymbol{x}}^{1:D}, x^d) \right) \right]$$

$\square$

## C.4 One Forward Pass

To evaluate the $\mathcal{L}_{\text{CT}}$ objective, we naively need to perform two forward passes of the denoising network: $p_{0|t}^{\theta}(x_0|x)$ to calculate $\hat{R}_t^{\theta}(x, x')$ and $p_{0|t}^{\theta}(x_0|\tilde{x})$ to calculate $\hat{R}_t^{\theta}(\tilde{x}, x)$. This is wasteful because $\tilde{x}$ is created from $x$ by applying a single forward transition which on multi-dimensional problems means $\tilde{x}$ differs from $x$ in only a single dimension. To exploit the fact that $\tilde{x}$ and $x$ are very similar, we approximate the sample $x \sim q_t(x)$ with the sample $\tilde{x} \sim \sum_x q_t(x)r_t(\tilde{x}|x)$. This gives the more efficient objective,

$$\mathcal{L}_{\text{eCT}}(\theta) = T\, \mathbb{E}_{t\sim\mathcal{U}(0,T)q_t(x)r_t(\tilde{x}|x)} \left[ \left\{ \sum_{x' \neq \tilde{x}} \hat{R}_t^{\theta}(\tilde{x}, x') \right\} - \mathcal{Z}^t(x) \log \left( \hat{R}_t^{\theta}(\tilde{x}, x) \right) \right] + C$$

Table 3: Metrics on the monophonic music dataset comparing training with the efficient $\mathcal{L}_{\text{eCT}}$ objective vs the original $\mathcal{L}_{\text{CT}}$ objective. We compute these over the test set showing mean±std with respect to 5 samples for each test song.

| Model | Hellinger Distance | Proportion of Outliers |
|---|---|---|
| $\tau$LDR-0 Uniform $\mathcal{L}_{\text{eCT}}$ | $0.3765 \pm 0.0013$ | $0.1106 \pm 0.0010$ |
| $\tau$LDR-0 Uniform $\mathcal{L}_{\text{CT}}$ | $0.3797 \pm 0.0009$ | $0.1128 \pm 0.0007$ |

The approximation is valid because $q_t(x)$ and $\sum_x q_t(x) r_t(\tilde{x}|x)$ are very similar distributions, as we now show.

$$\sum_x q_t(x) r_t(\tilde{x}|x) = \sum_x q_t(x)(1 - \delta_{x,\tilde{x}}) \frac{R_t(x, \tilde{x})}{\sum_{x' \neq x} R_t(x, x')}$$

$$\propto -q_t(\tilde{x}) R_t(\tilde{x}, \tilde{x}) + \sum_x q_t(x) R_t(x, \tilde{x})$$

$$= q_t(\tilde{x}) \sum_{x' \neq \tilde{x}} R_t(\tilde{x}, x') + \partial_t q_t(\tilde{x})$$

$$\propto q_t(\tilde{x}) + \frac{1}{\sum_{x' \neq x} R_t(\tilde{x}, x')} \partial_t q_t(\tilde{x})$$

$$= q_t(\tilde{x}) + \delta t \, \partial_t q_t(\tilde{x})$$

where on the third line we have used the Kolmogorov forward equation and defined $\delta_t = 1 / \sum_{x' \neq x} R_t(\tilde{x}, x')$. The distribution $\sum_x q_t(x) r_t(\tilde{x}|x)$ is therefore $q_{t+\delta t}(\tilde{x})$ approximated using a first-order Taylor expansion around $q_t(\tilde{x})$. We notice that $\delta t$ is the average time to the next transition at time $t$. $\delta t$ can be calculated for the practical settings we consider, its varies between $2 \times 10^{-6} T$ and $2 \times 10^{-8} T$ in the image modelling task and is $1 \times 10^{-3} T$ in the monophonic music task.

We perform an ablation experiment comparing between training with $\mathcal{L}_{\text{eCT}}$ and $\mathcal{L}_{\text{CT}}$ on the monophonic music dataset, the results are shown in Table 3. We find that we gain a small boost in performance when using the more efficient $\mathcal{L}_{\text{eCT}}$ objective alongside the improved efficiency. We hypothesize that this is because of a slight reduction in variance for the $\mathcal{L}_{\text{eCT}}$ objective due to increased negative correlation between the two terms in the objective when $\tilde{x}$ is shared between them.

## D   Direct Denoising Model Supervision

Following [8], we can introduce direct $p^\theta_{0|t}$ supervision into the optimization objective which has been found empirically to improve performance. We first contextualize the change by expressing $\mathcal{L}_{\text{CT}}$ with the dependence on $p^\theta_{0|t}$ made explicit.

$$\mathcal{L}_{\text{CT}} = T \, \mathbb{E}_{t \sim \mathcal{U}(0,T) q_t(x) r_t(\tilde{x}|x)} \left[ \left\{ \sum_{x' \neq x} R_t(x', x) \sum_{x_0} \frac{q_{t|0}(x'|x_0)}{q_{t|0}(x|x_0)} p^\theta_{0|t}(x_0|x) \right\} \right.$$

$$\left. - \mathcal{Z}^t(x) \log \left( R_t(x, \tilde{x}) \sum_{x_0} \frac{q_{t|0}(x|x_0)}{q_{t|0}(\tilde{x}|x_0)} p^\theta_{0|t}(x_0|\tilde{x}) \right) \right] + C$$

The signal for $p^\theta_{0|t}(x_0|x)$ comes through a sum over $x_0$ weighted by the ratio $\frac{q_{t|0}(x|x_0)}{q_{t|0}(\tilde{x}|x_0)}$. We can also provide a direct denoising signal by predicting the clean datapoint $x_0$ from the corrupted version $x$ and using the negative log-likelihood loss.

$$L_{ll}(\theta) = T \, \mathbb{E}_{t \sim \mathcal{U}(0,T) p_{\text{data}}(x_0) q_{t|0}(x|x_0)} \left[ -\log p^\theta_{0|t}(x_0|x) \right]$$

**Proposition 8.** *The true denoising distribution, $q_{0|t}$, minimizes $L_{ll}$*

*Proof.*

$$T\,\mathbb{E}_{t\sim\mathcal{U}(0,T)q_t(x)}\left[\mathrm{KL}\left(q_{0|t}(x_0|x)\,||\,p_{0|t}^\theta(x_0|x)\right)\right]$$

$$=T\,\mathbb{E}_{t\sim\mathcal{U}(0,T)p_{\mathrm{data}}(x_0)q_{t|0}(x|x_0)}\left[-\log p_{0|t}^\theta(x_0|x)\right]+C$$

where $C$ is a constant independent of $\theta$. Therefore, minimizing $L_{ll}$ is equivalent to minimizing the KL divergence between $q_{0|t}$ and $p_{0|t}^\theta$, which is minimized when $p_{0|t}^\theta=q_{0|t}$. $\qquad\square$

If we obtain the true denoising distribution, $p_{0|t}^\theta=q_{0|t}$, then we will have the true reverse rate, $\hat{R}_t$. [8] find that optimizing with an objective combining $L_{ll}$ and $\mathcal{L}_{\mathrm{DT}}$ performs best, which we can also do in continuous time

$$\min_\theta\quad\mathcal{L}_{\mathrm{CT}}(\theta)+\lambda L_{ll}(\theta)$$

where $\lambda$ is a hyperparameter. In [8], it was found that training with $L_{ll}$ alone resulted in poorer performance than when the ELBO was included in the loss. We provide a theoretical hypothesis as to why this may be the case here. We show that minimizing $L_{ll}$ is equivalent to minimizing an upper bound on the negative ELBO in discrete time and thus by training with $L_{ll}$ we are simply minimizing a looser bound on the negative model log-likelihood than if we were to use the negative ELBO directly.

**Proposition 9.** *Minimizing the sum of negative log-likelihoods*

$$\sum_{k=0}^{K-1}\mathbb{E}_{p_{\mathrm{data}}(x_0)q_{k+1|0}(x_{k+1}|x_0)}\left[-\log p_{0|k+1}^\theta(x_0|x_{k+1})\right]$$

*is equivalent to minimizing an upper bound on the negative ELBO.*

*Proof.*

$$\mathcal{L}_{\mathrm{DT}}(\theta)=\mathbb{E}_{p_{\mathrm{data}}(x_0)}\left[\mathrm{KL}(q_{K|0}(x_K|x_0)||p_{\mathrm{ref}}(x_K))-\mathbb{E}_{q_{1|0}(x_1|x_0)}\left[\log p_{0|1}^\theta(x_0|x_1)\right]\right.$$

$$\left.+\sum_{k=1}^{K-1}\mathbb{E}_{q_{k+1|0}(x_{k+1}|x_0)}\left[\mathrm{KL}(q_{k|k+1,0}(x_k|x_{k+1},x_0)||p_{k|k+1}^\theta(x_k|x_{k+1}))\right]\right]$$

Consider one term from the sum

$$L_k=\mathbb{E}_{p_{\mathrm{data}}(x_0)q_{k+1|0}(x_{k+1}|x_0)}\left[\mathrm{KL}(q_{k|k+1,0}(x_k|x_{k+1},x_0)||p_{k|k+1}^\theta(x_k|x_{k+1})\right]$$

$$=\mathbb{E}_{q_{k+1}(x_{k+1})q_{k|k+1}(x_k|x_{k+1})}\left[-\log p_{k|k+1}^\theta(x_k|x_{k+1})\right]$$

$$+\mathbb{E}_{p_{\mathrm{data}}(x_0)q_{k+1|0}(x_{k+1}|x_0)q_{k|k+1,0}(x_k|x_{k+1},x_0)}\left[\log q_{k|k+1,0}(x_k|x_{k+1},x_0)\right]$$

Now,

$$\mathbb{E}_{q_{k+1}(x_{k+1})q_{k|k+1}(x_k|x_{k+1})}\left[-\log p_{k|k+1}^\theta(x_k|x_{k+1})\right]$$

$$=\mathbb{E}_{q_{k+1}(x_{k+1})q_{k|k+1}(x_k|x_{k+1})}\left[-\log\sum_{\tilde{x}_0}q(x_k|\tilde{x}_0,x_{k+1})p_{0|k+1}^\theta(\tilde{x}_0|x_{k+1})\right]$$

$$=\mathbb{E}_{q_{k+1}(x_{k+1})q_{k|k+1}(x_k|x_{k+1})}\left[-\log\sum_{\tilde{x}_0}\frac{q_{0|k}(\tilde{x}_0|x_k)q_{k|k+1}(x_k|x_{k+1})}{q_{0|k+1}(\tilde{x}_0|x_{k+1})}p_{0|k+1}^\theta(\tilde{x}_0|x_{k+1})\right]$$

$$\leq\mathbb{E}_{q_{k+1}(x_{k+1})q_{k|k+1}(x_k|x_{k+1})q_{0|k}(\tilde{x}_0|x_k)}\left[-\log\frac{q_{k|k+1}(x_k|x_{k+1})}{q_{0|k+1}(\tilde{x}_0|x_{k+1})}p_{0|k+1}^\theta(\tilde{x}_0|x_{k+1})\right]$$

$$=\mathbb{E}_{p_{\mathrm{data}}(x_0)q_{k+1|0}(x_{k+1}|x_0)}\left[-\log p_{0|k+1}^\theta(x_0|x_{k+1})\right]$$

$$+\mathbb{E}_{p_{\mathrm{data}}(x_0)q_{k|0}(x_k|x_0)q_{k+1|k}(x_{k+1}|x_k)}\left[-\log\frac{q_{k|k+1}(x_k|x_{k+1})}{q_{0|k+1}(x_0|x_{k+1})}\right]$$

Therefore,

$$
\begin{aligned}
\mathcal{L}_{\text{DT}} \leq \sum_{k=0}^{K-1} & \left\{ \mathbb{E}_{p_{\text{data}}(x_0)q_{k+1|0}(x_{k+1}|x_0)} \left[ -\log p^{\theta}_{0|k+1}(x_0|x_{k+1}) \right] \right\} \\
& + \mathbb{E}_{p_{\text{data}}(x_0)} \left[ \text{KL}(q_{K|0}(x_K|x_0)||p_K(x_K)) \right] \\
& + \sum_{k=1}^{K-1} \left\{ \mathbb{E}_{p_{\text{data}}(x_0)q_{k+1|0}(x_{k+1}|x_0)q_{k|k+1,0}(x_k|x_{k+1},x_0)} \left[ \log q_{k|k+1,0}(x_k|x_{k+1},x_0) \right] \right. \\
& \qquad \left. + \mathbb{E}_{p_{\text{data}}(x_0)q_{k|0}(x_k|x_0)q_{k+1|k}(x_{k+1}|x_k)} \left[ -\log \frac{q_{k|k+1}(x_k|x_{k+1})}{q_{0|k+1}(x_0|x_{k+1})} \right] \right\}
\end{aligned}
$$

We can see that only the first term depends on $\theta$. □

## E  Choice of Forward Process

We need to choose the structure of $R_t$ such that we can analytically obtain $q_{t|0}$ marginals to enable efficient training.

**Proposition 10.** *If $R_t$ and $R_{t'}$ commute for all $t$, $t'$ then $q_{t|0}(x = j|x_0 = i) = \left( exp\left[ \int_0^t R_s ds \right] \right)_{ij}$ where exp here is understood to be the matrix exponential function.*

*Proof.* Let $(P_t)_{ij} = q_{t|0}(x = j|x_0 = i)$. We show that $P_t = \exp\left( \int_0^t R_s ds \right)$ is a solution to the Kolmogorov forward equation, which in matrix form reads, $\partial_t P_t = P_t R_t$. Writing the matrix exponential in sum form

$$
\begin{aligned}
P_t &= \sum_{k=0}^{\infty} \frac{1}{k!} \left( \int_0^t R_s ds \right)^k \\
&= \text{Id} + \int_0^t R_s ds + \frac{1}{2!} \left( \int_0^t R_s ds \right)^2 + \frac{1}{3!} \left( \int_0^t R_s ds \right)^3 + \ldots
\end{aligned}
$$

Now, differentiating and using the fact that $R_t$, $R_{t'}$ commute.

$$
\begin{aligned}
\partial_t P_t &= R_t + \int_0^t R_s ds R_t + \frac{1}{2!} \left( \int_0^t R_s ds \right)^2 R_t + \ldots \\
&= \left\{ \sum_{k=0}^{\infty} \frac{1}{k!} \left( \int_0^t R_s ds \right)^k \right\} R_t \\
&= P_t R_t
\end{aligned}
$$

□

As stated in the main text, we achieve the commutative property by selecting $R_t = \beta(t)R_b$ where $\beta(t)$ is a time dependent scalar and $R_b$ is a constant base matrix. We can utilize the eigendecomposition

of $R_b = Q\Lambda Q^{-1}$ to efficiently calculate $P_t$,

$$P_t = \exp\left(\int_0^t \beta(s)R_b ds\right)$$

$$= \sum_{k=0}^{\infty} \frac{1}{k!}\left(\int_0^t \beta(s)R_b ds\right)^k$$

$$= \sum_{k=0}^{\infty} \frac{1}{k!}\left(Q\Lambda Q^{-1}\int_0^t \beta(s)ds\right)^k$$

$$= \sum_{k=0}^{\infty} \frac{1}{k!}Q\left(\Lambda \int_0^t \beta(s)ds\right)^k Q^{-1}$$

$$= Q\left\{\sum_{k=0}^{\infty} \frac{1}{k!}\left(\Lambda \int_0^t \beta(s)ds\right)^k\right\}Q^{-1}$$

$$= Q\exp\left[\Lambda \int_0^t \beta(s)ds\right]Q^{-1}$$

Since $\Lambda$ is a diagonal matrix, the matrix exponential coincides with the element wise exponential making the final expression tractable to compute. We choose $\beta(t) = ab^t \log b$ because this makes the integral which dictates the variance of $q_{t|0}$ have a simple form $\int_0^t \beta(s)ds = ab^t - a$.

For categorical problems, we found a uniform rate matrix works well, $R_t = \beta\mathbb{1}\mathbb{1}^T - \beta S\mathrm{Id}$. This is directly analogous to the discrete time uniform transition matrix: $P = \alpha\mathbb{1}\mathbb{1}^T + (1 - S\alpha)\mathrm{Id}$ with $\alpha$ depending on the time discretization used. Indeed, if one calculates the corresponding discrete transition matrix for the uniform $R_t$ rate through the matrix exponential, the uniform transition matrix is obtained. Another categorical corruption process is the absorbing state process. In discrete time, the transition matrix is given by $P = \alpha\mathbb{1}\mathbf{e}_*^T + (1 - \alpha)\mathrm{Id}$ where $\mathbf{e}_*$ is the one-hot encoding of the absorbing state. The corresponding absorbing state continuous time process has transition rate matrix: $R_t = \beta\mathbf{1}\mathbf{e}_*^T - \beta\mathrm{Id}$. The correspondence for more complex transition matrices e.g. the Discretized Gaussian matrix in [8] is much harder to find analytically especially if the time inhomogeneous case is considered. For datasets with an ordinal structure, we construct a new rate matrix that maintains a bias towards nearby states using a similar approach as that taken by [8] to construct the Discretized Gaussian matrix.

We construct this matrix by first picking a desired stationary distribution, $p_{\text{ref}}$, and then filling in matrix entries such that we encourage transitions to nearby states whilst keeping $p_{\text{ref}}$ as our stationary distribution. Specifically, we let $p_{\text{ref}}$ be a discretized Gaussian over the state space, i.e.

$$p_{\text{ref}}(x) \propto \exp\left[-\frac{(x - \mu_0)^2}{2\sigma_0^2}\right]$$

To find a condition on the rate such that this is the case, recall the Kolmogorov differential equation for the marginals

$$\partial_t q_t(x) = \sum_{\tilde{x}} q_t(\tilde{x})R_b(\tilde{x}, x)$$

Now, consider a rate that is in detailed balance with $p_{\text{ref}}$

$$p_{\text{ref}}(\tilde{x})R_b(\tilde{x}, x) = p_{\text{ref}}(x)R_b(x, \tilde{x})$$

Substituting this rate into the Kolmogorov equation, we see that $p_{\text{ref}}$ is the stationary distribution

$$\partial_t p_{\text{ref}}(x) = \sum_{\tilde{x}} p_{\text{ref}}(\tilde{x})R_b(\tilde{x}, x)$$

$$= \sum_{\tilde{x}} p_{\text{ref}}(x)R_b(x, \tilde{x})$$

$$= p_{\text{ref}}(x)\sum_{\tilde{x}} R_b(x, \tilde{x})$$

$$= 0$$

where the last line follows from the fact that the row sum of a rate matrix is zero. Note that any $R_t = \beta(t)R_b$ will also have this stationary distribution as the multiplication by $\beta(t)$ can be seen as just a scaling of the time axis. From the detailed balance equation, we gain a condition on $R_b$ such that our desired $p_{\text{ref}}$ is the stationary distribution

$$\frac{R_b(\tilde{x}, x)}{R_b(x, \tilde{x})} = \frac{p_{\text{ref}}(x)}{p_{\text{ref}}(\tilde{x})} = \exp\left[\frac{(\tilde{x} - \mu_0)^2}{2\sigma_0^2} - \frac{(x - \mu_0)^2}{2\sigma_0^2}\right]$$

This gives constraints on diagonal elements within $R_b$ but does not fully define the entire matrix. To do this, we first make the assumption that $\mu$ is selected to be at the center of the state space. Then we set off diagonal terms to the right of the diagonal in the top half of the rate matrix and off diagonal terms to the left of the diagonal in the bottom half to be 1. Finally, progressing in from the top and bottom of the rate matrix we make definitions of rate matrix values that have not already been defined by the detailed balance condition. For clarity, we provide a pictorial representation of this scheme for an $8 \times 8$ rate matrix below

$$\begin{bmatrix}
\cdot & 1 & \square & \square & \square & \square & \square & \square \\
\triangle & \cdot & 1 & \square & \square & \square & \square & \triangle \\
\triangle & \triangle & \cdot & 1 & \square & \square & \triangle & \triangle \\
\triangle & \triangle & \triangle & \cdot & 1 & \triangle & \triangle & \triangle \\
\triangle & \triangle & \triangle & \triangle & \cdot & \triangle & \triangle & \triangle \\
\triangle & \triangle & \triangle & \square & 1 & \cdot & \triangle & \triangle \\
\triangle & \triangle & \square & \square & \square & 1 & \cdot & \triangle \\
\triangle & \square & \square & \square & \square & \square & 1 & \cdot
\end{bmatrix}$$

where $\square$ represents a value we will define, $\triangle$ represents a value that is defined relative to another entry through the detailed balance condition and $\cdot$ is a diagonal entry that is equal to the negative off diagonal row sum. We could define $\square$ values to be 0 to gain a sparse rate matrix, however, we found in early experiments that allowing transitions to further away states greatly reduces the mixing time and gives better performance. We define $\square$ in each row similarly, by setting it equal to $\exp[-i^2/\sigma_r^2]$ where $i$ is the distance away from the '1' value in that row and $\sigma_r$ is a hyperparameter defining the length scale in state space of a typical transition. This biases our forward process to make transitions between nearby states, at a length scale of $\sigma_r$.

# F  CTMC Simulation

## F.1  Exact CTMC and Tau-Leaping

In this section, we first describe exact CTMC simulation before giving an algorithmic description of tau-leaping.

When a CTMC has a time-homogeneous rate matrix, we can use Gillespie's Algorithm [21, 22, 23] to exactly simulate it. This algorithm is based on the jump chain/holding time definition of the CTMC. It repeats the following two steps:

- Draw a holding time from an exponential distribution with mean $-1/R(x, x)$ and wait in the current state $x$ for that amount of time.
- Sample the next state from $r(\tilde{x}|x) = (1 - \delta_{x,\tilde{x}})\frac{R(x,\tilde{x})}{\sum_{x' \neq x} R(x,x')}$

This Algorithm can be adjusted for the case when we have a time-inhomogeneous rate matrix using the modified next reaction method [33]. However, both algorithms still step through each transition in the CTMC individually and are thus unsuitable in our case because only one dimension would change for each simulation step making it very computationally expensive to produce a sample. Instead we use tau-leaping that allows multiple dimensions to change in a single simulation step. We detail this method in Algorithm 1.

## F.2  Predictor-Corrector Discussion

In this section we compare predictor-corrector sampling schemes as applied to continuous state spaces and discrete state spaces.

**Algorithm 1:** Generative Reverse Process Simulation with Tau-Leaping

---

$t \leftarrow T$
$\boldsymbol{x}_t^{1:D} \sim p_{\text{ref}}(\boldsymbol{x}_T^{1:D})$
**while** $t > 0$ **do**
    Compute $p_{0|t}^\theta(x_0^d|\boldsymbol{x}_t^{1:D}), d = 1, \ldots, D$ with one forward pass of the denoising network
    **for** $d = 1, \ldots, D$ **do**
        **for** $s = 1, \ldots, S \backslash x_t^d$ **do**
            $\hat{R}_t^{\theta\,d}(\boldsymbol{x}_t^{1:D}, s) \leftarrow R_t^d(s, x_t^d) \sum_{x_0^d} p_{0|t}^\theta(x_0^d|\boldsymbol{x}_t^{1:D}) \frac{q_{t|0}(s|x_0^d)}{q_{t|0}(x_t^d|x_0^d)}$
            $P_{ds} \leftarrow \text{Poisson}\left(\tau \hat{R}_t^{\theta\,d}(\boldsymbol{x}_t^{1:D}, s)\right)$
        **end**
    **end**
    **for** $d = 1, \ldots, D$ **do**
        **if** *data is categorical AND* $\sum_{s=1}^S P_{ds} > 1$ **then**
            $x_{t-\tau}^d \leftarrow x_t^d$ // `reject change`
        **else**
            $x_{t-\tau}^d \leftarrow x_t^d + \sum_{s=1}^S P_{ds} \times (s - x_t^d)$
        **end**
    **end**
    $\boldsymbol{x}_{t-\tau}^{1:D} \leftarrow \text{Clamp}(\boldsymbol{x}_{t-\tau}^{1:D}, \min = 1, \max = S)$
    $t \leftarrow t - \tau$
**end**

---

The predictor-corrector scheme in continuous state spaces was introduced in [4]. It consists of alternating between a predictor step and a corrector step:

$$\text{Predictor} \quad \boldsymbol{x}_i \leftarrow \boldsymbol{x}_{i+1} + \gamma_i s_\theta(\boldsymbol{x}_{i+1}, i+1) + \sqrt{\gamma_i} z, \quad z \sim \mathcal{N}(0, I)$$
$$\text{Corrector} \quad \boldsymbol{x}_i \leftarrow \boldsymbol{x}_i + \epsilon_i s_\theta(\boldsymbol{x}_i, i) + \sqrt{2\epsilon_i} z, \quad z \sim \mathcal{N}(0, I)$$

where $\boldsymbol{x}_i$ is the state at sampling step $i$, $s_\theta$ is the learned score model approximating $\nabla_{\boldsymbol{x}} \log q_t(\boldsymbol{x})$ and $\gamma_i, \epsilon_i$ are the step sizes for the predictor and corrector respectively. We see that both take similar forms, except the corrector adds in a factor $\sqrt{2}$ more Gaussian noise during the update step.

In discrete state spaces, rather than sampling using gradient guided stochastic steps as in the continuous state space case, we sample by simulating CTMCs with defined rates. When we take a predictor step, we simulate using $\hat{R}_t^\theta$ and when we take a corrector step we simulate using $R_t^{c\,\theta} = \hat{R}_t^\theta + R_t$. If we simulate the CTMC exactly, we have seen in the previous section that this amounts to sampling next states from the categorical distribution defined by normalizing the row of the rate matrix corresponding to the current state. Therefore, corrector sampling can be seen as sampling from a slightly noisier categorical distribution defined through $R_t^{c\,\theta}$ as compared to the predictor categorical distribution defined through $\hat{R}_t^\theta$. This is analogous to the increased Gaussian noise applied during a corrector step in continuous state spaces.

Adding corrector steps brings the marginal of the samples closer to $q_t(\boldsymbol{x})$ and continued application of the corrector will further explore the domain of $q_t(\boldsymbol{x})$. In previous work on continuous state predictor-corrector methods, the number of corrector steps has been small (e.g. 1 or 2 corrector steps per predictor step) or indeed the corrector steps have been removed altogether. In this work we have found that using up to 10 corrector steps per predictor steps can be beneficial during certain regions of the reverse generative process. Additionally, in continuous state spaces, it has been observed that too many corrector steps can result in unwanted noise in the generated data [34].

We hypothesize that corrector steps are better utilized in discrete state spaces to explore the domain of $q_t(\boldsymbol{x})$ than in continuous state spaces. This is because, the corrector update is defined largely through the reverse rate itself, $\hat{R}_t^\theta$, just with the categorical probabilities being annealed slightly more towards uniform through the addition of the forward rate $R_t$. This may be a more effective update than simply adding extra Gaussian noise in the continuous state space case. Furthermore, the denoising model in continuous state spaces can be seen as outputting a point estimate of $\boldsymbol{x}_0$ of dimension $D$. However,

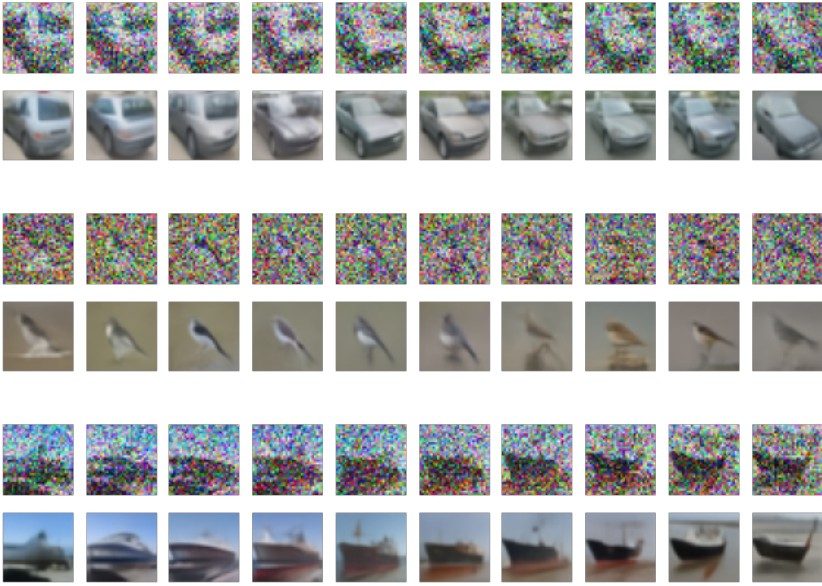

Figure 7: Progression of $\boldsymbol{x}_t$ for $t = 0.4$ by repeated application of corrector steps. In each pair of rows, the top row is $\boldsymbol{x}_t$ whilst the bottom row is the $\boldsymbol{x}_0$ prediction made by $p^\theta_{0|t}(\boldsymbol{x}_0|\boldsymbol{x}_t)$ (argmax of the categorical probabilities in each dimension). Each column represents an additional 100 corrector steps.

in discrete state spaces, the denoising model outputs a categorical distribution over every dimension (output dimension $D \times S$) allowing it to express some uncertainty information in the $\boldsymbol{x}_0$ prediction, albeit with conditional independence between the dimensions. Adding corrector steps in discrete state spaces would then allow information to mix between dimensions for the current time step, exploring modes of $q_t(\boldsymbol{x})$.

We explore this idea on the image modelling task in Figure 7. We run the reverse generative process until time $t = 0.4$ at which point we hold the time constant and apply 1000 corrector steps. We see that the resulting progression of $\boldsymbol{x}_t$ states explores potential local modes of $q_t(\boldsymbol{x})$ in the local region of image space.

## G    Implicit Dimensional Assumptions Made in Discrete Time

In discrete time, the parametric reverse kernel, $p^\theta_{k|k+1}$, is commonly defined through a denoising model $p^\theta_{0|k+1}$. Here, we examine this definition in the multi-dimensional case where the forward process factorizes, as in Appendix C.3 and previous discrete time work [8]. We begin by writing the true full dimensional reverse kernel, $q_{k|k+1}$, in terms of the true denoising distribution, $q_{0|k+1}$.

$$
\begin{aligned}
q_{k|k+1}(\boldsymbol{x}_k^{1:D}|\boldsymbol{x}_{k+1}^{1:D}) &= \prod_{d=1}^{D} q_{k|k+1}(x_k^d|\boldsymbol{x}_k^{1:d-1}, \boldsymbol{x}_{k+1}^{1:D}) \\
&= \prod_{d=1}^{D} \sum_{x_0^d} q_{k,0|k+1}(x_k^d, x_0^d|\boldsymbol{x}_k^{1:d-1}, \boldsymbol{x}_{k+1}^{1:D}) \\
&= \prod_{d=1}^{D} \sum_{x_0^d} q_{0|k+1}(x_0^d|\boldsymbol{x}_k^{1:d-1}, \boldsymbol{x}_{k+1}^{1:D}) q_{k|0,k+1}(x_k^d|x_0^d, x_{k+1}^d)
\end{aligned}
$$

where on the final line we have used the fact that the forward process is independent across dimensions. To create our approximate reverse kernel, $p^\theta_{k|k+1}$, we approximate $q_{0|k+1}(x_0^d|\boldsymbol{x}_k^{1:d-1}, \boldsymbol{x}_{k+1}^{1:D})$ with

$$p_{0|k+1}^{\theta}(x_0^d|\boldsymbol{x}_{k+1}^{1:D}),$$

$$p_{k|k+1}^{\theta}(\boldsymbol{x}_k^{1:D}|\boldsymbol{x}_{k+1}^{1:D}) = \prod_{d=1}^{D} \sum_{x_0^d} p_{0|k+1}^{\theta}(x_0^d|\boldsymbol{x}_{k+1}^{1:D}) q_{k|0,k+1}(x_k^d|x_0^d, x_{k+1}^d)$$

We throw away the extra $\boldsymbol{x}_k^{1:d-1}$ conditioning because we use a non-autoregressive model that takes in $\boldsymbol{x}_{k+1}^{1:D}$ and in a single forward pass gives conditionally independent probabilities over $x_0^d$, $d = 1, \ldots, D$. For finite $K$, this approximation can never match the true kernel because we are not conditioning on all relevant information. Of course, as $K$ gets larger, this approximation becomes more accurate. Since we operate in the continuous regime, we do not have to make this approximation because the conditionally independent denoising model, $q_{0|t}(x_0^d|\boldsymbol{x}^{1:D})$, appears directly in our reverse rate, $\hat{R}_t^{1:D}$, when we factorize the forward process (see Proposition 3).

## H    Experimental Details

In this section, we provide additional details for the experiments we performed applying our method to practical problems. The code for our models is available at `https://github.com/andrew-cr/tauLDR`. Before describing the specifics for each experiment, we first explain the implementation details common to all.

When we evaluate the objective $\mathcal{L}_{\text{CT}}$ on each minibatch of training datapoints, we must sample a time for each from $t \sim \mathcal{U}(0, T)$ which represents the point in the forward process which we will noise to. Training instabilities can be found if $t$ is sampled very close to $0$ because the reverse rate, $\hat{R}_t$, becomes ill-conditioned in this region. This phenomenon is also observed in continuous state space models because the score, $\nabla_x \log q_t(x)$, becomes ill-conditioned close to $t = 0$. The reverse rate and score become ill conditioned close to the start of the forward process because the marginal probability, $q_t(x)$, will be highly peaked around the data manifold and $\log q_t(x)$ will explode in regions that are not close to the data. To avoid these issues, a common trick is to set a minimum time such that $t \sim \mathcal{U}(\epsilon, T)$. $\epsilon$ is set such that the level of noising at $t = \epsilon$ is very small and reverse sampling to this point will produce samples very close to $p_{\text{data}}$. In our experiments, we set $\epsilon = 0.01T$.

During reverse sampling, we use tau-leaping to simulate the reverse process from $t = T$ until $t = \epsilon$ because the reverse rate is not trained for $t < \epsilon$. This produces a sample close to $p_{\text{data}}$. We found improved performance in metrics such as FID if we then complete a final step to remove the small amount of noise that may still be present in the sample. Specifically, we pass the sample through the denoising model $p_{0|t}^{\theta}(x_0|x_t)$ with $t = \epsilon$ to obtain an output of shape $D \times S$ where $D$ is the dimensionality of the problem. This is a probability distribution over the states for each of the dimensions. We set the value of each dimension to the state with the highest probability. This then produces a sample which has all of the noise removed.

The specific value of $T$ within our model is arbitrary because the forward process can be scaled in the time axis to provide the same noising process for any $T$. Therefore, we simply set $T = 1$.

### H.1    Demonstrative Example

Our 2d dataset is created by sampling 1M 2d points from a $32 \times 32$ state space with probability proportional to the pixel values of a $32 \times 32$ grayscale image of a $\tau$ character.

For our forward process, we use a Gaussian rate (see Appendix E) with stationary distribution standard deviation $\sigma_0 = 8$ and rate length scale $\sigma_r = 1$. We use a rate schedule of $\beta(t) = 5 \times 5^t \log(5)$.

To represent $p_{0|t}^{\theta}$ we use a residual MLP. The architecture consists of an input linear layer to lift the input dimension of 2 to the internal network dimension of 16. Then, there are 2 residual blocks each consisting of: a single hidden layer MLP of hidden dimension 32, a residual connection to the

input of the MLP, a layer norm, and finally a FiLM layer [35] modulated by the time embedding. At the output, there is a single linear layer with output size of $2 \times 32 = 64$ representing state probabilities in each of the 2 dimensions. The time is embedded using the Transformer sinusoidal position embedding [36] creating an embedding of size 32. Then, the embedding is further processed by a single hidden layer MLP with hidden layer size 32 and output size 128. To create the FiLM parameters in each residual block, the time embedding is passed through a linear layer with output of size 32 to provide a multiplicative and additive modulation to the state dimension of 16. We minimize the $\mathcal{L}_{\text{CT}}$ objective using Adam with a learning rate of 0.0001 and batch size of 32 for 1M steps.

For the exact simulation we use the next reaction method with modifications for time dependent transition rates [33]. This method steps through each transition in the exact simulation path individually by calculating the time to the next occurrence of each transition type and applying the transition that occurs soonest. Exact algorithmic details can be found in [33]. To calculate the time to the next occurrence for a transition, we need to integrate the reverse rate matrix (eq (13) in [33]). We do this with euler integration with a step size of 0.001.

## H.2   Image Modeling

We train on the CIFAR10 training dataset that contains 50000 images of dimension $3 \times 32 \times 32$. We evaluate the test ELBO on the CIFAR10 test dataset which consists of 10000 images. For the forward noising process, we use the the Gaussian rate (see Appendix E) with stationary distribution standard deviation of $\sigma_0 = 512$ and rate length scale $\sigma_r = 6$. This effectively defines a uniform stationary distribution since the state space is of size 256. We found this performs better than a more concentrated Gaussian. Our $\beta$ schedule is $\beta(t) = 3 \times 100^t \log 100$. This was selected in accordance with $\sigma_r$ such that the overall shape of progression of the $q_{t|0}$ variances approximately matches that of the schedule proposed in [3].

Our $p^\theta_{0|t}$ model is parameterized with the standard U-net [37] architecture introduced in [3]. The network follows the PixelCNN++ backbone [38] with group normalization layers. There are four feature map resolutions ($32 \times 32$ to $4 \times 4$) in the downsampling/upsampling stacks. At each resolution there are two convolutional residual blocks. There is a self-attention block between the residual blocks at the $16 \times 16$ resolution level [39]. The time is input into the network by first embedding with the Transformer sinusoidal position embedding [36]. This time embedding is passed into each residual block by passing it through a SiLU activation [40] and then a linear layer before adding it onto the hidden state within the residual block between the two convolution operations.

The original architecture of [3] has an output of dimension $3 \times 32 \times 32$ as it makes a point prediction of $x_0$ given $x_t$. In order for the model to output probabilities over $x_0$ (i.e. an output dimension of $3 \times 32 \times 32 \times 256$) we make the adjustments suggested in [8]. Specifically, we use their truncated logistic distribution parameterization where the model outputs the mean and log scale of a logistic distribution i.e. an output dimension of $3 \times 32 \times 32 \times 2$. The probability for a state is then the integral of this continuous distribution between this state and the next when mapped onto the real line. To impart a residual inductive bias on the output, the mean of the logistic distribution is taken to be $\tanh(x_t + \mu')$ where $x_t$ is the normalized input into the model and $\mu'$ is mean outputted from the network. The normalization operation takes the input in the range $0, \ldots, 255$ and maps it to $[-1, 1]$. In total, our network has approximately 35.7 million parameters.

We optimize with the auxiliary objective described in Appendix D with $\lambda = 0.001$. Within the auxiliary objective, we use the one-forward pass version of the continuous time ELBO, $\mathcal{L}_{\text{eCT}}$. We optimize with Adam for 2M steps with a learning rate of 0.0002 and batch size of 128. We use the standard set of training tricks to improve optimization [3, 4]. Throughout training we maintain an exponential moving average of the parameters with decay factor 0.9999. These average parameters are used during testing. At the start of optimization we use a linear learning rate warm-up for the first 5000 steps. We clip the gradient norm at a norm value of 1.0. We set the dropout rate for the network at 0.1. The skip connections for each residual block are rescaled by a factor of $\frac{1}{\sqrt{2}}$. The input images have random horizontal flips applied to them during training.

For sampling in Table 1 we set $\tau = 0.001$ for $\tau$LDR-0 and set $\tau = 0.002$ for $\tau$LDR-10. The 10 corrector steps per predictor steps for $\tau$LDR-10 are introduced after $t < 0.1T$. We found that introducing the corrector steps near the end of the reverse sampling process had the best improvement in sample quality for the smallest increase in computational cost. When performing tau-leaping with the corrector rate, $R_t^c$, we have control over what $\tau$ we use since we are sampling a different CTMC (with $q_t$ as its stationary distribution) to the original reverse CTMC. We found that setting the corrector rate $\tau$ to be 1.5 times the original $\tau$ for the reverse CTMC achieves the best performance in this example.

We train using 4 V100 GPUs on an academic research cluster. To calculate Inception and FID values, we use pytorch-fid [41] and a further development [1]. We verified this library produced comparable values to previous work by calculating the Inception and FID scores for the published images from the DDPM [3] method.

We show a large array of unconditional samples from the $\tau$LDR-10 model in Figure 8. We now also present statistics from the reverse sampling process with standard tau-leaping with $\tau = 0.001$. Figure 9 shows the proportion of dimensions that transition during a single step of tau-leaping. We see that during the initial stages, every dimensions changes during every tau-leaping step, but nearer the end of the process, more dimensions will have settled in their final positions and the proportion is less. In Figure 10 we show the proportion of dimensions that are clipped due to proposing an out of bounds jump. Overall, the proportion is small. It is largest at the start of the process when we have initially sampled from the approximately uniform $p_{\text{ref}}$ and there will be dimensions close to the boundary. As pixel values settle to their final values, the proportion reduces. Figure 11 shows the progression of a selection of dimensions during the reverse sampling process. A similar picture emerges where dimensions eventually settle in a region of the state space. We also note that larger jumps are made in a single tau-leaping step nearer the start of the reverse process and smaller jumps are made nearer the end.

### H.3 Monophonic Music

We generate our training dataset from the Lakh pianoroll dataset [29, 30] (license CC By 4.0). This dataset consists of 174,154 multitrack pianorolls. We go through all songs and all tracks within each song and select sequences that match the following criteria: they are monophonic (only one note played at a time), there is not a period longer than one bar in which no note is played, there is more than one type of note played in the sequence and finally there is no one note played for more than 50 time steps out of the total 256 time steps. This removes the uninteresting and trivial sequences present within the dataset. We then remove any duplicates in the result. This leaves us with 6000 training examples and 950 testing examples. Each song consists of 256 time steps (16 per bar) and each time step takes one from 129 values i.e. we have $D = 256$ and $S = 129$. This state value represents either a note from 128 options or a rest. We scramble the ordering of this state space when mapping to the integers from 0 to 128. When we input into the denoising network, we input as one-hot 129 dimensional vectors.

For the forward noising process, we use a uniform rate matrix, $R_b = \mathbb{1}\mathbb{1}^T - S\text{Id}$ and set $\beta(t) = 0.03$. We found a constant in time $\beta(t)$ was sufficient for this dataset. In our comparison, we used a birth/death rate matrix defined as

$$
\begin{bmatrix}
-\lambda & \lambda & 0 & 0 & \dots & 0 & 0 \\
\lambda & -2\lambda & \lambda & 0 & \dots & 0 & 0 \\
0 & \lambda & -2\lambda & \lambda & \dots & 0 & 0 \\
\vdots & \vdots & \vdots & \vdots & \ddots & \vdots & \vdots \\
0 & 0 & 0 & 0 & \dots & \lambda & -\lambda
\end{bmatrix}
$$

this is the rate matrix for a birth/death process. We set $\lambda = 1$ and $\beta(t) = \frac{1}{2} \times 10000^t \log 10000$. These hyperparameters were selected such that the forward process has a steady rate of noising whilst still having $q_T$ very close to $p_{\text{ref}}$. We chose to compare these types of rate matrix because the birth/death rate is inappropriate for this categorical data as adjacent states have no meaning

---
[1] https://github.com/w86763777/pytorch-gan-metrics

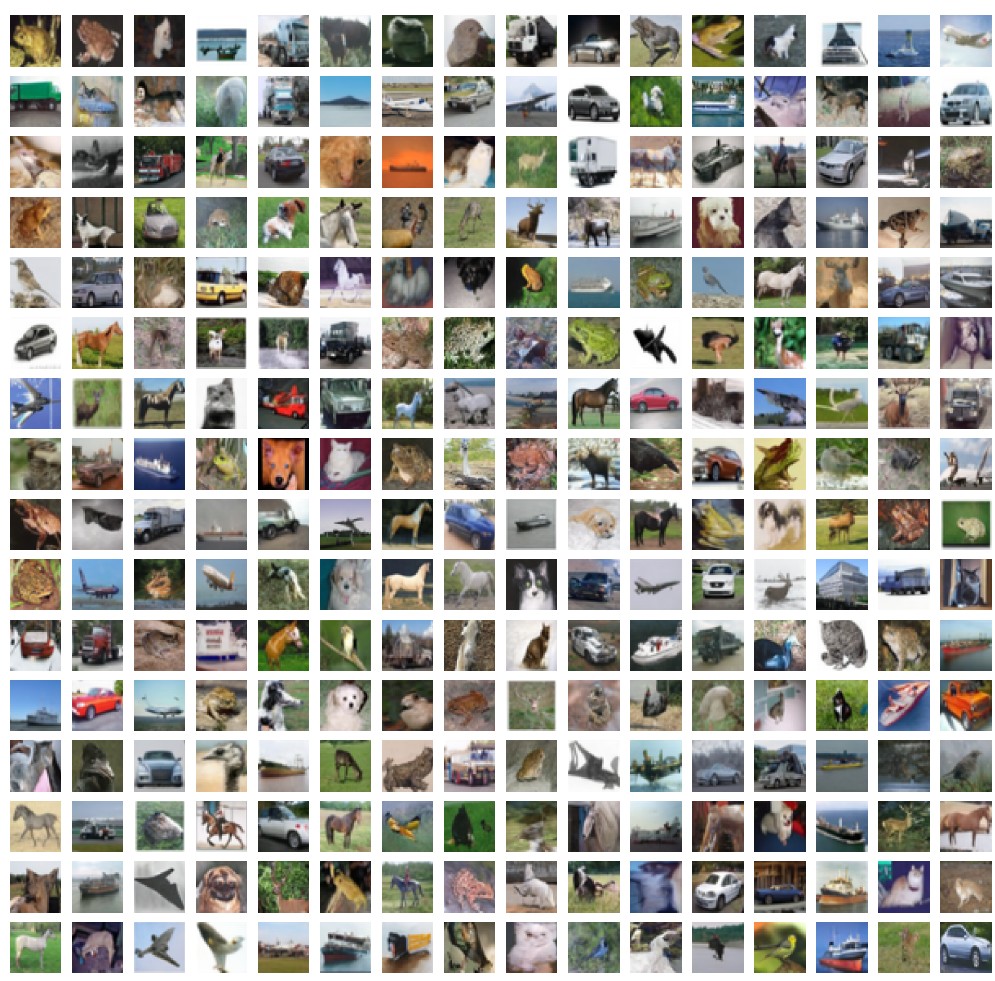

Figure 8: Unconditional CIFAR10 samples from our $\tau$LDR-10 model.

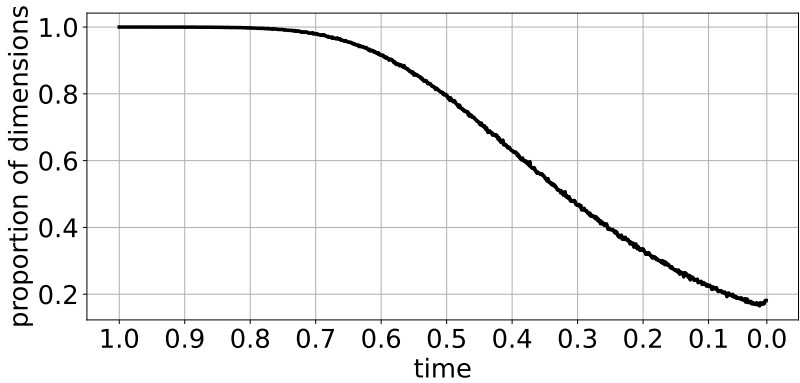

Figure 9: Proportion of dimensions that transition during a single step of tau-leaping during the reverse sampling process.

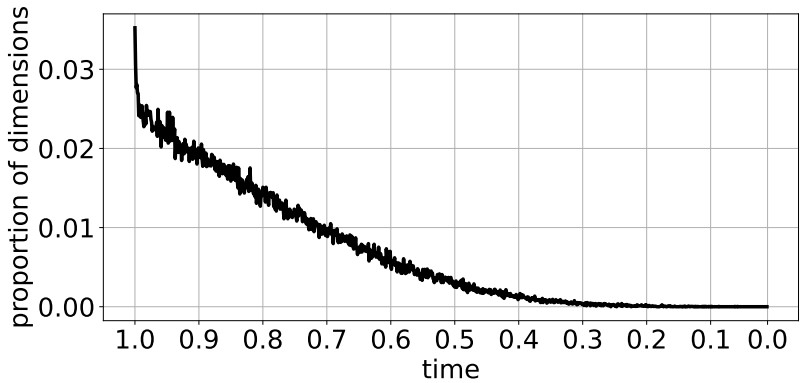

Figure 10: Proportion of dimensions that are clipped during a tau-leaping step due to proposing an out of bounds jump during the reverse sampling process.

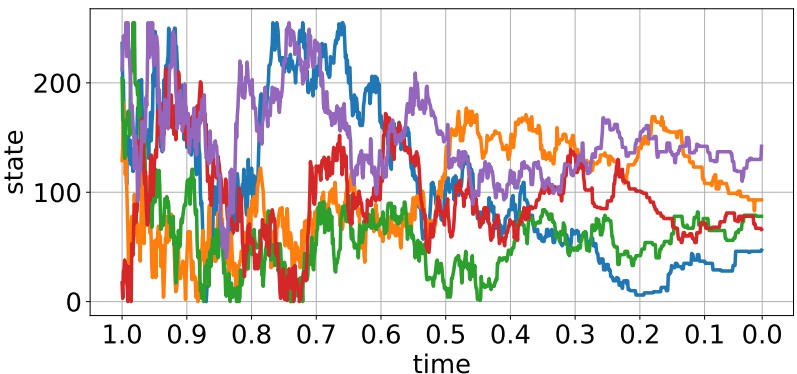

Figure 11: The progression of a selection of dimensions during the reverse sampling process.

since the mapping to the integers was arbitrary. The uniform rate is suitable for this categorical data because, during a time interval, it has a uniform probability to transition to any other state. The D3PM baseline was implemented also with a time homogeneous uniform forward kernel set such that the rate of noising is matched in the discrete and continuous time cases.

We define our conditional denoising network, $p_{0|t}^{\theta}(x_0|x, y)$ using a transformer architecture inspired by [31]. It takes an input of shape $(B, D, S)$ where $B$ is the batch size, $D$ is the dimensionality (256) and $S$ is the state size (129). This final dimension contains the one-hot vectors. The conditioning on the initial bars is achieved by concatenating the conditioning information $y$ with the noisy input $x$. At the start of the network, there is an input embedding linear layer with output of size 128 which is our model dimension for the transformer. Then a transformer positional embedding is added to the hidden state. Next a stack of 6 transformer encoder layers are applied which consist of a self attention block and a one hidden layer MLP. The self attention block uses 8 heads and the MLP has a hidden layer size of 2048. At the output of each internal block, we apply dropout with rate 0.1. Finally, there is a stack of 2 residual MLP layers. Each consists of a one hidden layer MLP with a hidden dimension of 2048. There is a residual connection between the input and output of the MLP. A layer norm is applied to the output of the block. To create the output of the network, there is an output linear layer with an output shape of $(B, D, S)$ where now the $S$ dimension has logit probabilities. To instill a residual bias into the network, we add the one-hot input to the output logits. All activations are ReLU. The time is input into the network through FiLM layers [35]. First, the time is embedded using the sinusoidal transformer position embedding as in the U-net architecture used for image modeling to create an embedding size of 128. This is then passed into a single hidden layer MLP with hidden size

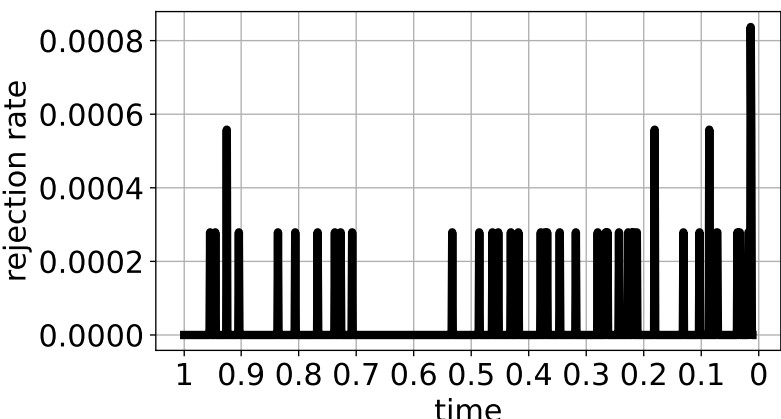

Figure 12: Proportion of jumps rejected during reverse sampling. The rejection rate is calculated as the proportion of dimensions in a tau leaping step that have their jump rejected. The results are averaged over a batch of 16.

2048 and output size 512. Within each encoder and residual MLP block, there is a FiLM linear layer which takes in the 512 time embedding and outputs two FiLM parameters each of size 128. These are the scale and offset applied to the hidden state. In the encoder blocks, this FiLM transform is applied after the self attention block and again after the fully connected block. In the residual MLP blocks, it is applied after the layer norm operation. Our network has approximately 7 million parameters in total.

We optimize using Adam for 1M steps with a batch size of 64 and learning rate of 0.0002. We use the conditional $\bar{\mathcal{L}}_{CT}$ objective with additional direct $p_{0|t}^\theta$ supervision as described in Appendix D with weight $\lambda = 0.001$. We also make the same one forward pass approximation as explained in Appendix C.4. We use the standard set of training tricks to improve optimization [3, 4]. Throughout training we maintain an exponential moving average of the parameters with decay factor 0.9999. These average parameters are used during testing. At the start of optimization we use a linear learning rate warm-up for the first 5000 steps. We clip the gradient norm at a norm value of 1.0. We train on a single V100 GPU on an academic cluster.

For sampling with $\tau$LDR-0 we use $\tau = 0.001$ and for sampling with $\tau$LDR-2 we include 2 corrector steps per predictor step after $t < 0.9T$. The corrector rate is simulated with $\tau = 0.0001$ which we found to perform best. We reject any dimension in which 2 or more jumps are proposed as this is categorical data. We plot the rejection rate in Figure 12. Most of the time, the rejection rate is zero and there are few steps for which it increases slightly. We show a large batch of samples from the first 10 songs in the test dataset in Figure 13. We see that there is variation between the sampled completions and they consistently follow the style of the conditioning first two bars of the song. Audio samples from the model are available at `https://github.com/andrew-cr/tauLDR`. Finally, we examine the progression of a random selection of dimensions during reverse sampling for the uniform and birth/death rate matrix cases. Figure 14 shows the progression for the uniform case, we see that large jumps through the state space are made throughout the reverse process. Figure 15 shows the progression for the birth/death case. At the start of reverse sampling, no dimensions move as the rejection rate is high in this case because the rate matrix is not suitable for categorical data. Nearer the end, small jumps are made between adjacent states but since large jumps between any category do not occur for this rate matrix, the performance will overall be worse.

## I   Ethical Considerations

Our work increases our theoretical understanding of denoising generative models and also improves generation capabilities within some discrete datasets. Deep generative models are generic methods

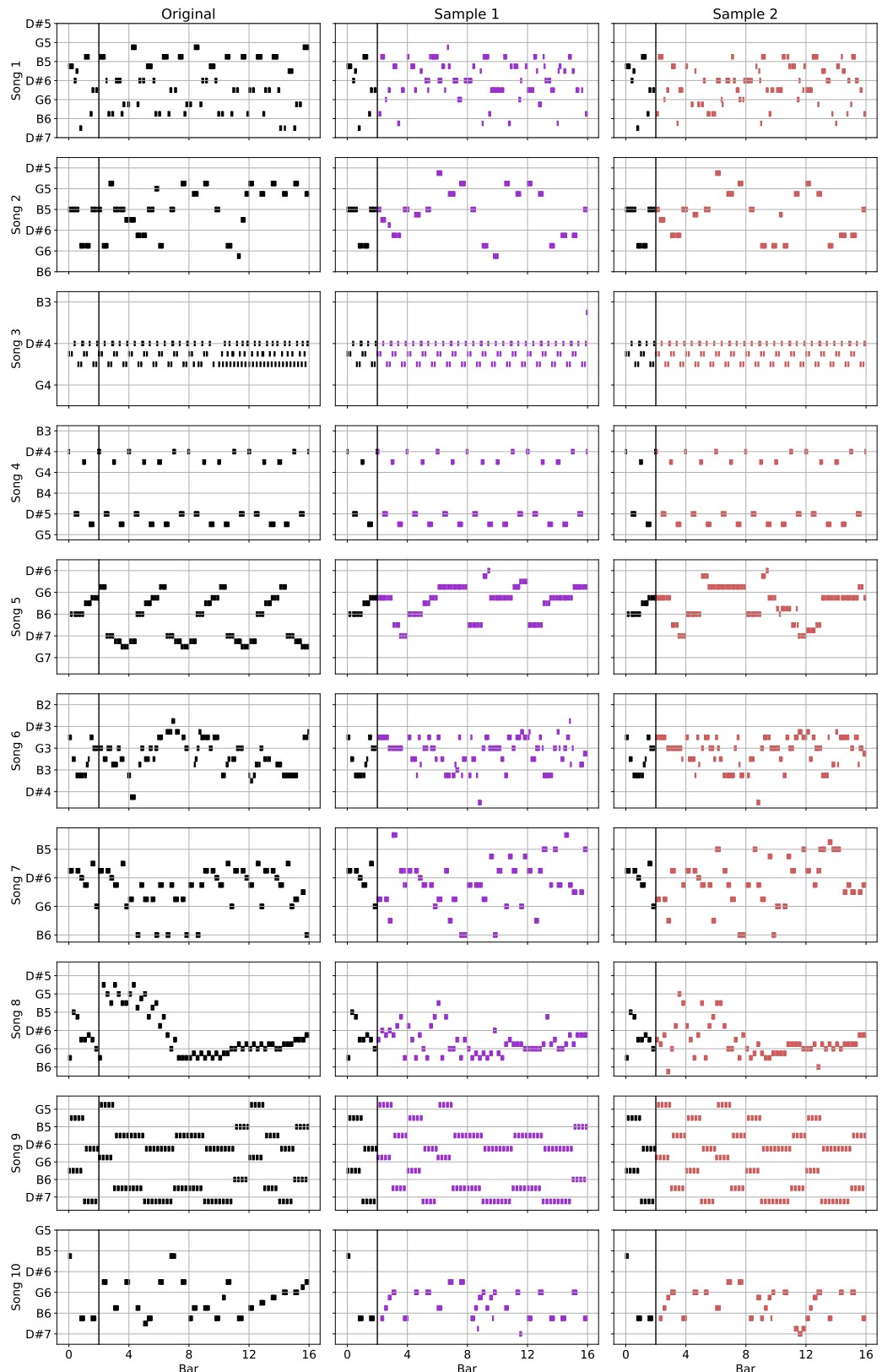

Figure 13: Two conditional samples from the $\tau$LDR-0 model for each of the first 10 songs in the test dataset.

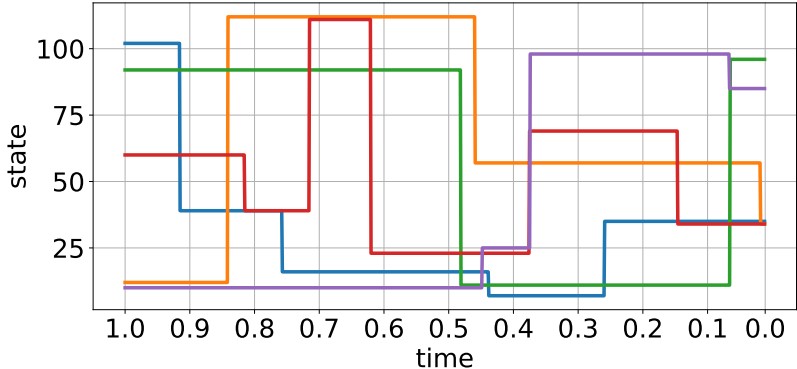

Figure 14: The progression of a selection of dimensions during the reverse sampling process for the uniform rate matrix.

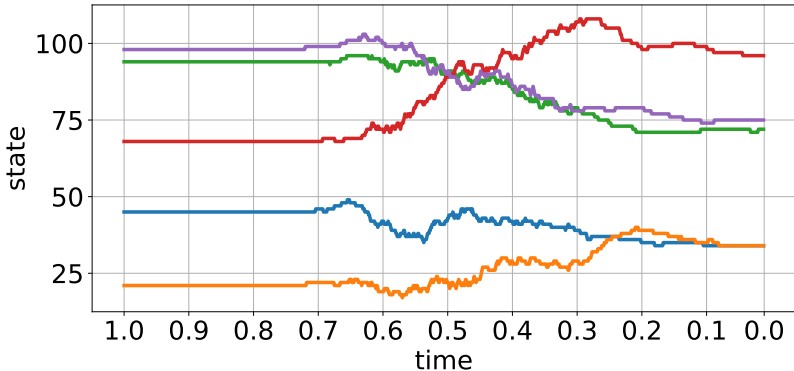

Figure 15: The progression of a selection of dimensions during the reverse sampling process for the birth/death rate matrix.

for learning from unstructured data and can have negative social impacts when misused. For example, they can be used to spread misinformation by reducing the resources required to create realistic fake content. Furthermore, generative models will produce samples that accurately reflect the statistics of their training dataset. Therefore, if samples from these models are interpreted as an objective truth without fully considering the biases present in the original data, then they can perpetuate discrimination against minority groups.

In this work, we train on datasets that contain less sensitive data such as pictures of objects and music samples. The methods we presented, however, could be used to model images of people or text from the internet which will contain biases and potentially harmful content that the model will then learn from and reproduce. Great care must be taken when training these models on real world datasets and when deploying them so as to mitigate and prevent the harms that they can cause.