# OpenReview forum: "A Continuous Time Framework for Discrete Denoising Models"
_NeurIPS.cc/2022/Conference — NeurIPS 2022 Accept_

### Official Review · Reviewer_Lc6p · 2022-07-02

**Rating:** 7
**Confidence:** 4
**Soundness:** 3 good
**Presentation:** 3 good
**Contribution:** 3 good

**Summary:**

The paper proposes a continuous-time (CT) diffusion model for discrete data. In contrast to discrete-time (DT) diffusion models for discrete data, the CT diffusion model replaces the discrete time forward kernel with a transition rate matrix. It is then shown that that there exist a reverse transition matrix which can be used for generation. A CT ELBO to learn said reverse transition matrix is derived. The authors describe different choices of transition rate matrices than can be used. For high-dimensional data, the memory requirement for the transition matrices becomes large, and therefore the authors discuss a factorization technique. The paper proposes to sample from the CT diffusion model via tau-leaping, a well-known technique in chemical physics. The CT framework allows for bounds on the discrepancy between the true data distribution and the sampling distribution induced by tau-leaping. The author test their method on various experiments, outperforming existing DT diffusion models on discrete data.

**Questions:**

* Experiment 6.1: How many NFEs it the model using for the different values of $\tau$? I expect $\tau=0.004$ needing to use significantly more than say $\tau=0.1$.
* Experiment 6.2: Are all baseline methods in Table 1 using 1000 NFEs? How well does the model perform when you use significantly less than 1000 NFEs, say 50?
* Experiment 6.3: What's the intuition that your method outperforms D3PM on CIFAR-10 by a large margin but only slightly (correct me if I am wrong) on the Monophonic Music dataset?  Why do you compare to different versions of D3PM (absorbing/gaussian vs uniform) on the two modalities?

**Strengths And Weaknesses:**

Strengths:
* The proposed method is novel.
* Using tau-leaping to reduce the NFEs needed for generation is clever.
* The paper is very detailed and the appendix is complete (as far as I can tell)
* I like that the paper shows experiments over two modalities. (On that note, I would have been really interested to also see how the method performs on text data. This is just a personal preference and won't influence my review/rating.)

Weaknesses:
* There are some interesting ablations missing, for example, on the choice of $R_b$, the impact of factorizing over dimensions, and the training approximation described in C.4. I think these experiment would be very valuable.
* Some details/explanations in the experiment section are missing. See Qs below.
* The method performs very well when many corrector steps are used but what's the intuition behind that? For continuous diffusion models it seems that one (or no) corrector step(s) is sufficient. This should have been discussed/investigated.

Suggestions:
* The training approximation in C.4 is interesting/important and I would have liked to see some discussion/intuition in the main paper and not just the appendix.
* I think the tau-leaping idea is very cool and breaks the stigma that discrete diffusion models are inherently slow at inference time. Given the success of accelerating inference for continuous diffusion models, I think it would be very valuable to have more experiments/investigations/outlooks how to even further improve acceleration? Maybe adapt tau-leaping towards the special use case of sampling from discrete diffusion?

All in all, I think the paper propose a very interesting method (as well as very good results). I intend to raise my score if weaknesses/questions are sufficiently addressed.

---

> ### Author Response · Authors · 2022-08-02
> **Response to Reviewer Lc6p (Part 1)**
>
> We thank the reviewer for their engagement with our proposed methodology and helpful suggestions. We are grateful that our work is considered novel and provided in good detail. We answer questions and respond to comments below.
>
> > ***"There are some interesting ablations missing, for example, on the choice of $R_b$, the impact of factorizing over dimensions, and the training approximation described in C.4. I think these experiment would be very valuable.''***
>
> > ***"The training approximation in C.4 is interesting/important and I would have liked to see some discussion/intuition in the main paper and not just the appendix.''***
>
> Thank you for highlighting questions regarding the training approximation, we agree that having more discussion on it as well as experimental results will improve the description of our objective. We will add the intuition as to its validity in the main text as well as our new results with the extra space of a camera ready revision if accepted. Expanding on this here, the general idea behind the approximation is that to evaluate the CT ELBO objective naively requires us to input both $x^{1:D}$ and $\\tilde{x}^{1:D}$ into the denoising network even though $\\tilde{x}^{1:D}$ differs to $x^{1:D}$ in only a single dimension. For example, in the CIFAR10 experiment, that is to say $\\tilde{x}^{1:D}$ is generated from $x^{1:D}$ by picking a single channel in a single pixel and perturbing its value slightly. Since the denoising network will treat these two inputs as largely identical, we approximate the output for $x^{1:D}$ with the value output for $\\tilde{x}^{1:D}$, cutting training time in half.
> We perform an extra experiment to verify the approximation's validity on the music dataset. We train 6 separate models, 3 using the one forward pass objective and 3 using the two forward pass objective. We calculate the mean and standard deviations for the evaluation metrics across the 3 runs for each objective. We find that there is no significant difference in their performance after 1M training iterations.
>
>  | |Hellinger Distance | Proportion of Outliers |
> |---|-------|---|
>  | One forward pass | $0.378 \\pm 0.002$ | $0.112 \\pm 0.003$ |
> | Two forward pass | $0.379 \\pm 0.001$ | $0.114 \\pm 0.003$ |
>
>
> Regarding the choice of the rate matrix, $R_b$, we find that it is important to choose a noising process appropriate for the type of discrete data used. We ablate on $R_b$ for the one-hot monophonic music dataset because it is instructive to see that a rate assuming ordinal structure (the Birth/Death rate) performs worse than the more suitable uniform rate matrix. When modelling CIFAR10 as discrete data, Austin et al. (2021) have already performed an extensive investigation into a range of corruption matrices and found uniform rate matrices perform poorly whilst Gaussian type rate matrices perform best. Our early preliminary experiments found the same result with uniform rate matrices giving an FID of $\\sim 30$ and so we decided to build on their best performing corruption matrices but converted into continuous time versions.
>
>
> With respect to the factorization over dimensions, we would like to note that all other diffusion models (continuous e.g. DDPM and NCSN or discrete e.g. D3PM) factorize the forward process over dimensions. This is necessary for all models because otherwise extra dependencies would be introduced into the reverse process that would require some form of autoregressive model to parameterize. That would be a whole new model class. Furthermore, in our specific case, the factorization of the forward process is fundamental to making our method tractable at all on realistic problems because otherwise we would need to store the full $\\mathbb{R}^{S^D \times S^D}$ rate matrix which is astronomically large for real datasets, e.g. in CIFAR10 it is of size $\\mathbb{R}^{256^{3072} \times 256^{3072}}$.

---

> > ### Author Response · Authors · 2022-08-02
> > **Response to Reviewer Lc6p (Part 2)**
> >
> > > ***"The method performs very well when many corrector steps are used but what's the intuition behind that? For continuous diffusion models it seems that one (or no) corrector step(s) is sufficient. This should have been discussed/investigated.''***
> >
> > This question raises some very interesting points highlighting the fundamental differences between a corrector scheme in continuous space versus one in discrete space. I will here summarize and compare the corrector schemes in both cases and then provide some hypotheses explaining this phenomenon.
> >
> > In a continuous state space, the predictor and corrector step are similar apart from the extra $\\sqrt{2}$ scaling on the Gaussian noise addition versus the scale of the score. Specifically, looking at Algorithm 2 in Song et al. (2021)
> > \\begin{equation}
> >     \\text{Predictor} \\quad x_i \\leftarrow x_{i+1} + (\\sigma_{i+1}^2 - \\sigma_i^2) s_\\theta(x_{i+1}, \\sigma_{i+1}) + \\sqrt{\\sigma_{i+1}^2 - \\sigma_i^2} z, \\quad z \\sim \\mathcal{N}(0, I)
> > \\end{equation}
> > \\begin{equation}
> >     \\text{Corrector} \\quad x_i \\leftarrow x_i + \\epsilon_i s_\\theta(x_i, \\sigma_i) + \\sqrt{2 \\epsilon_i} z, \\quad z \\sim \\mathcal{N}(0, I)
> > \\end{equation}
> >
> > What we can take away from this is that the corrector step in continuous space tries to perform exploration of the $p_t(x)$ marginal through injecting extra Gaussian noise into the updates.
> >
> > In discrete state space, for a predictor step we simulate according to the reverse rate matrix $\\hat{R}_t^\\theta$ and for a corrector step we simulate according to a `corrector rate matrix' $R^{c, \\theta}_t = \\hat{R}_t^\\theta + R_t$. Exact simulation involves sampling a categorical distribution based on a normalized version of the rate. Therefore, the corrector step in discrete state space can then be thought of as exploring the $p_t(x)$ marginal through a slightly noisier version of the categorical distribution corresponding to $\\hat{R}_t^\\theta$. Another distinction is that in the continuous space case, the denoising model outputs a point estimate for $x_0$/score (output dim of $D$) whereas in discrete state spaces, the denoising model outputs a conditionally independent categorical distribution in each dimension (output dim of $S \times D$). This allows the discrete denoising model some form of uncertainty expression in its $x_0$ prediction (albeit still with conditionally independent dimensions).
> >
> > In light of these differences, our hypothesis for why the discrete data model benefits from more corrector steps, is that the discrete data framework can better utilize them to explore the $p_t(x)$ marginal. Continuous state space corrector steps explore via adding Gaussian noise with the score giving a point wise directional bias, but it has been found that having too many corrector steps can make the resulting generations noisy see e.g. Furusawa et al. (2021) Fig 9 and Song & Ermon (2019) which can be seen as only having corrector steps and results in poorer image generation. On the other hand, in discrete state space, the noise introduced is based solely on sampling a categorical distribution that is largely defined through the denoising model and $\\hat{R}_t^\\theta$. The corrector steps also give a chance for multi-modal $x_0$ prediction information to mix between conditionally independent dimensions  potentially exploring separate modes of $p_t(x)$ which is an effect completely absent from continuous state space models. We investigate this by following our standard reverse tau-leaping sampling until $t=0.4$ and then applying very many corrector steps whilst holding time constant (see the new experiment in Section F.2 in the Appendix). We find that the $x_0$ predictions cycle through many different possibilities corresponding to exploring modes of $p_t(x)$.
> >
> > It would be an exciting piece of further work to fully investigate the similarities and differences between the predictor-corrector procedure in continuous and discrete state spaces now we have shown the equivalence using our continuous time framework.

---

> > > ### Author Response · Authors · 2022-08-02
> > > **Response to Reviewer Lc6p (Part 3)**
> > >
> > > > ***"I think the tau-leaping idea is very cool and breaks the stigma that discrete diffusion models are inherently slow at inference time. Given the success of accelerating inference for continuous diffusion models, I think it would be very valuable to have more experiments/investigations/outlooks how to even further improve acceleration? Maybe adapt tau-leaping towards the special use case of sampling from discrete diffusion?''***
> > >
> > > This is an exciting avenue for further research and we hope that our continuous time perspective can provide additional tools for speeding up discrete denoising models just as it has done in the continuous state space case. As for outlooks on the possible future avenues to be explored, we agree that adapting the tau-leaping algorithm for our specific use-case here is a good basis for more research. We hope that through showing the link with the chemical physics field, we can invite expertise in CTMC simulation to be applied to this problem. We admit that we are not experts in state of the art CTMC simulation techniques but some themes that could be explored are: using advanced tau selection methods to take larger jumps or using more exotic forms of CTMC integrator. In this work, we focused on augmenting our approach with corrector steps to improve sample quality and demonstrate discrete denoising models can become a viable model for discrete datasets. The corrector method also helps us demonstrate an immediate benefit of the continuous time interpretation as this is what allowed us to very simply derive our corrector rate.
> > >
> > > > ***Experiment 6.1: How many NFEs it the model using for the different values of $\tau$? I expect $\tau=0.004$ needing to use significantly more than say $\\tau=0.1$.''***
> > >
> > > Since our continuous time process is defined from $t=0$ to $t=1$, the number of denoising model evaluations is simply $1/\\tau$. So for $\\tau = 0.004$, there are $250$ NFEs and for $\\tau=0.1$ there are $10$ NFEs. We have made this clearer in an update to the paper, thanks!
> > >
> > > > ***"Experiment 6.2: Are all baseline methods in Table 1 using 1000 NFEs? How well does the model perform when you use significantly less than 1000 NFEs, say 50?''***
> > >
> > > Yes, all methods in Table 1 are using 1000 NFEs in the reverse sampling process for fair comparison. We investigate performance for low NFE values in Figure 2, performance does degrade for very low NFE numbers, extrapolating the curves, we would expect poor performance at 50 NFEs. This is expected for an initial diffusion framework, e.g. Lu et al. (2022)  Figure 2 shows DDPM for low NFE values and shows a similar degradation in performance. We have shown that adding corrector steps can push the pareto optimal frontier down for higher NFE values, we hope that with further work the frontier could be pushed further down and to the left.
> > >
> > > > ***"Experiment 6.3: What's the intuition that your method outperforms D3PM on CIFAR-10 by a large margin but only slightly (correct me if I am wrong) on the Monophonic Music dataset? Why do you compare to different versions of D3PM (absorbing/gaussian vs uniform) on the two modalities?''***
> > >
> > > The intuition is that the CIFAR-10 dataset is more complex than the music dataset because it is higher dimensional, has an increased state space cardinality and likely has more complex structure/correlations within the datapoints themselves. Therefore, the benefits that our method brings for improving sample quality are less visible in the absolute changes in the metrics themselves. We also note that the scalings and reference points for the metrics between the images and the music dataset are very different so the specific sizes of absolute changes in metrics are less meaningful.
> > >
> > > We use the most competitive form of D3PM for each experiment as our baseline for fair comparison, thank you for raising this point we have made this clearer in the paper. On the image dataset, it was found by Austin et al. (2021) that the Gaussian corruption matrix performs the best so we include that. On the music dataset, since the data is categorical, the Gaussian corruption matrix would perform poorly because it assumes ordinal structure. The appropriate corruption process here is the uniform corruption process which we use for both D3PM and our method. We include the birth/death ablation for our method to demonstrate that corruption processes that assume ordinal structure are not suitable here.
> > >
> > > **References**
> > >
> > > Austin et al., "Structured Denoising Diffusion Models in Discrete State-Spaces", NeurIPS 2021
> > >
> > > Song et al., "Score-Based Generative Modeling through Stochastic Differential Equations", ICLR 2021
> > >
> > > Furusawa et al., "Generative Probabilistic Image Colorization", preprint 2021
> > >
> > > Song & Ermon, "Generative Modeling by Estimating Gradients of the Data Distribution", NeurIPS 2019
> > >
> > > Lu et al., "DPM-Solver: A Fast ODE Solver for Diffusion Probabilistic Model Sampling in Around 10 Steps", preprint 2022

---

> > > > ### Comment · Reviewer_Lc6p · 2022-08-07
> > > > **Thank you for detailed response**
> > > >
> > > > I thank the authors for their detailed response. All of my questions have been answered and I raised my score accordingly.

---

### Official Review · Reviewer_JFfK · 2022-07-08

**Rating:** 7
**Confidence:** 4
**Soundness:** 3 good
**Presentation:** 3 good
**Contribution:** 3 good

**Summary:**

This paper study diffusion models with discrete states. This paper considers continuous time by introducing the transition rate matrix. A relationship between the denoiser q(x0|xt) and the reverse transition rate matrix is built, and the author uses a denoising model p(x0|xt) to approximate q(x0|xt). The author further proposes a continuous time ELBO, which can be used to train p(x0|xt). The author also applies the Tau-Leaping for sampling from the diffusion model, which can be faster than the exact simulation. Experiments show the effectiveness of this method.


**Questions:**

1. It seems directly applying the discrete time diffusion model to high dimensional data needs to model a complex $q_{0|k}(x_0^{1:D}|x_k^{1:D})$, according to $q(x_{k-1}^{1:D}|x_k^{1:D}) = E_{q(x_0^{1:D}|x_k^{1:D})} q(x_{k-1}^{1:D}|x_k^{1:D}, x_0^{1:D})$. However, according to Proposition 3, this work only needs to model a one-dimensional $q_{0|t}(x_0^d|x_t^{1:D})$. What is the intristic reason between this difference? This is important to further understand this work.

2. When discussing improving sampling speed in Section 7, [1] can be cited, which is also an important work on speeding up diffusion models.

[1] Bao et al., Analytic-DPM: an Analytic Estimate of the Optimal Reverse Variance in Diffusion Probabilistic Models.

**Limitations:**

The authors adequately addressed the limitations and potential negative societal impact of their work.

**Strengths And Weaknesses:**

Strengths:
1. This paper develops a continuous time framework for discrete data. This framework includes optimum analysis, training, sampling and error analysis, which are nice. This paper also provides sufficient math pre knowledge to understand this framework, which makes this work more readable.

Weakness:
1. **Relationship to prior works can be discussed more.** Prior to this work, there are various works on discrete state diffusion models, e.g., as mentioned by the author, uniform kernel and absorbing kernel. The relationship to these works can be discussed more, for example from these aspects: (a) What is the corresponding transition rate matrix of these prior methods; (b) Will the reverse process of prior discrete time methods (between Line 70 and Line 71)  converge to the continuous time reverse process proposed in this paper? (c) Is the sampling method in the discrete time case applicable to the continuous time case? How about comparing Tau-Leaping and discrete time sampling?

2. **More ablation study.** This paper provides various techniques for continuous time diffusion models. For example, discrete time ELBO -> continuous time ELBO, discrete time sampling -> Tau-Leaping sampling. Which one makes the greatest contribution to the performance?

---------------------

While there is still a (relatively small) performance gap between this work and continuous diffusion models, I think it is an important work due to its technical contributions. I'd like to further raise my score if my concerns are addressed.

---

> ### Author Response · Authors · 2022-08-02
> **Response to Reviewer JFfK (Part 1)**
>
> We thank the reviewer for their review and interesting questions. We appreciate the reviewer's praise of our methodological framework and the paper's clear readability. The reviewer raises some very good points regarding the correspondence between discrete time and continuous time objects. We have made these links clearer in an update to Section 4.1 and Appendix E. We comment specifically here in more detail on each of these correspondence questions as well as other comments raised in the review.
>
> > ***"a) What is the corresponding transition rate matrix of these prior methods''***
>
> For some simple time homogeneous cases, the relationship between the transition rate matrix and the corresponding discrete time kernel is quite clear. For example, if in discrete time we have the uniform kernel, $P = \\alpha \\mathbf{1}\\mathbf{1}^T + (1 - S \\alpha) I$ where $\\mathbf{1}$ is a vector of ones and $I$ is the identity, then the corresponding continuous time transition rate matrix is $R = \\beta \\mathbf{1} \\mathbf{1}^T - \\beta S I$ with $\\beta$ depending on what time discretization is used. Similarly, if in discrete time we have an absorbing state kernel, $P = \\alpha \\mathbf{1} \\mathbf{e}_\\ast^T + (1-\\alpha) I$ where $\\mathbf{e}_\\ast$ is the one-hot encoding of the absorbing state, then the corresponding transition rate matrix is $R = \\beta \\mathbf{1} \\mathbf{e}_\\ast^T - \beta I$. For the Gaussian kernel defined by Austin et al. (2021) the relationship is less clear. However, if we remain in the time homogeneous case, then it could be numerically calculated by finding the matrix logarithm of the discrete time kernel.
>
> Things become much more difficult when we move to the time inhomogeneous case if we would like to find some continuous time transition rate matrix that results in discrete time kernels that interpolate between kernels at known time points. This would amount to solving the Forward Kolmogorov matrix differential equation $\\partial_t P_{t|s} = P_{t|s} R_t$ for $R_t$ given a desired interpolation scheme for the discrete time kernel from time $s$ to time $t$, $P_{t|s}$. If you were to assume that the corresponding rate $R_t$ commutes for any two times $t$, $t'$ (which may not be known to be true if you only know $P_t$) then you could have $P_t = \\text{exp} \\left( \\int_0^t R_s ds \\right)$ (see Appendix E). However, this still doesn't get you all the way as you would then need to analytically calculate the matrix logarithm of $P_t$ in order to choose an $R_s$ that integrates to the desired value which would be very difficult for complex discrete time kernels e.g. the Gaussian kernel.
>
> > ***"(b) Will the reverse process of prior discrete time methods (between Line 70 and Line 71) converge to the continuous time reverse process proposed in this paper?''***
>
> Assuming we start with a discrete time noising process where the corresponding matrix embedding problem is solvable, (i.e we can find a rate matrix $R_t$ such that the transition probabilities associated to $R_t$ in each forward time step $t_k$ to $t_{k+1}$ correspond to the transition probabilities used by the discrete time method), then the discrete time process has the same distribution as a CTMC with rate matrix $R_t$ sampled at the discrete set of times $t_0, t_1, \\dots, t_K$. Consequently, learning the reverse process in discrete time will correspond to learning a sub-sampled version of the continuous time reverse process. In the limit where the step size $\\tau$ is taken to be very small, the reverse process of the discrete time methods will thus converge to the continuous time reverse process.
>
> > ***"(c) Is the sampling method in the discrete time case applicable to the continuous time case? How about comparing Tau-Leaping and discrete time sampling?''***
>
> This observation is very important in giving the intuition behind precisely why we need to use tau-leaping for sampling. Thank you for raising this point, including this discussion will improve the clarity of the tau-leaping section which we plan to update for the camera-ready if accepted.
>
> One could, in theory, use the discrete time sampling method and apply this to the continuous case. However, in order to calculate the transition probabilities of the reverse process between times $t_k$ and $t_{k-1}$ which the discrete method requires, one would have to either integrate the reverse process directly between these times, or calculate the exponential of a matrix with dimensions $S^D \times S^D$. Both of these are computationally infeasible for the datasets which we study; this difficulty arises because the discrete time transition probabilities no longer factorize over the dimensions of the state space, unlike our rate matrix. The tau-leaping algorithm allows us to sidestep these computational challenges by exploiting the special structure induced into the reverse rate matrix through the forward process factorization in continuous time.

---

> > ### Author Response · Authors · 2022-08-02
> > **Response to Reviewer JFfK (Part 2)**
> >
> > > ***"More ablation study. This paper provides various techniques for continuous time diffusion models. For example, discrete time ELBO -> continuous time ELBO, discrete time sampling -> Tau-Leaping sampling. Which one makes the greatest contribution to the performance?''***
> >
> > It is interesting to think about the separate parts of our proposed continuous time framework in isolation, however, they would not, by themselves, give an ablation study that provides the intuition we are looking for. That is because the CT framework requires both the objective and sampling procedure to be novel. First examining the objective, the continuous time ELBO gives us a way to learn the generative reverse rate but since the rate formulation is specific to continuous time, we would not be able to learn this rate directly using the discrete time ELBO. In theory, we could train a denoising model using the discrete time ELBO and then plug that into a reverse rate formulation, however, it would still be an inherently discrete time object, only trained to denoise at a finite selection of time points. Further, if we were to use it anyway in our continuous time sampling procedure, this would no longer match the generative procedure that the ELBO objective is built on since we have been training the denoiser to maximize likelihood with a different generative process in mind. Overall, this would mean that training with the discrete time objective but sampling with tau-leaping wouldn't fully elucidate the benefit of tau-leaping since we have introduced a number of other incompatibilities that would reduce performance and we wouldn't gain the intuition we are looking for.
> >
> > Conversely, the contrasting ablation study, where we train with the continuous time objective and sample using a discrete time sampler wouldn't be possible due to the impracticality of calculating the matrix exponential of the learned reverse rate matrix as we described previously in point (c).
> >
> > Taken together, our continuous time objective and continuous time sampler, forms a direct continuous time analogue of the discrete time method and we find they perform similarly. What we find really helps performance is the addition of corrector steps that we derived using the continuous time framework. We ablate on these steps in the image and music experiments, finding they can improve performance significantly.
> >
> > > ***"It seems directly applying the discrete time diffusion model to high dimensional data needs to model a complex $q_{0|k}(x_0^{1:D} | x_k^{1:D})$ ... ''***
> >
> > This is a very good point in understanding the difference between discrete and continuous time models (for the full details of the following argument, please refer to Appendix G in the supplement).
> >
> > For discrete time models, they also use a conditionally independent denoising model over $x_0$, i.e. $p_{0|k}^\theta (x_0^d | x_k^{1:D})$. The reason they are also able to do this is because the forward noising process in discrete time is also factorized across dimensions just as in our work. However, an important subtlety is that to accurately model the full dimensional true reverse kernel in discrete time, $q_{k-1|k}(x_{k-1}^{1:D} | x_k^{1:D})$, the true denoising model that actually needs to be approximated is $q_{0|k}(x_0^d | x_{k-1}^{1:d-1}, x_k^{1:D})$ which would in theory require some form of autoregressive model to parameterize faithfully. This would be expensive, so an approximation is made in discrete time to model the true $q_{0|k}(x_0^d | x_{k-1}^{1:d-1}, x_k^{1:D})$ with $p_{0|k}^\theta (x_0^d | x_k^{1:D})$. This approximation is more accurate with an increased number of steps in the noising process. Since we operate in the continuous time limit, effectively with an infinite number of steps in the noising process, we don't have to make this approximation, and the true denoising model to approximate is $q_{0|t}(x_0^d | x_t^{1:D})$ as you have mentioned.
> >
> > > ***"When discussing improving sampling speed in Section 7, [1] can be cited, which is also an important work on speeding up diffusion models''***
> >
> > We are happy to add this citation in an update to Section 7, thank you for making us aware of this work.
> >
> >
> > **References**
> >
> > Austin et al,. "Structured Denoising Diffusion Models in Discrete State-Spaces" NeurIPS 2021

---

> > > ### Comment · Reviewer_JFfK · 2022-08-07
> > > **Update**
> > >
> > > Thanks for the clear response. I have raised my score.

---

### Official Review · Reviewer_yvGD · 2022-07-12

**Rating:** 7
**Confidence:** 4
**Soundness:** 4 excellent
**Presentation:** 2 fair
**Contribution:** 3 good

**Summary:**

This paper proposed to extend the continuous-time diffusion/score-based modeling idea (Song et al., 2021) to discrete state space. In discrete state space, there is no SDEs therefore the forward process needs to be replaced by a discrete-state CTMC, which is characterized by the infinitesimal generator (transition rate matrix). Effective parameterizing and optimizing such models are nontrivial and the authors introduce several innovations to tackle these problems, including a continuous time lower bound for discrete state space, computable representations of the rate matrix, and a way to trade off sample speed and quality. Experiment results show it outperforms prior discrete diffusion models despite still lag behind continuous diffusion models.

**Questions:**

* Could the authors comment on possible reasons why discrete state modeling is still outperformed by their continuous counterpart? Also related to the limitation question below.

**Limitations:**

The authors discussed some limitations but I am additionally interested in the above question --- what limits the model to perform as well as continuous state diffusions on discrete image data? Is it because of the flexibility of generator parameterization? or the tradeoff caused by computational cost?

**Strengths And Weaknesses:**

This is a nice paper and I have no much complain about it. Particular strong points are
* Principled characterization of discrete CTMC via the generator (transition rate) matrix
* A well-thought way to represent the exponentially large rate matrix that is both flexible and computationally cheap.
* The observation that the sum of forward and backward generator has the marginal distribution at that time as stationary (Proposition 4), which enables the adoption of the predictor-corrector sampling proved effective by Song et al. (2021) in continuous diffusions.

The only suggestion I have is to improve the readability of section 4.3 by first explaining the canonical way of sampling from a discrete-state CTMC using competing exponential clock (I haven't checked but I suspect this corresponds to the Gillespie's algorithm). The tau-leaping idea could be explained in more detail in appendix and an algorithm box in main text would definitely help.

---

> ### Author Response · Authors · 2022-08-01
> **Response to Reviewer yvGD**
>
> We would like to thank the reviewer for their review and positive comments. We are pleased to hear that the reviewer appreciates our characterization of the CTMC and our derivation of the predictor-corrector sampler in discrete spaces. We address the suggestions and questions below.
>
> > ***"The only suggestion I have is to improve the readability of section 4.3 by first explaining the canonical way of sampling from a discrete-state CTMC using competing exponential clock''***
>
> Thank you for highlighting this, we agree that it would improve readability to first introduce standard CTMC simulation. Unfortunately, we had to cut this introduction in the submitted draft as we did not have enough space in the main text, we hope to be able to add this back in with the extra space in a camera ready revision if accepted.
>
> You are correct that the competing exponential clock method is indeed Gillespie's algorithm. The comparison would also be helpful in understanding why tau-leaping is useful here: we do not need to consider each transition individually which would be necessary for Gillespie's algorithm. Given the extra space, we agree an algorithm box would also be very helpful for tau leaping sampling given that many people will be unfamiliar with it.
>
>
>
> > ***"Could the authors comment on possible reasons why discrete state modeling is still outperformed by their continuous counterpart?''***
>
> We would like to refer the reviewer also to our answer to reviewer QzER. Overall, we believe there are multiple reasons for this. The first is that a large engineering effort has gone into optimizing diffusion models for continuous data. We reuse the same network architectures here, so there shouldn't be a difference in flexibility of generative parameterization or computation cost but they may be less optimized for the discrete data task. Secondly, the continuous state space inductive bias may slightly help model these data types that are very conducive to mapping from discrete space to the real line. We provide a generic method here for any discrete data type, we are not just proposing an image model, however, despite this we are still approaching becoming a viable alternative on images with our initial choices of forward process and network architecture. We have updated our limitations section to discuss this gap between the modelling of images as discrete or continuous data.

---

> > ### Comment · Reviewer_yvGD · 2022-08-09
> > **Response**
> >
> > Thank you for the answer. I will keep my recommendation for acceptance.

---

### Official Review · Reviewer_QzER · 2022-07-12

**Rating:** 7
**Confidence:** 4
**Soundness:** 4 excellent
**Presentation:** 3 good
**Contribution:** 4 excellent

**Summary:**

This paper studies discrete state denoising diffusion-based generative models, where the data to be modelled is discrete, categorical, etc. In these cases, the standard diffusion model approach, which assumes that the data is continuous and can be perturbed in a smooth and continuous manner, breaks down. Previous discrete state diffusion models relied on frameworks that used discrete, step-wise perturbations, this is, in each step, some states are flipped to other states. This paper, in contrast, proposes a continuous-time framework for discrete state diffusion models, leveraging *Continuous Time Markov Chains* (CTMCs). Intuitively, we can think of the approach as a diffusion that runs in continuous time, and at certain times states flip to other states. The frequency how often that happens is described by transition rates (rate matrices $R_t(\tilde{x},x')$). This seems like a very elegant and appropriate formalism for discrete state diffusion models. The paper shows how, using Bayes rule, a corresponding reverse generative CTMC can be derived. This generative CTMC isn't analytically tractable, but the paper derives an objective, a continuous-time ELBO, that can be used to learn an approximate generative CTMC. To efficiently simulate the learnt generative CTMC, the paper then further suggests to use $\tau$-leaping, a technique taken from the chemical physics literature. While $\tau$-leaping makes certain approximations, as a side the paper also derives a corrector scheme to essentially clean up errors made by $\tau$-leaping. Finally, the paper also derives some error bounds between the generated and ground truth distribution.

Experimentally, the paper analyses the approach on 3 generative tasks (a toy dataset, CIFAR10 image generation, symbolic music generation). The results are favourable compared to previous discrete-state diffusion models and provide insights into the different proposed components, this is, $\tau$-leaping and its approximations as well as the corrector scheme.

**Questions:**

I also have a few minor questions and comments:
- It's nice that the paper derived these errors bound. However, how useful are they in practice? How loose are those bounds? Would it be insightful to compute these bounds for the experiments that are run?
- It could make sense to add Fig. 4 from the Appendix into the main paper. The main paper doesn't have any larger or nice pipeline figure and this figure is actually quite insightful and helpful for the non-expert reader to get a very quick impression about what is happening in CTMCs.
- It could be helpful for the reader to provide some more intuitions about the difference between the different transition rate matrices $R(\tilde{x},x)$ and $r(\tilde{x}|x)$.
- While going over the appendix, in Proposition 7, $\psi_t(\tilde{\mathbf{x}}^{1:D}|\mathbf{x}_0^{1:D})$ in line 765 does not seem to be properly defined anywhere. $\psi_t(\tilde{\mathbf{x}}^{1:D}|\mathbf{x}_0^{1:D})$ sort of appears in the proposition but is then never really used.

**Limitations:**

Some limitations have been briefly mentioned at the end of the paper (slow sampling). Ethical considerations and potential negative societal impact are discussed in some detail in the appendix. I think these discussions are satisfactory and appropriate.

**Strengths And Weaknesses:**

**Strengths:**
- *Novelty and Originality*: As mentioned in the summary the paper proposes, derives, or suggests multiple technical innovations for discrete state diffusion models. This includes the general CTMC framework, the ELBO loss for training, the $\tau$-leaping sampling, and the error bounds. Some of the derivations are a bit related to similar derivations from regular continuous-time diffusion models, but overall I think the paper presents a lot of methodological novelty and is very original in that regard.
- *Clarity and Presentation*: The paper is overall well-written and nicely presented. It is a mathematically very dense paper and naturally requires very careful reading, but considering that it is relatively smooth to read. The authors make a good job in providing a lot of details and additional background in the appendix to help with that.
- *Significance and Impact:* Diffusion models have become a very popular and promising class of generative models, and also discrete state diffusion models are used more frequently. Considering that, I think this is a significant contribution, since the paper shows a fundamentally different, potentially better way, to set up discrete state diffusion models. I believe the method will find further usage and there will be follow-up papers, building on this paper and further improving the method.

**Weaknesses:**

Methodologically, I have no major concerns at all. However, the experimental results, while insightful, are overall not particularly impressive.
- Performance-wise, the method is still behind standard continuous state diffusion models on image data.
- It is also still very slow to sample from the learnt CTMC models. In that regard, it also seems $\tau$-leaping only generates really strong results when combined with the corrector scheme or when doing very small steps $\tau$.
- It would be interesting to consider more experiments on other data. The recent paper [1] tackles text generation, a very natural discrete state task. I would be very curious how the model performs here.
- The ELBO numbers in the CIFAR10 experiments (Table 1) are behind those of D3PM and also generally quite a bit worse than more modern diffusion models such as DDPM++ [2], which also leverage a continuous time framework. This might imply that the models have reduced generation diversity, as evidently some validation data has somewhat low probability under the learnt distribution.

**Conclusions and Summary:**
Overall, I think this is a high-quality paper that should be accepted. It presents a lot of new ideas and a lot of methodological novelty and demonstrates the advantages of the proposed method. The CTMC approach to discrete state diffusion models seems overall like a very sensible idea, and maybe the way to go in the future for this model class. On the other hand, the experiments seem appropriate to test the method, but the results aren't overly impressive when compared to the broader literature.

[1] Austin et al., "Structured Denoising Diffusion Models in Discrete State-Spaces", NeurIPS, 2021

[2] Song et al., "Score-Based Generative Modeling through Stochastic Differential Equations", ICLR, 2021

---

> ### Author Response · Authors · 2022-08-01
> **Response to Reviewer QzER (Part 1)**
>
> We would like to thank the reviewer for the thorough engagement with the work and detailed review. We greatly appreciate the summary that our work is high-quality and presents a lot of new ideas and methodological novelty. We address the specific comments and questions from the review here.
>
>  > ***"Performance-wise, the method is still behind standard continuous state diffusion models on image data.''***
>
> A large combined research effort has gone into modelling images as continuous data in the diffusion field. A lot of engineering effort has gone into architecture search and hyperparameter selection for this specific task, achieving very impressive results. In this work, we have directly utilized the same neural network architectures and aimed for a similar forward noising process as this continuous space work to give us a reasonable starting point for experiments. In the global model configuration space *when using discrete data*, these choices are likely sub-optimal. Despite this, it is nice to see that the gap between previous discrete state image modelling work and the state of the art continuous image modelling techniques can be significantly reduced when using the tau-leaping with corrector steps method. We have updated the limitations section to discuss the gap between continuous and discrete modelling of images.
>
> Many real world discrete datasets cannot be modelled using the mapping to a continuous space that current generative image models use. Our method is generic in that it can be applied to all these discrete data problems whilst still retaining reasonable performance on images where the inductive bias of a continuous state space process may slightly favour other methods. Applying the same engineering research effort to discrete models of images should further improve performance, fine tuning the forward noising process schedules and adapting the network architecture specifically for the discrete task.
>
>
> > ***"It is also still very slow to sample from the learnt CTMC models. In that regard, it also seems $\tau$-leaping only generates really strong results when combined with the corrector scheme or when doing very small steps $\tau$.''***
>
> The first iterations of diffusion models tend to be quite slow e.g. DDPM (Ho et al. 2020) and NCSN (Song et al. 2021) use 1000 or 2000 steps to sample the reverse process. In this work, we have also presented a new model class and picked a standard setup for the diffusion process, achieving similar speeds as these initial continuous space models. A lot of research effort has gone into improving sampling speed in continuous state spaces, largely helped by the additional insight into the reverse process gained through the continuous time / SDE framing. These are worthy contributions in their own right and we hope that our new framework can catalyse research in this direction for discrete state spaces too. For example, we hope that our link to chemical physics through tau-leaping will encourage experts in that field to join the research effort and provide additional insights into how we can make our sampling approaches more efficient in further work.
>
> > ***"It would be interesting to consider more experiments on other data. The recent paper [1] tackles text generation, a very natural discrete state task. I would be very curious how the model performs here.''***
>
> Text is indeed a very interesting application of discrete generative models, however, we do not have access to the resources required to appropriately compare to prior approaches since text models and datasets are larger than image/music models. In our work, we aimed to cover a variety of data modalities: images and monophonic music, that don't exceed our available resources and can still appropriately verify the method's performance.
>
> **References**
>
> Ho et al., "Denoising Diffusion Probabilistic Models", NeurIPS 2020
>
> Song et al., "Score-Based Generative Modeling through Stochastic Differential Equations", ICLR 2021

---

> > ### Author Response · Authors · 2022-08-01
> > **Response to Reviewer QzER (Part 2)**
> >
> > > ***"The ELBO numbers in the CIFAR10 experiments (Table 1) are behind those of D3PM ... ''***
> >
> > Thank you for the comment, we have updated the limitations section to discuss this comparison with D3PM further. We also note that some care should be taken when interpreting ELBO scores for diffusion models. The first thing to note is that the ELBO value is independent of the specific method used for sampling the reverse process e.g. with corrector steps or not. This changes the generated data distribution and so to assess the model's learnt distribution in terms of fidelity and diversity, we should really focus on the samples themselves as this includes the effect of the reverse process sampling method. We show a large selection of samples include Figure 5 in the Appendix and see there is good generational diversity. Also, our training objective is a proper bound on the data likelihood (i.e. not using a re-weighting scheme in the loss like DDPM) and so we have increased confidence in not dropping modes of $p_{data}$. The second thing to note is that we are reporting ELBO values in Table 1, so upper bounds on NLL, whereas the DDPM++ (Song et al. 2021) results mentioned are NLL values using the probability flow ODE therefore it is not a like for like comparison. Finally, we are not focusing on using our method as a likelihood model in this work and are more concerned with generating high quality samples from the data distribution.
> >
> > > ***"It's nice that the paper derived these errors bound. However, how useful are they in practice? How loose are those bounds? Would it be insightful to compute these bounds for the experiments that are run?''***
> >
> > The main purpose of our error bound is to show that for any error tolerance $\epsilon$ it is possible to find a choice of $T$, $M$ and $\tau$ such that the total variation error of our method is less than $\epsilon$. Moreover, the required choices of $T$, $M$ and $\tau$ suggested by the bound are computationally feasible; we require $T$ to be chosen on the order of $t_{\textup{mix}} \log (1/\epsilon) \log D$, $M$ to be on the order of $\epsilon/T$ and $\tau$ on the order of $\epsilon/(T(|R|SDC_1)^2)$, none of which collapses or explodes exponentially in the dimension $D$. This result is reassuring as it provides proof that our method can generate samples of arbitrarily high fidelity at computational cost growing only moderately with the dimensionality of the problem.
> >
> > However, beyond this we do not expect the bound of Theorem 1 to be particularly tight in practice, especially with respect to constants. The bounds of Propositions 5 and 6 used as ingredients to the proof are generally pretty loose, so it is likely too much to hope that our theorem provides a bound that is tight in practice.
> >
> > A further complication when using or trying to test the tightness of the bound in practice is that the constants $C_1$, $C_2$ and $M$ are not directly accessible. Indeed, $C_1$ and $C_2$ are determined by $S$ and $R_t$ according to Assumptions 1 and 2 respectively, but are not easy to calculate explicitly, while $M$ is determined by the accuracy of our neural network approximation which is again not known in practice.
> >
> > One may be able to investigate the asymptotics of the approximation error, for example by performing experiments to hold all variables constant except, say, varying the time $T$ that the CTMC is run for, or the accuracy $M$ of the neural network approximation, and seeing how the error changes. This may allow one to test whether the error behaves asymptotically as suggested by Theorem 1. We leave this to future work.

---

> > > ### Author Response · Authors · 2022-08-01
> > > **Response to Reviewer QzER (Part 3)**
> > >
> > > > ***"It could make sense to add Fig. 4 from the Appendix into the main paper ...''***
> > >
> > > > ***"It could be helpful for the reader to provide some more intuitions about the difference between the different transition rate matrices $R(\tilde{x}, x)$ and $r(\tilde{x} | x)$.''***
> > >
> > > Thank you for these suggestions for how to improve the clarity of the introduction to the concept of CTMCs within the main text. We would very much like to expand the introductory section at the start of Section 3 to include more intuitions and this figure given the extra space in a camera ready revision if accepted. The link between $R(\tilde{x}, x)$ and $r(x | \tilde{x})$ is indeed an important stepping stone in going from understanding discrete time Markov chains to understanding continuous time Markov chains. $r_t(x | \tilde{x})$ is the probability of transitioning from state $\tilde{x}$, to state $x$, at time $t$ *given that we already know a transition occurs at time $t$*. To know if a transition occurs, we need to run holding time simulations using the values contained within the rate matrix $R_t$ that specify the speed at which transitions occur. $r_t(x | \tilde{x})$ is a normalized version of $R_t$ because the transitions are more likely to occur between state pairs that have higher transition rates as specified by $R_t(\tilde{x}, x)$.
> > >
> > > > ***"While going over the appendix, in Proposition 7, $\psi_t(\tilde{x}^{1:D} | x_0^{1:D}) $ in line 765 does not seem to be properly defined anywhere. $\psi_t(\tilde{x}^{1:D} | x_0^{1:D})$ sort of appears in the proposition but is then never really used.''***
> > >
> > > We are just using $\psi_t(\tilde{x}^{1:D} | x_0^{1:D})$ to represent the $\tilde{x}^{1:D}$ marginal of the joint $q_{t|0}(x^{1:D} | x_0^{1:D}) r_t(\tilde{x}^{1:D} | x^{1:D})$ distribution and is only needed to take the expectation. We have made this clearer in an update to the appendix, thanks!

---

> > > > ### Comment · Reviewer_QzER · 2022-08-06
> > > > **Thank you for the reply**
> > > >
> > > > I would like to thank the authors for their detailed reply and acknowledge that I have read it. Overall, my impression of the paper hasn't changed and I still think that this is a high-quality work that should be accepted.

---

### Public Comment · ~Yeongbin_Seo1 · 2025-01-16
****Questions about Section B.2: Proof of Proposition 2****

1. The terms $ p(dx) $ and $ p(dw)$ appear in this section. What are their exact definitions?
Sometimes I’ve seen expressions like $ p(dx) = p(x) dx $. Based on the context, it seems like $  q_{T|0}(dx_T) = q_{T|0}(x_T | x_0) $, but could you clarify the precise definition?

2. There’s no definition given for $ w $ and $ \hat{W}$. What exactly do these terms represent?

3. In the third line, the term $ Q^{x_T} \\{ \hat{W}_0 = x_0 \\} $ appears on the right-hand side. Where does this term come from?
   It seems that the derivation would still hold without it.

3-2. Is the notation $ Q^{x_T}\\{\hat{W}_0 = x_0\\} $ correct?
   The superscript $ x_T $ typically indicates that the trajectory starts at $ x_T $, but if $ Q $ represents a forward path, it shouldn’t be starting from perfect noise like $ x_T $, right?
   Additionally, $ \hat{W}_0$ is a term from the reverse process, so can it be validly used within $ Q $?

4. In the fourth line, the integral $ \int \hat{R}^\theta_{T-s}(\hat{W}_s) ds $ appears.
Is this derived from the denominator of the earlier log term (i.e., $ d\hat{Q}^{x_T} $)?
If so, shouldn’t this integral also include a log?
If not, where does this integration term come from?

5. I would also appreciate if the exact definition and formulation or reference of the term 'path measure' (Q) is provided.

---

### Meta-Review · Area_Chair_C41k · 2022-08-25

**Recommendation:** Accept
**Confidence:** Certain

**Metareview:**

The work proposes a continuous-time generalization of diffusion models on a discrete space. The description uses continuous-time Markov chain (CTMC), in parallel to the existing stochastic differential equation description for continuous space. Reverse CTMC and modeling and ELBO objective are described. Some practical considerations and inspirations are also discussed, including avoiding exponentially large model in high dimensions, efficient reverse (generation) process simulation, and a corrector technique that further exploit the model to improve simulation (generation) quality. An error bound on the learned data distribution is also presented that shows a mild dependency on data dimensionality.

All the reviewers agree that this work presents the very right way to describe the continuous-time version of diffusion model on discrete space, and thereafter inspired techniques make a desired contribution to the community. Some concerns are raised, including still inferior performance than the continuous counterpart, and on the independence among dimensions. The authors provide reasonable remarks on them. Hence, I recommend accept to this paper.

One minor point: In Sec. 4.2, it would be clearer if the independence is specified both among the random variables $x^{1:D}$ in “output” and between each $x^d$ in “output” and $x^{1:D\backslash d}$ in “input”. Conventionally independence refers to the former, in which case the size is only reduced to $S^D \times D S^2$.


**Award:**

Yes

---

### Decision · Program_Chairs · 2022-09-14

Accept